# TabM: Advancing Tabular Deep Learning with Parameter-Efficient Ensembling

**Yury Gorishniy** [*]
Yandex

**Akim Kotelnikov**
HSE University, Yandex

**Artem Babenko**
Yandex

## Abstract

Deep learning architectures for supervised learning on tabular data range from simple multilayer perceptrons (MLP) to sophisticated Transformers and retrieval-augmented methods. This study highlights a major, yet so far overlooked opportunity for designing substantially better MLP-based tabular architectures. Namely, our new model TabM relies on *efficient ensembling*, where one TabM efficiently imitates an ensemble of MLPs and produces multiple predictions per object. Compared to a traditional deep ensemble, in TabM, the underlying implicit MLPs are trained simultaneously, and (by default) share most of their parameters, which results in significantly better performance and efficiency. Using TabM as a new baseline, we perform a large-scale evaluation of tabular DL architectures on public benchmarks in terms of both task performance and efficiency, which renders the landscape of tabular DL in a new light. Generally, we show that MLPs, including TabM, form a line of stronger and more practical models compared to attention- and retrieval-based architectures. In particular, we find that TabM demonstrates the best performance among tabular DL models. Then, we conduct an empirical analysis on the ensemble-like nature of TabM. We observe that the multiple predictions of TabM are weak individually, but powerful collectively. Overall, our work brings an impactful technique to tabular DL and advances the performance-efficiency trade-off with TabM — a simple and powerful baseline for researchers and practitioners. The code is available at: https://github.com/yandex-research/tabm.

## 1 Introduction

Supervised learning on tabular data is a ubiquitous machine learning (ML) scenario in a wide range of industrial applications. Among classic non-deep-learning methods, the state-of-the-art solution for such tasks is gradient-boosted decision trees (GBDT) (Prokhorenkova et al., 2018; Chen & Guestrin, 2016; Ke et al., 2017). Deep learning (DL) models for tabular data, in turn, are reportedly improving, and the most recent works claim to perform on par or even outperform GBDT on academic benchmarks (Hollmann et al., 2023; Chen et al., 2023b;a; Gorishniy et al., 2024).

However, from the practical perspective, it is unclear if tabular DL offers any obvious go-to baselines beyond simple architectures in the spirit of a multilayer perceptron (MLP). *First*, the scale and consistency of performance improvements of new methods w.r.t. simple MLP-like baselines are not always explicitly analyzed in the literature. Thus, one has to infer those statistics from numerous per-dataset performance scores, which makes it hard to reason about the progress. At the same time, due to the extreme diversity of tabular datasets, consistency is an especially valuable and hard-to-achieve property for a hypothetical go-to baseline. *Second*, efficiency-related properties, such as training time, and especially inference throughput, sometimes receive less attention. While methods are usually equally affordable on small-to-medium datasets (e.g. <100K objects), their applicability to larger datasets remains uncertain. *Third*, some recent work generally suggests that the progress on academic benchmarks may not transfer that well to real-world tasks (Rubachev et al., 2024). With all the above in mind, in this work, we thoroughly evaluate existing tabular DL methods and find that non-MLP models do not yet offer a convincing replacement for MLPs.

At the same time, we identify a previously overlooked path towards more powerful, reliable, and reasonably efficient tabular DL models. In a nutshell, we find that the parameter-efficient approach to

---

[*]The corresponding author: yurygorishniy@gmail.com

deep ensembling, where most weights are shared between ensemble members, allow one to make simple and strong tabular models out of plain MLPs. For example, MLP coupled with BatchEnsemble (Wen et al., 2020) — a long-existing method — right away outperforms popular attention-based models, such as FT-Transformer (Gorishniy et al., 2021), while being simpler and more efficient. This result alone suggests that efficient ensembling is a low-hanging fruit for tabular DL.

Our work builds on the above observations and offers TabM — a new powerful and practical model for researchers and practitioners. Drawing an informal parallel with GBDT (an ensemble of decision trees), TabM can also be viewed as a simple base model (MLP) combined with an ensembling-like technique, providing high performance and simple implementation at the same time.

**Main contributions.** We summarize our main contributions as follows:

1. We present TabM — a simple DL architecture for supervised learning on tabular data. TabM is based on MLP and parameter-efficient ensembling techniques closely related to BatchEnsemble (Wen et al., 2020). In particular, TabM produces **M**ultiple predictions per object. TabM easily competes with GBDT and outperforms prior tabular DL models, while being more efficient than attention- and retrieval-based DL architectures.
2. We provide a fresh perspective on tabular DL models in a large-scale evaluation along four dimensions: performance ranks, performance score distributions, training time, and inference throughput. One of our findings is that MLPs, including TabM, hit an appealing performance-efficiency tradeoff, which is not the case for attention- and retrieval-based models.
3. We show that the two key reasons for TabM's high performance are the collective training of the underlying implicit MLPs and the weight sharing. We also show that the multiple predictions of TabM are weak and overfitted individually, while their average is strong and generalizable.

## 2 RELATED WORK

**Decision-tree-based models.** Gradient-boosted decision trees (GBDT) (Chen & Guestrin, 2016; Ke et al., 2017; Prokhorenkova et al., 2018) is a strong and efficient baseline for tabular tasks. GBDT is a classic machine learning model, specifically, an ensemble of decision trees. Our model TabM is a deep learning model, specifically, a parameter-efficient ensemble of MLPs.

**Tabular deep learning architectures.** A large number of deep learning architectures for tabular data have been proposed over the recent years. That includes attention-based architectures (Song et al., 2019; Gorishniy et al., 2021; Somepalli et al., 2021; Kossen et al., 2021; Yan et al., 2023), retrieval-augmented architectures (Somepalli et al., 2021; Kossen et al., 2021; Gorishniy et al., 2024; Ye et al., 2024), MLP-like models (Gorishniy et al., 2021; Klambauer et al., 2017; Wang et al., 2020) and others (Arik & Pfister, 2020; Popov et al., 2020; Chen et al., 2023b; Marton et al., 2024; Hollmann et al., 2023). Compared to prior work, the key difference of our model TabM is its computation flow, where one TabM imitates an ensemble of MLPs by producing multiple independently trained predictions. Prior attempts to bring ensemble-like elements to tabular DL (Badirli et al., 2020; Popov et al., 2020) were not found promising (Gorishniy et al., 2021). Also, being a simple feed-forward MLP-based model, TabM is significantly more efficient than some of the prior work. Compared to attention-based models, TabM does not suffer from quadratic computational complexity w.r.t. the dataset dimensions. Compared to retrieval-based models, TabM is easily applicable to large datasets.

**Improving tabular MLP-like models.** Multiple recent studies achieved competitive performance with MLP-like architectures on tabular tasks by applying architectural modifications (Gorishniy et al., 2022), regularizations (Kadra et al., 2021; Jeffares et al., 2023a; Holzmüller et al., 2024), custom training techniques (Bahri et al., 2021; Rubachev et al., 2022). Thus, it seems that tabular MLPs have good potential, but one has to deal with overfitting and optimization issues to reveal that potential. Our model TabM achieves high performance with MLP in a different way, namely, by using it as the base backbone in a parameter-efficient ensemble in the spirit of BatchEsnsemble (Wen et al., 2020). Our approach is orthogonal to the aforementioned training techniques and architectural advances.

**Deep ensembles.** In this paper, by a deep ensemble, we imply multiple DL models of the same architecture trained independently (Jeffares et al., 2023b) for the same task under different random seeds (i.e. with different initializations, training batch sequences, etc.). The prediction of a deep ensemble is the mean prediction of its members. Deep ensembles often significantly outperform single DL models of the same architecture (Fort et al., 2020) and can excel in other tasks like uncertainty

estimation or out-of-distribution detection (Lakshminarayanan et al., 2017). It was observed that individual members of deep ensembles can learn to extract diverse information from the input, and the power of deep ensembles depends on this diversity (Allen-Zhu & Li, 2023). The main drawback of deep ensembles is the cost and inconvenience of training and using multiple models.

**Parameter-efficient deep "ensembles".** To achieve the performance of deep ensembles at a lower cost, multiple studies proposed architectures that imitate ensembles by producing multiple predictions with one model (Lee et al., 2015; Zhang et al., 2020; Wen et al., 2020; Havasi et al., 2021; Antorán et al., 2020; Turkoglu et al., 2022). Such models can be viewed as "ensembles" where the implicit ensemble members share a large amount of their weights. There are also non-architectural approaches to efficient ensembling, e.g. FGE (Garipov et al., 2018), but we do not explore them, because we are interested specifically in architectural techniques. In this paper, we highlight parameter-efficient ensembling as an impactful paradigm for tabular DL. In particular, we describe two simple variations of BatchEnsemble (Wen et al., 2020) that are highly effective for tabular MLPs. One variation uses a more efficient parametrization, and another one uses an improved initialization.

## 3 TABM

In this section, we present TabM — a **Tab**ular DL model that makes **M**ultiple predictions.

### 3.1 PRELIMINARIES

**Notation.** We consider classification and regression tasks on tabular data. $x$ and $y$ denote the features and a label, respectively, of one object from a given dataset. A machine learning model takes $x$ as input and produces $\hat{y}$ as a prediction of $y$. $N \in \mathbb{N}$ and $d \in \mathbb{N}$ respectively denote the "depth" (e.g. the number of blocks) and "width" (e.g. the size of the latent representation) of a given neural network. $d_y \in \mathbb{N}$ is the output representation size (e.g. $d_y = 1$ for regression tasks, and $d_y$ equals the number of classes for classification tasks).

**Datasets.** Our benchmark consists of 46 publicly available datasets used in prior work, including Grinsztajn et al. (2022); Gorishniy et al. (2024); Rubachev et al. (2024). The main properties of our benchmark are summarized in Table 1, and more details are provided in Appendix C.

Table 1: The overview of our benchmark. The "Split type" property is explained in the text.

| #Datasets | Train size | | | | #Features | | | | Task type | | Split type | |
|---|---|---|---|---|---|---|---|---|---|---|---|---|
| | Min. | Q50 | Mean | Max. | Min. | Q50 | Mean | Max. | #Regr. | #Classif. | Random | Domain-aware |
| 46 | 1.8K | 12K | 76K | 723K | 3 | 20 | 108 | 986 | 28 | 18 | 37 | 9 |

**Domain-aware splits.** We pay extra attention to datasets with what we call "domain-aware" splits, including the eight datasets from the TabReD benchmark (Rubachev et al., 2024) and the Microsoft dataset (Qin & Liu, 2013). For these datasets, their original real-world splits are available, e.g. time-aware splits as in TabReD. Such datasets were shown to be challenging for some methods because they naturally exhibit a certain degree of distribution shift between training and test parts (Rubachev et al., 2024). The random splits of the remaining 37 datasets are inherited from prior work.

**Experiment setup.** We use the setup from Gorishniy et al. (2024), and describe it in detail in subsection D.2. Most importantly, on each dataset, a given model undergoes hyperparameter tuning on the *validation* set, then the tuned model is trained from scratch under multiple random seeds, and the *test* metric averaged over the random seeds becomes the final score of the model on the dataset.

**Metrics.** We use RMSE (the root mean square error) for regression tasks, and accuracy or ROC-AUC for classification tasks depending on the dataset source. See subsection D.3 for details.

Also, throughout the paper, we often use the relative performance of models w.r.t. MLP as the key metric. This metric gives a unified perspective on all tasks and allows reasoning about the scale of improvements w.r.t. to a simple baseline (MLP). Formally, on a given dataset, the metric is defined as $\left(\frac{\text{score}}{\text{baseline}} - 1\right) \cdot 100\%$, where "score" is the metric of a given model, and "baseline" is the metric of MLP. In this computation, for regression tasks, we convert the raw metrics from RMSE to $R^2$ to better align the scales of classification and regression metrics.

### 3.2 A QUICK INTRODUCTION TO BATCHENSEMBLE.

For a given architecture, let's consider any linear layer $l$ in it: $l(x) = Wx + b$, where $x \in \mathbb{R}^{d_1}$, $W \in \mathbb{R}^{d_2 \times d_1}$, $b \in \mathbb{R}^{d_2}$. To simplify the notation, let $d_1 = d_2 = d$. In a traditional deep ensemble, the $i$-th member has its own set of weights $W_i, b_i$ for this linear layer: $l_i(x_i) = W_i x_i + b_i$, where $x_i$ is the object representation within the $i$-th member. By contrast, in BatchEnsemble, this linear layer is either (1) fully shared between all members, or (2) mostly shared: $l_i(x_i) = s_i \odot (W(r_i \odot x_i)) + b_i$, where $\odot$ is the elementwise multiplication, $W \in \mathbb{R}^{d \times d}$ is shared between all members, and $r_i, s_i, b_i \in \mathbb{R}^d$ are *not* shared between the members. This is equivalent to defining the $i$-th weight matrix as $W_i = W \odot (s_i r_i^T)$. To ensure diversity of the ensemble members, $r_i$ and $s_i$ of all members are initialized randomly with $\pm 1$. All other layers are fully shared between the members of BatchEnsemble.

The described parametrization allows packing all ensemble members in one model that simultaneously takes $k$ objects as input, and applies all $k$ implicit members in parallel, without explicitly materializing each member. This is achieved by replacing one or more linear layers of the original neural network with their BatchEnsemble versions: $l_{\text{BE}}(X) = ((X \odot R)W) \odot S + B$, where $X \in \mathbb{R}^{k \times d}$ stores $k$ object representations (one per member), and $R, S, B \in \mathbb{R}^d$ store the non-shared weights $(r_i, s_i, b_i)$ of the members, as shown at the lower left part of Figure 1.

**Terminology.** In this paper, we call $r_i, s_i, b_i, R, S$ and $B$ *adapters*, and the implicit members of parameter-efficient emsembles (e.g. BatchEnsemble) — *implicit submodels* or simply *submodels*.
**Overhead to the model size.** With BatchEnsemble, adding a new ensemble member means adding only one row to each of the matrices $R, S$, and $B$, which results in $3d$ new parameters per layer. For typical values of $d$, this is a negligible overhead to the original layer size $d^2 + d$.
**Overhead to the runtime.** Thanks to the modern hardware, the large number of shared weights and the parallel execution of the $k$ forward passes, the runtime overhead of BatchEnsemble can be (significantly) lower than $\times k$ (Wen et al., 2020). Intuitively, if the original workload underutilizes the hardware, there are more chances to pay less than $\times k$ overhead.

### 3.3 ARCHITECTURE

TabM is one model representing an ensemble of $k$ MLPs. Contrary to conventional deep ensembles, in TabM, the $k$ MLPs are trained in parallel and share most of their weights by default, which leads to better performance and efficiency. We present multiple variants of TabM that differ in their weight-sharing strategies, where TabM and TabM$_{\text{mini}}$ are the most effective variants, and TabM$_{\text{packed}}$ is a conceptually important variant potentially useful in some cases. We obtain our models in several steps, starting from essential baselines. We always use the ensemble size $k = 32$ and analyze this hyperparameter in subsection 5.3. In subsection A.1, we explain that using MLP as the base model is crucial because of its excellent efficiency.

**MLP.** We define MLP as a sequence of $N$ simple blocks followed by a linear prediction head: $\text{MLP}(x) = \text{Linear}(\text{Block}_N(\ldots(\text{Block}_1(x)))$, where $\text{Block}_i(x) = \text{Dropout}(\text{ReLU}(\text{Linear}((x))))$.

**MLP$^{\times k}$ = MLP + Deep Ensemble.** We denote the traditional deep ensemble of $k$ independently trained MLPs as MLP$^{\times k}$. To clarify, this means tuning hyperparameters of one MLP, then independently training $k$ tuned MLPs under different random seeds, and then averaging their predictions. The performance of MLP$^{\times k}$ is reported in Figure 2. Notably, the results are already better and more stable than those of FT-Transformer (Gorishniy et al., 2021) — the popular attention-based baseline.

Although the described approach is a somewhat default way to implement an ensemble, it is not optimized for the task performance of the ensemble. First, for each of the $k$ MLPs, the training is stopped based on the individual validation score, which is optimal for each individual MLP, but can be suboptimal for their ensemble. Second, the hyperparameters are also tuned for one MLP without knowing about the subsequent ensembling. All TabM variants are free from these issues.

**TabM$_{\text{packed}}$ = MLP + Packed-Ensemble.** As the first step towards better and more efficient ensembles of MLPs, we implement $k$ MLPs as one large model using Packed-Ensemble (Laurent et al., 2023). This results in TabM$_{\text{packed}}$ illustrated in Figure 1. As an architecture, TabM$_{\text{packed}}$ is equivalent to MLP$^{\times k}$ and stores $k$ independent MLPs without any weight sharing. However, the critical difference is that TabM processes $k$ inputs in parallel, which means that one training step of TabM consists of $k$ parallel training steps of the individual MLPs. This allows monitoring the

performance of the ensemble during the training and stopping the training when it is optimal for the whole ensemble, not for individual MLPs. As a consequence, this also allows tuning hyperparameters for TabM$_{packed}$ as for one model. As shown in Figure 2, TabM$_{packed}$ delivers significantly better performance compared to MLP$^{\times k}$. Efficiency-wise, for typical depth and width of MLPs, the runtime overhead of TabM$_{packed}$ is noticeably less than $\times k$ due to the parallel execution of the $k$ forward passes on the modern hardware. Nevertheless, the $\times k$ overhead of TabM$_{packed}$ to the model size motivates further exploration.

**TabM$_{naive}$ = MLP + BatchEnsemble.** To reduce the size of TabM$_{packed}$, we now turn to weight sharing between the MLPs, and naively apply BatchEnsemble (Wen et al., 2020) instead of Packed-Ensemble, as described in subsection 3.2. This gives us TabM$_{naive}$— a preliminary version of TabM. In fact, the architecture (but not the initialization) of TabM$_{naive}$ is already equivalent to that of TabM, so Figure 1 is applicable. Interestingly, Figure 2 reports higher performance of TabM$_{naive}$ compared to TabM$_{packed}$. Thus, constraining the ensemble with weight sharing turns out to be a highly effective regularization on tabular tasks. The alternatives to BatchEnsemble are discussed in subsection A.1.

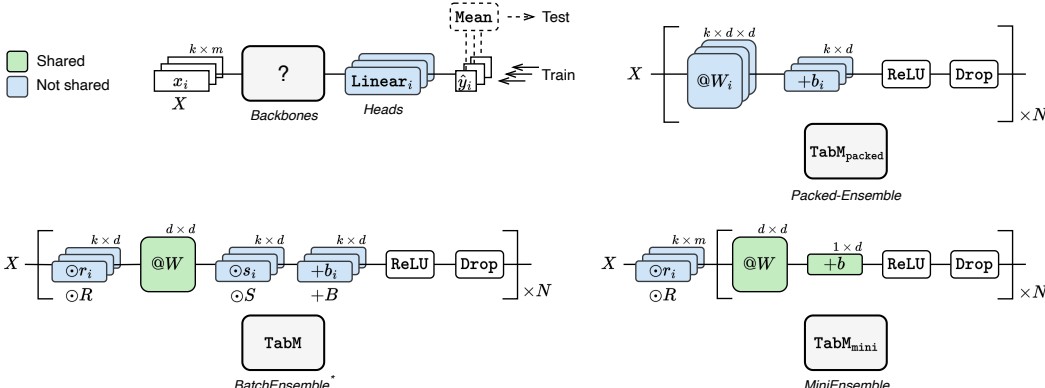

Figure 1: *(Upper left)* A high-level illustration of TabM. One TabM represents an ensemble of $k$ MLPs processing $k$ inputs in parallel. The remaining parts of the figure are three different parametrizations of the $k$ MLP backbones. *(Upper right)* TabM$_{packed}$ consists of $k$ fully independent MLPs. *(Lower left)* TabM is obtained by injecting three non-shared adapters $R$, $S$, $B$ in each of the $N$ linear layers of *one* MLP (* the initialization differs from Wen et al. (2020)). *(Lower right)* TabM$_{mini}$ is obtained by keeping only the very first adapter $R$ of TabM and removing the remaining $3N - 1$ adapters. *(Details)* Input transformations such as one-hot-encoding or feature embeddings (Gorishniy et al., 2022) are omitted for simplicity. `Drop` denotes dropout (Srivastava et al., 2014).

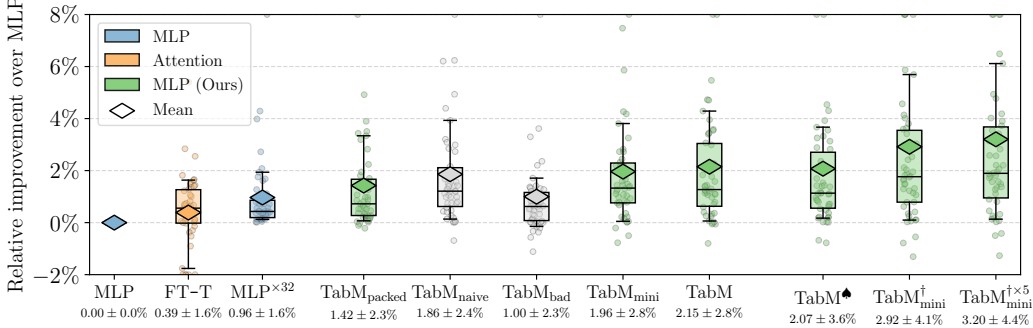

Figure 2: The performance of models described in subsection 3.3 on 46 datasets from Table 1; plus several baselines on the left. For a given model, one dot on a jitter plot describes the performance score on one of the 46 datasets. The box plots describe the percentiles of the jitter plots: the boxes describe the 25th, 50th, and 75th percentiles, and the whiskers describe the 10th and 90th percentiles. Outliers are clipped. The numbers at the bottom are the mean and standard deviations over the jitter plots. For each model, hyperparameters are tuned. "Model$^{\times k}$" denotes an ensemble of $k$ models.

**TabM$_{\text{mini}}$ = MLP + MiniEnsemble.** By construction, the just discussed TabM$_{\text{naive}}$ (illustrated as "TabM" in Figure 1) has $3N$ adapters: $R$, $S$ and $B$ in each of the $N$ blocks. Let's consider the very first adapter, i.e. the first adapter $R$ in the first linear layer. Informally, its role can be described as mapping the $k$ inputs living in the same representation space to $k$ different representation spaces *before* the tabular features are mixed with $@W$ for the first time. A simple experiment reveals that this adapter is critical. First, we remove it from TabM$_{\text{naive}}$ and keep the remaining $3N - 1$ adapters untouched, which gives us TabM$_{\text{bad}}$ with worse performance, as shown in Figure 2. Then, we do the opposite: we keep only the very first adapter of TabM$_{\text{naive}}$ and remove the remaining $3N - 1$ adapters, which gives us TabM$_{\text{mini}}$ — the minimal version of TabM. TabM$_{\text{mini}}$ is illustrated in Figure 1, where we call the described approach "MiniEnsemble". Figure 2 shows that TabM$_{\text{mini}}$ performs even slightly better than TabM$_{\text{naive}}$, despite having only one adapter instead of $3N$ adapters.

**TabM = MLP + BatchEnsemble + Better initialization.** The just obtained results motivate the next step. We go back to the architecture of TabM$_{\text{naive}}$ with all $3N$ adapters, but initialize all multiplicative adapters $R$ and $S$, except for the very first one, deterministically with 1. As such, at initialization, the deterministically initialized adapters have no effect, and the model behaves like TabM$_{\text{mini}}$, but these adapters are free to add more expressivity during training. This gives us TabM, illustrated in Figure 1. Figure 2 shows that TabM is the best variation so far.

**Hyperparameters.** Compared to MLP, the only new hyperparameter of TabM is $k$ — the number of implicit submodels. We heuristically set $k = 32$ and do not tune this value. We analyze the influence of $k$ in subsection 5.3. We also share additional observations on the learning rate in subsection A.3.

**Limitations and practical considerations** are commented in subsection A.4.

### 3.4 IMPORTANT PRACTICAL MODIFICATIONS OF TABM

♠ $\sim$ **Shared training batches**. Recall that the order of training objects usually varies between ensemble members, because of the random shuffling with different seeds. For TabM, in terms of Figure 1, that corresponds to $X$ storing $k$ different training objects $\{x_i\}_{i=1}^{k}$. We observed that reusing the training batches between the TabM's submodels results in only minor performance loss on average (depending on a dataset), as illustrated with TabM♠ in Figure 2. In practice, due to the simpler implementation and better efficiency, sharing training batches can be a reasonable starting point.

† $\sim$ **Non-linear feature embeddings**. In Figure 2, TabM$_{\text{mini}}^{\dagger}$ denotes TabM$_{\text{mini}}$ with non-linear feature embeddings from (Gorishniy et al., 2022), which demonstrates the high utility of feature embeddings for TabM. Specifically, we use a slightly modified version of the piecewise-linear embeddings (see subsection D.8 for details).

×N $\sim$ **Deep ensemble**. In Figure 2, TabM$_{\text{mini}}^{\dagger \times 5}$ denotes an ensemble of five independent TabM$_{\text{mini}}^{\dagger}$ models, showing that TabM itself can benefit from the conventional deep ensembling.

### 3.5 SUMMARY

The story behind TabM shows that technical details of *how* to construct and train an ensemble have a major impact on task performance. Most importantly, we highlight simultaneous training of the (implicit) ensemble members and weight sharing between them. The former is responsible for the ensemble-aware stopping of the training, and the latter apparently serves as a form of regularization.

## 4 EVALUATING TABULAR DEEP LEARNING ARCHITECTURES

Now, we perform an empirical comparison of many tabular models, including TabM.

### 4.1 BASELINES

In the main text, we use the following baselines: MLP (defined in subsection 3.3), FT-Transformer denoted as "FT-T" (the attention-based model from Gorishniy et al. (2021)), SAINT (the attention- and retrieval-based model from Somepalli et al. (2021)), T2G-Former denoted as "T2G" (the attention-based model from Yan et al. (2023)), ExcelFormer denoted as "Excel" (the attention-based model from Chen et al. (2023a)), TabR (the retrieval-based model from Gorishniy et al. (2024)), ModernNCA

denoted as "MNCA" (the retrieval-based model from Ye et al. (2024)) and GBDT, including XGBoost (Chen & Guestrin, 2016), LightGBM (Ke et al., 2017) and CatBoost (Prokhorenkova et al., 2018).

The models with non-linear feature embeddings from Gorishniy et al. (2022) are marked with † or ‡ depending on the embedding type (see subsection D.8 for details on feature embeddings):

- MLP† and TabM$^{\dagger}_{mini}$ use a modified version of the piecewise-linear embeddings.
- TabR‡, MNCA‡, and MLP‡ (also known as MLP-PLR) use various periodic embeddings.

More baselines are evaluated in Appendix B. Implementation details are provided in Appendix D.

## 4.2 TASK PERFORMANCE

We evaluate all models following the protocol announced in subsection 3.1 and report the results in Figure 3 (see also the critical difference diagram in Figure 9). We make the following observations:

1. The performance ranks render TabM as the top-tier DL model.
2. The middle and right parts of Figure 3 provide a fresh perspective on the per-dataset metrics. TabM holds its leadership among the DL models. Meanwhile, many DL methods turn out to be no better or even worse than MLP on a non-negligible number of datasets, which shows them as less reliable solutions, and changes the ranking, especially on the domain-aware splits (right).
3. One important characteristic of a model is the *weakest* part of its performance profile (e.g. the 10th or 25th percentiles in the middle plot) since it shows how reliable the model is on "inconvenient" datasets. From that perspective, MLP† seems to be a decent practical option between the plain MLP and TabM, especially given its simplicity and efficiency compared to retrieval-based alternatives, such as TabR and ModernNCA.

**Summary.** TabM confidently demonstrates the best performance among tabular DL models, and can serve as a reliable go-to DL baseline. This is not the case for attention- and retrieval-based models. Overall, MLP-like models, including TabM, form a representative set of tabular DL baselines.

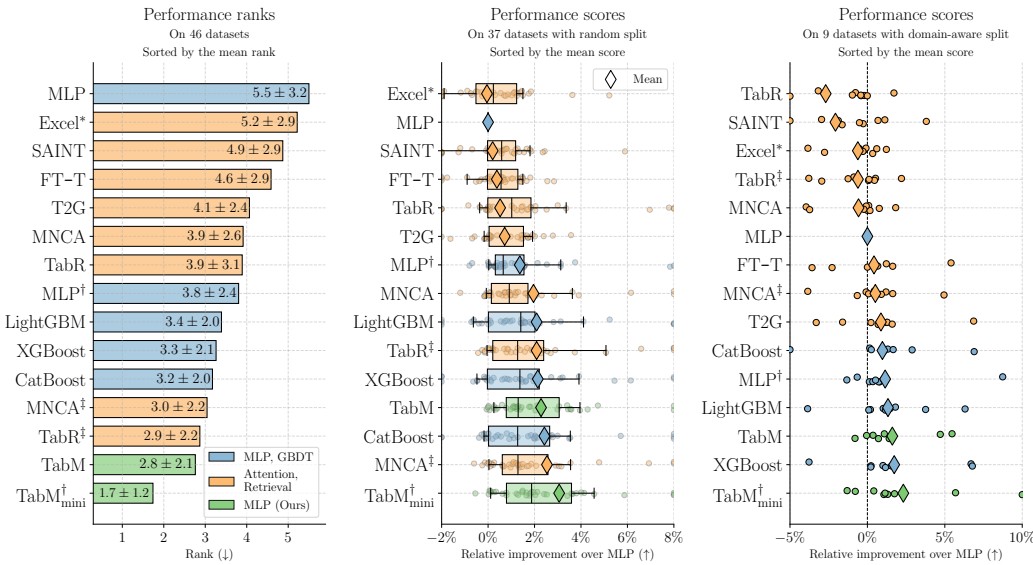

Figure 3: The task performance of tabular models on the 46 datasets from Table 1. *(Left)* The mean and standard deviations of the performance ranks over all datasets summarize the head-to-head comparison between the models on all datasets. *(Middle & Right)* The relative performance w.r.t. the plain multilayer perceptron (MLP) allows reasoning about the scale and consistency of improvements over this simple baseline. One dot of a jitter plot corresponds to the performance of a model on one of the 46 datasets. The box plots visualize the 10th, 25th, 50th, 75th, and 90th percentiles of the jitter plots. Outliers are clipped. The separation in random and domain-aware dataset splits is explained in subsection 3.1. (*Evaluated under the common protocol without data augmentations)

## 4.3 EFFICIENCY

Now, we evaluate tabular models in terms of training and inference efficiency, which becomes a serious reality check for some of the methods. We benchmark exactly those hyperparameter configurations of models that are presented in Figure 3 (see subsection B.3 for the motivation).

**TabM$_{mini}^{†*}$ & TabM$_{mini}^{†♠*}$.** Additionally, in this section, we mark with the asterisk (*) the versions of TabM enhanced with two efficiency-related plugins available out-of-the-box in PyTorch (Paszke et al., 2019): the automatic mixed precision (AMP) and `torch.compile` (Ansel et al., 2024). The purpose of those TabM variants is to showcase the potential of the modern hardware and software for a powerful tabular DL model, and they should not be directly compared to other DL models. However, the implementation simplicity of TabM plays an important role, because it facilitates the seamless integration of the aforementioned PyTorch plugins.

**Training time.** We focus on training times on larger datasets, because on small datasets, all methods become almost equally affordable, regardless of the formal relative difference. Nevertheless, in Figure 10, we provide measurements on small datasets as well. The left side of Figure 4 reveals that TabM offers practical training times. By contrast, the long training times of attention- and retrieval-based models become one more limitation of these methods.

**Inference throughput.** The right side of Figure 4 tells essentially the same story as the left side. In subsection B.3, we also report the inference throughput on GPU with large batch sizes.

**Applicability to large datasets.** In Table 2, we report metrics on two large datasets. As expected, attention- and retrieval-based models struggle, yielding extremely long training times, or being simply inapplicable without additional effort. See subsection D.4 for implementation details.

**Parameter count.** Most tabular networks are overall compact. This, in particular, applies to TabM, because its size is by design comparable to MLP. We report model sizes in subsection B.3.

**Summary.** Simple MLPs are the fastest DL models, with TabM being the runner-up. The attention- and retrieval-based models are significantly slower. Overall, MLP-like models, including TabM, form a representative set of practical and accessible tabular DL baselines.

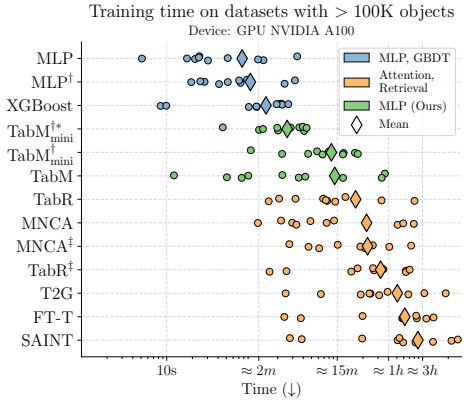 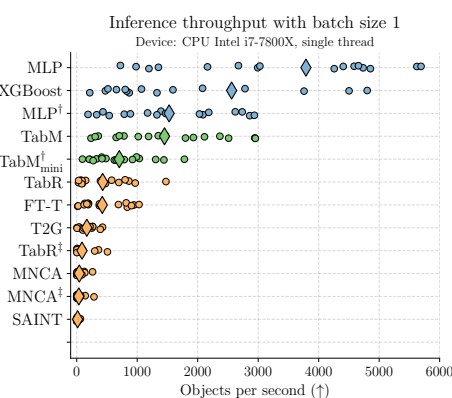

Figure 4: Training times (left) and inference throughput (right) of the models from Figure 3. One dot represents a measurement on one dataset. TabM$_{mini}^{†*}$ is the optimized TabM$_{mini}^{†}$ (see subsection 4.3).

Table 2: RMSE (upper rows) and training times (lower rows) on two large datasets. The best values are in bold. The meaning of model colors follows Figure 3.

| | #Objects | #Features | XGBoost | MLP | TabM$_{mini}^{†♠*}$ | TabM$_{mini}^{†}$ | FT-T | TabR |
|---|---|---|---|---|---|---|---|---|
| Maps Routing | 6.5M | 986 | 0.1601 | 0.1592 | 0.1583 | **0.1582** | 0.1594 | OOM |
| | | | 28m | **15m** | 2h | 13.5h | 45.5h | |
| Weather | 13M | 103 | 1.4234 | 1.4842 | **1.4090** | **1.4112** | 1.4409 | OOM |
| | | | **10m** | 15m | 1.3h | 3.3h | 13.5h | |

## 5 ANALYSIS

### 5.1 PERFORMANCE AND TRAINING DYNAMICS OF THE INDIVIDUAL SUBMODELS

Recall that the prediction of TabM is defined as the mean prediction of its $k$ implicit submodels that share most of their weights. In this section, we take a closer look at these submodels.

For the next experiment, we intentionally simplify the setup as described in detail in subsection D.5. Most importantly, all models have the same depth 3 and width 512, and are trained without early stopping, i.e. the training goes beyond the optimal epochs. We use TabM$_{mini}$ from Figure 1 with $k = 32$ denoted as TabM$_{mini}^{k=32}$. We use TabM$_{mini}^{k=1}$ (i.e. essentially one plain MLP) as a natural baseline for the submodels of TabM$_{mini}^{k=32}$, because each of the 32 submodels has the architecture of TabM$_{mini}^{k=1}$.

We visualize the training profiles on four diverse datasets (two classification and two regression problems of different sizes) in Figure 5. As a reminder, the mean of the $k$ **individual** losses is what is explicitly optimized during the training of TabM$_{mini}$, the loss of the **collective** mean prediction corresponds to how TabM$_{mini}$ makes predictions on inference, and TabM$_{mini}^{k=1}$ is just a **baseline**.

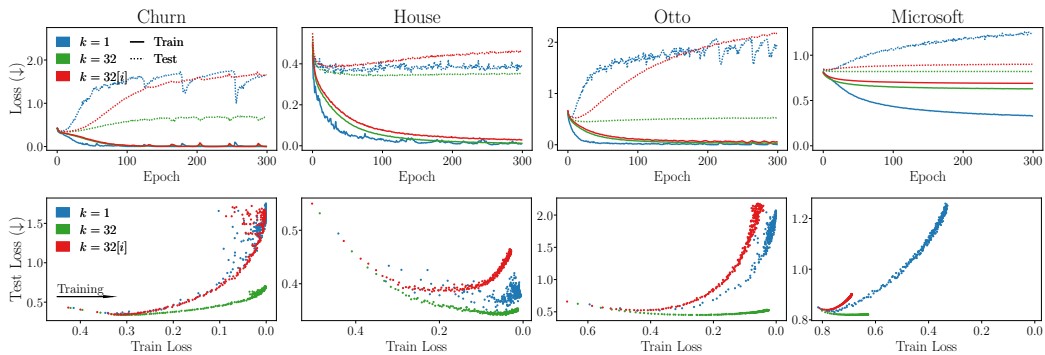

Figure 5: The training profiles of TabM$_{mini}^{k=32}$ and TabM$_{mini}^{k=1}$ as described in subsection 5.1. *(Upper)* The training curves. $k = 32[i]$ represents the mean **i**ndividual loss over the 32 submodels. *(Lower)* Same as the first row, but in the train-test coordinates: each dot represents some epoch from the first row, and the training generally goes from left to right. This allows reasoning about overfitting by comparing test loss values for a given train loss value.

In the upper row of Figure 5, the collective mean prediction of the submodels is superior to their individual predictions in terms of both training and test losses. After the initial epochs, the training loss of the baseline MLP is lower than that of the collective and individual predictions.

In the lower row of Figure 5, we see a stark contrast between the individual and collective performance of the submodels. Compared to the baseline MLP, the submodels look overfitted individually, while their collective prediction exhibits substantially better generalization. This result is strict evidence of a non-trivial diversity of the submodels: without that, their collective test performance would be similar to their individual test performance. Additionally, we report the performance of the **B**est submodel of TabM across many datasets under the name TabM[B] in Figure 6. As such, individually, even the best submodel of TabM is no better than a simple MLP.

**Summary.** TabM draws its power from the collective prediction of weak, but diverse submodels.

### 5.2 SELECTING SUBMODELS AFTER TRAINING

The design of TabM allows selecting only a subset of submodels after training based on any criteria, simply by pruning extra prediction heads and the corresponding rows of the adapter matrices. To showcase this mechanics, after the training, we **G**reedily construct a subset of TabM's submodels with the best collective performance on the validation set, and denote this "pruned" TabM as TabM[G]. The performance reported in Figure 6 shows that TabM[G] is slightly behind the vanilla TabM. On average over 46 datasets, the greedy submodel selection results in $8.8 \pm 6.6$ submodels out of the initial $k = 32$, which can result in faster inference. See subsection D.6 for implementation details.

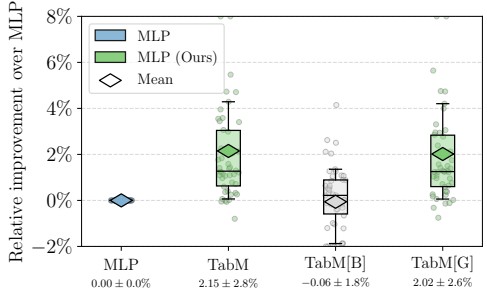
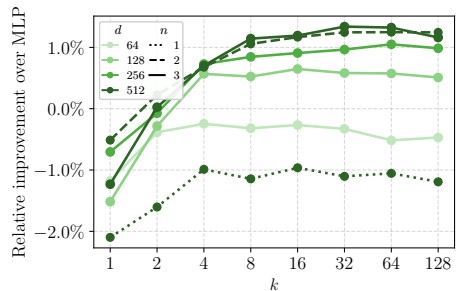

Figure 6: The performance on the 46 datasets from Table 1. TabM[B] and TabM[G] are described in subsection 5.1 and subsection 5.2.

Figure 7: The average performance of TabM with $n$ layers of the width $d$ across 17 datasets as a function of $k$.

### 5.3 HOW DOES THE PERFORMANCE OF TABM DEPEND ON $k$?

To answer the question in the title, we consider TabM with $n$ layers of the size $d$ and different values of $k$, and report the average performance over multiple datasets in Figure 7 (the implementation details are provided in subsection D.7). The solid curves correspond to $n = 3$, and the dark green curves correspond to $d = 512$. Our main observations are as follows. *First,* it seems that the "larger" TabM is (i.e. when $n$ and $d$ increase), the more submodels it can accommodate effectively. For example, note how the solid curves corresponding to different $d$ diverge at $k = 2$ and $k = 4$. *Second,* too high values of $k$ can be detrimental. Perhaps, weight sharing limits the number of submodels that can productively "coexist" in one network, despite the presence of non-shared adapters. *Third,* too narrow ($d = 64$) or too shallow ($n = 1$) configurations of TabM can lead to suboptimal performance, at least in the scope of middle-to-large datasets considered in this work.

### 5.4 PARAMETER-EFFICIENT ENSEMBLING REDUCES THE NUMBER OF DEAD NEURONS

Here, we show empirically that the design of TabM naturally leads to higher utilization of the backbone's weights. Even without technical definitions, this sounds intuitive, since TabM has to implement $k$ (diverse) computations using the amount of weights close to that of one MLP.

Let's consider TabM$_{mini}$ as illustrated in Figure 1. By design, each of the shared neurons of TabM$_{mini}$ is used $k$ times per forward pass, where "neuron" refers to the combination of the linear transformation and the subsequent nonlinearity (e.g. ReLU). By contrast, in plain MLP (or in TabM$_{mini}$ with $k = 1$), each neuron is used only once per forward pass. Thus, technically, a neuron in TabM$_{mini}$ has more chances to be activated, which overall may lead to lower portion of dead neurons in TabM$_{mini}$ compared to MLP (a dead neuron is a neuron that never activates, and thus has no impact on the prediction). Using the experiment setup from subsection 5.1, we compute the portion of dead neurons in TabM$_{mini}$ using its best validation checkpoint. On average across 46 datasets, for $k = 1$ and $k = 32$, we get $0.29 \pm 0.17$ and $0.14 \pm 0.09$ portion of dead neurons, respectively, which is in line with the described intuition. Technically, on a given dataset, this metric is computed as the percentage of neurons that never activate on a fixed set of 2048 training objects.

## 6 CONCLUSION & FUTURE WORK

In this work, we have demonstrated that tabular multilayer perceptrons (MLPs) greatly benefit from parameter-efficient ensembling. Using this insight, we have developed TabM — a simple MLP-based model with state-of-the-art performance. In a large-scale comparison with many tabular DL models, we have demonstrated that TabM is ready to serve as a new powerful and efficient tabular DL baseline. Along the way, we highlighted the important technical details behind TabM and discussed the individual performance of the implicit submodels underlying TabM.

One idea for future work is to bring the power of (parameter-)efficient ensembles to other, non-tabular, domains with optimization-related challenges and, ideally, lightweight base models. Another idea is to evaluate TabM for uncertainty estimation and out-of-distribution (OOD) detection on tabular data, which is inspired by works like Lakshminarayanan et al. (2017).

**Reproducibility statement.** The code is provided in the following repository: link. It contains the implementation of TabM, hyperparameter tuning scripts, evaluation scripts, configuration files with hyperparameters (the TOML files in the `exp/` directory), and the report files with the main metrics (the JSON files in the `exp/` directory). In the paper, the model is described in section 3, and the implementation details are provided in Appendix D.

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

# A  ADDITIONAL DISCUSSION ON TABM

## A.1  MOTIVATION

**Why BatchEnsemble?**  Among relatively ease-to-use "efficient ensembling" methods, beyond BatchEnsemble, there are examples such as dropout ensembles (Lakshminarayanan et al., 2017), naive multi-head architectures, TreeNet (Lee et al., 2015). However, in the literature, they were consistently outperformed by more advanced methods, including BatchEnsemble (Wen et al., 2020), MIMO (Havasi et al., 2021), FiLM-Ensemble (Turkoglu et al., 2022).

Among advanced methods, BatchEnsemble seems to be one of the simplest and most flexible options. For example, FiLM-Ensemble (Turkoglu et al., 2022) requires normalization layers to be presented in the original architecture, which is not always the case for tabular MLPs. MIMO (Havasi et al., 2021), in turn, imposes additional limitations compared to BatchEnsemble. *First*, it requires *concatenating* (not *stacking*, as with BatchEnsemble) all $k$ input representations, which increases the input size of the first linear layer. With the relatively high number of submodels $k = 32$ used in our paper, this can be an issue on datasets with a large number of features, especially when feature embeddings (Gorishniy et al., 2022) are used. For example, for $k = 32$, the number of features $m = 1000$, and the feature embedding size $l = 32$, the input size approaches one million resulting in an extremely large first linear layer of MLP. *Second*, with BatchEnsemble, it is easy to explicitly materialize, analyze, and prune individual submodels. By contrast, in MIMO, all submodels are implicitly entangled within one MLP, and there is no easy way to access individual submodels.

**Why MLPs?** Despite the applicability of BatchEnsemble (Wen et al., 2020) to almost any architecture, we focus specifically on MLPs. The key reason is *efficiency*. *First,* to achieve high performance, throughout the paper, we use the relatively large number of submodels $k = 32$. However, the desired less-than-$\times k$ runtime overhead of BatchEnsemble typically happens only when the original model underutilizes the power of parallel computations of a given hardware. This will not be the case for attention-based models on datasets with a large number of features, as well as for retrieval-based models on datasets with a large number of objects. *Second,* as we show in subsection 4.3, attention- and retrieval-based models are already slow as-is. By contrast, MLPs are exceptionally efficient, to the extent that slowing them down even by an order of magnitude will still result in practical models.

Also, generally speaking, the definition of MLP suggested in subsection 3.3 and used in TabM is not special, and more advanced MLP-like backbones can be used. However, in preliminary experiments, we did not observe the benefits of more advanced backbones. Perhaps, small technical differences between backbones become less impactful in the context of parameter-efficient ensembling, at least in the scope of middle-to-large-sized datasets.

## A.2  TABM WITH FEATURE EMBEDDINGS

**Notation.** In this paper, we use † to mark TabM variants with the piecewise-linear embeddings (e.g. $\text{TabM}^{\dagger}_{\text{mini}}$, $\text{TabM}^{\dagger}$, etc.).

**Implementation details.** In fact, there are no changes in the usage of feature embeddings compared to plain MLPs: feature embeddings are applied, and the result is flattened, before being passed to the backbones in terms of Figure 1. For example, if a dataset has $m$ continuous features and all of them are embedded, the very first adapter $R$ will have the shape $k \times md_e$, where $d_e$ is the feature embedding size. For $\text{TabM}^{\dagger}_{\text{mini}}$ and $\text{TabM}^{\dagger}$, we initialize the first multiplicative adapter $R$ of the first linear layer from the standard normal distribution $\mathcal{N}(0, 1)$. The remaining details are best understood from the source code.

**Efficiency.**  When feature embeddings are used, the simplified batching strategy from subsection 3.4 allows for more efficient implementation, when the feature embeddings are applied to the original `batch_size` objects, and the result is simply cloned $k$ times (compared to embedding $k \times$ `batch_size` objects with the original batching strategy).

## A.3  HYPERPARAMETERS

We noticed that the typical optimal learning rate for TabM is higher than for MLP (note that, on each dataset, the batch size is the same for all DL models). We hypothesize that the reason is the

effectively larger batch size for TabM because of how the training batches are constructed (even if the simplified batching strategy from subsection 3.4 is used).

### A.4 LIMITATIONS AND PRACTICAL CONSIDERATIONS

TabM does not introduce any new limitations compared to BatchEnsemble (Wen et al., 2020). Nevertheless, we note the following:

- The MLP backbone used in TabM is one of the simplest possible, and generally, more advanced backbones can be used. That said, some backbones may require additional care when used in TabM. For example, we did not explore backbones with normalization layers. For such layers, it is possible to allocate non-shared trainable affine transformations for each implicit submodel by adding one multiplicative and one additive adapter after the normalization layer (i.e. like in FiLM-Ensemble (Turkoglu et al., 2022)). Additional experiments are required to find the best strategy.
- For ensemble-like models, such as TabM, the notion of "the final object embedding" changes: now, it is not a single vector, but a set of $k$ vectors. If exactly one object embedding is required, then additional experiments may be needed to find the best way to combine $k$ embeddings into one. The presence of multiple object embeddings can also be important for scenarios when TabM is used for solving more than one task, in particular when it is pretrained as a generic feature extractor and then reused for other tasks. The main practical guideline is that the $k$ prediction branches should not interact with each other (e.g. through attention, pooling, etc.) and should always be trained separately.

## B  EXTENDED RESULTS

This section complements section 4.

### B.1  ADDITIONAL BASELINES

In addition to the models from subsection 4.1, we consider the following baselines:

- MLP-PLR Gorishniy et al. (2022), that is, an MLP with periodic embeddings.
- ResNet (Gorishniy et al., 2021)
- SNN (Klambauer et al., 2017)
- DCNv2 (Wang et al., 2020)
- AutoInt (Song et al., 2019)
- MLP-Mixer is our adaptation of Tolstikhin et al. (2021) for tabular data.
- Trompt (Chen et al., 2023b) (our reimplementation, since there is no official implementation)

We also evaluated TabPFN (Hollmann et al., 2023), where possible. The results for this model are available only in Appendix E because this model is by design not applicable to regression tasks, which is a considerable number of our datasets. Overall, TabPFN specializes in small datasets. In line with that, the performance of TabPFN on our benchmark was not competitive.

### B.2  TASK PERFORMANCE

Figure 8 is a different version of Figure 3 with additional baselines. Overall, none of the additional baselines affect our main story.

Figure 9 is the critical difference diagram (CDD) computed over exactly the same results that were used for building Figure 3.

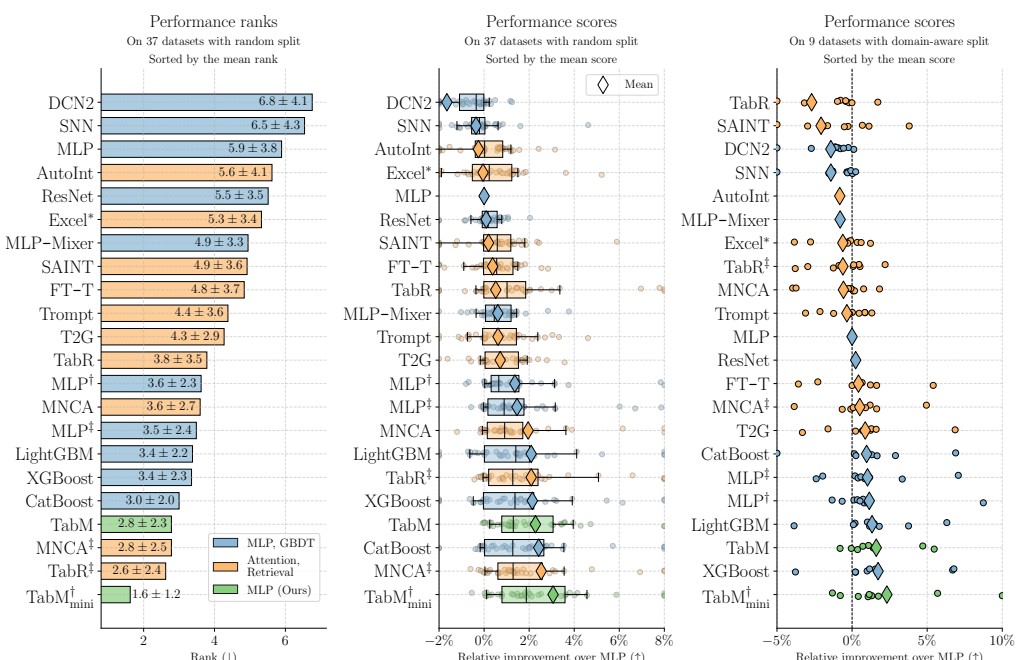

Figure 8: An extended comparison of tabular models as in Figure 3. Note that the ranks (left) are computed only over the 37 datasets with random splits because ResNet, AutoInt, and MLP-Mixer were evaluated only on one 1 out of 9 datasets with domain-aware splits.

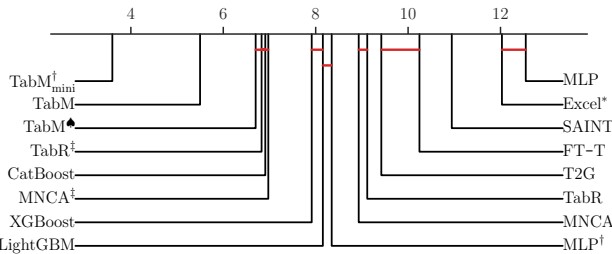

Figure 9: Critical difference diagram. The computation method is taken from the Kim et al. (2024).

## B.3 EFFICIENCY

This section complements subsection 4.3.

**Additional results.**

Figure 10 complements Figure 4 by providing the training times on smaller datasets and the inference throughput on GPU with large batch sizes.

Table 3 provide the number of trainable parameters for some of the models from Figure 3.

**Motivation for the benchmark setup.** Comparing models under all possible kinds of budgets (task performance, the number of parameters, training time, etc.) on all possible hardware (GPU, CPU, etc.) with all possible batch sizes is rather infeasible. As such, we set a narrow goal of *providing a high-level intuition on the efficiency in a transparent setting*. Thus, benchmarking the transparently obtained tuned hyperparameter configurations works well for our goal. Yet, this choice also has a limitation: the hyperparameter tuning process is not aware of the efficiency budget, so it can prefer much heavier configurations even if they lead to tiny performance improvements, which will negatively affect efficiency without a good reason. Overall, we hope that the large number of datasets compensates for potentially imperfect per-dataset measurements.

**Motivation for the two setups for measuring inference throughput.**

- The setup on the right side of Figure 4 simulates the online per-object predictions.
- The setup on the right side of Figure 10 simulates the offline batched computations.

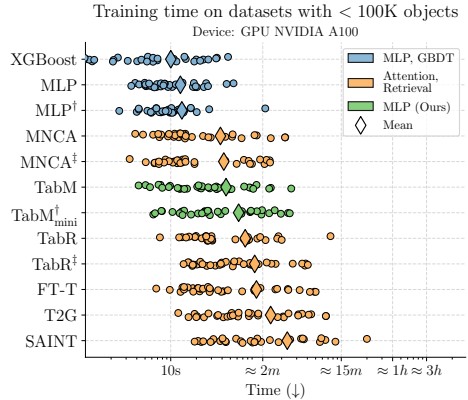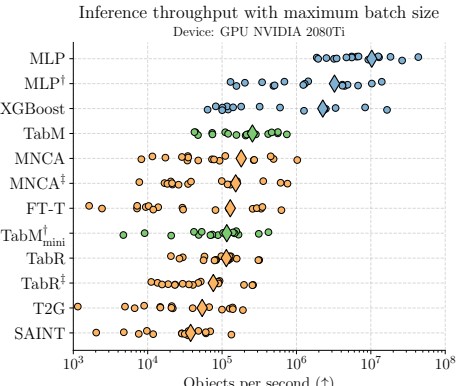

Figure 10: (*Left*) Training time on datasets with less than 100K objects. (*Right*) Inference throughput on GPU with maximum possible batch size (i.e. the batch size depends on a model).

Table 3: Mean number of parameters with std. dev. for 7 different tuned models across all 46 datasets.

| TabM | MLP | FT-T | T2G | TabR | ModernNCA | SAINT |
|------|-----|------|-----|------|-----------|-------|
| $1.4M \pm 1.3M$ | $1.0M \pm 1.0M$ | $1.2M \pm 1.2M$ | $2.1M \pm 1.6M$ | $858K \pm 1.4M$ | $1.0M \pm 1.1M$ | $175.4M \pm 565.4M$ |

## C  DATASETS

In total, we use 46 datasets:

1. 38 datasets are taken from Gorishniy et al. (2024), which includes:
   (a) 28 datasets from Grinsztajn et al. (2022). See the original paper for the precise dataset information.
   (b) 10 datasets from other sources. Their properties are provided in Table 4.
2. 8 datasets from the TabReD benchmark (Rubachev et al., 2024). Their properties are provided in Table 5.

In fact, the aforementioned 38 datasets from Gorishniy et al. (2024) is only a subset of the datasets used in Gorishniy et al. (2024). Namely, we did not include the following of the remaining datasets:

- The datasets that, according to Rubachev et al. (2024), have incorrect splits and/or label leakage, including: Bike_Sharing_Demand, compass, electricity, SGEMM_GPU_kernel_performance, sulfur, visualizing_soil, and the weather forecasting dataset (it is replaced by the correct weather forecasting dataset from TabReD (Rubachev et al., 2024)).
- rl from (Grinsztajn et al., 2022). We observed abnormal results on these datasets. This is an anonymous dataset, which made the investigation impossible, so we removed this dataset to avoid confusion.
- yprop_4_1 from (Grinsztajn et al., 2022). Strictly speaking, this dataset was omitted due to a mistake on our side. For future work, we note that the typical performance gaps on this dataset have low absolute values in terms of RMSE. Perhaps, $R^2$ may be a more appropriate metric for this dataset.

Table 4: Properties of those datasets from Gorishniy et al. (2024) that are not part of Grinsztajn et al. (2022) or TabReD Rubachev et al. (2024). "# Num", "# Bin", and "# Cat" denote the number of numerical, binary, and categorical features, respectively. The table is taken from (Gorishniy et al., 2024).

| Name | # Train | # Validation | # Test | # Num | # Bin | # Cat | Task type | Batch size |
|---|---|---|---|---|---|---|---|---|
| Churn Modelling | 6 400 | 1 600 | 2 000 | 7 | 3 | 1 | Binclass | 128 |
| California Housing | 13 209 | 3 303 | 4 128 | 8 | 0 | 0 | Regression | 256 |
| House 16H | 14 581 | 3 646 | 4 557 | 16 | 0 | 0 | Regression | 256 |
| Adult | 26 048 | 6 513 | 16 281 | 6 | 1 | 8 | Binclass | 256 |
| Diamond | 34 521 | 8 631 | 10 788 | 6 | 0 | 3 | Regression | 512 |
| Otto Group Products | 39 601 | 9 901 | 12 376 | 93 | 0 | 0 | Multiclass | 512 |
| Higgs Small | 62 751 | 15 688 | 19 610 | 28 | 0 | 0 | Binclass | 512 |
| Black Friday | 106 764 | 26 692 | 33 365 | 4 | 1 | 4 | Regression | 512 |
| Covertype | 371 847 | 92 962 | 116 203 | 10 | 4 | 1 | Multiclass | 1024 |
| Microsoft | 723 412 | 235 259 | 241 521 | 131 | 5 | 0 | Regression | 1024 |

Table 5: Properties of the datasets from the TabReD benchmark (Rubachev et al., 2024). "# Num", "# Bin", and "# Cat" denote the number of numerical, binary, and categorical features, respectively.

| Name | # Train | # Validation | # Test | # Num | # Bin | # Cat | Task type | Batch size |
|---|---|---|---|---|---|---|---|---|
| Sberbank Housing | 18 847 | 4 827 | 4 647 | 365 | 17 | 10 | Regression | 256 |
| Ecom Offers | 109 341 | 24 261 | 26 455 | 113 | 6 | 0 | Binclass | 1024 |
| Maps Routing | 160 019 | 59 975 | 59 951 | 984 | 0 | 2 | Regression | 1024 |
| Homesite Insurance | 224 320 | 20 138 | 16 295 | 253 | 23 | 23 | Binclass | 1024 |
| Cooking Time | 227 087 | 51 251 | 41 648 | 186 | 3 | 3 | Regression | 1024 |
| Homecredit Default | 267 645 | 58 018 | 56 001 | 612 | 2 | 82 | Binclass | 1024 |
| Delivery ETA | 279 415 | 34 174 | 36 927 | 221 | 1 | 1 | Regression | 1024 |
| Weather | 340 596 | 42 359 | 40 840 | 100 | 3 | 0 | Regression | 1024 |

## D IMPLEMENTATION DETAILS

### D.1 HARDWARE

Most of the experiments were conducted on a single NVIDIA A100 GPU. In rare exceptions, we used a machine with a single NVIDIA 2080 Ti GPU and Intel(R) Core(TM) i7-7800X CPU @ 3.50GHz.

### D.2 EXPERIMENT SETUP

We mostly follow the experiment setup from Gorishniy et al. (2024). As such, some of the text below is copied from (Gorishniy et al., 2024).

**Data preprocessing.** For each dataset, for all DL-based solutions, the same preprocessing was used for fair comparison. For numerical features, by default, we used a slightly modified version of the quantile normalization from the Scikit-learn package (Pedregosa et al., 2011) (see the source code), with rare exceptions when it turned out to be detrimental (for such datasets, we used the standard normalization or no normalization). For categorical features, we used one-hot encoding. Binary features (i.e. the ones that take only two distinct values) are mapped to $\{0, 1\}$ without any further preprocessing. We completely follow Rubachev et al. (2024) on Table 5 datasets.

**Training neural networks.** For DL-based algorithms, we minimize cross-entropy for classification problems and mean squared error for regression problems. We use the AdamW optimizer (Loshchilov & Hutter, 2019). We do not apply learning rate schedules. We do not use data augmentations. We apply global gradient clipping to $1.0$. For each dataset, we used a predefined dataset-specific batch size. We continue training until there are `patience` consecutive epochs without improvements on the validation set; we set `patience` $= 16$ for the DL models.

**Hyperparameter tuning.** In most cases, hyperparameter tuning is performed with the TPE sampler (typically, 50-100 iterations) from the Optuna package (Akiba et al., 2019). Hyperparameter tuning

spaces for most models are provided in individual sections below (example for TabM: subsection D.9). We follow Rubachev et al. (2024) and use 25 iterations on some datasets from Table 5.

**Evaluation.** On a given dataset, for a given model, the tuned hyperparameters are evaluated under multiple (in most cases, 15) random seeds. The mean test metric and its standard deviation over these random seeds are then used to compare algorithms as described in subsection D.3.

### D.3    METRICS

We use Root Mean Squared Error for regression tasks, ROC-AUC for classification datasets from Table 5 (following Rubachev et al. (2024)), and accuracy for the rest of datasets (following Gorishniy et al. (2024)). We also tried computing ROC-AUC for all classification datasets, but did not observe any significant changes (see Figure 11), so we stuck to prior work. By default, the mean test score and its standard deviation are obtained by training a given model with tuned hyperparameters from scratch on a given dataset under 15 different random seeds.

**How we compute ranks.** Our method of computing ranks used in Figure 3 does not count small improvements as wins, hence the reduced range of ranks compared to other studies. Intuitively, our ranks can be considered as "tiers".

Recall that, on a given dataset, the performance of a given model A is expressed with the mean $A_{mean}$ and the standard deviation $A_{std}$ of the performance score computed after the evaluation under multiple random seeds. Assuming the higher score the better, we define that the model A is better than the model B if: $A_{mean} - A_{std} > B_{mean}$. In other words, a model is considered better if it has a better mean score and the margin is larger than the standard deviation.

On a given dataset, when there are many models, we sort them in descending score order. Starting from the best model (with a rank equal to 1) we iterate over models and assign the rank 1 to all models that are no worse than the best model according to the above rule. The first model in descending order that is worse than the best model is assigned rank 2 and becomes the new reference model. We continue the process until all models are ranked. Ranks are computed independently for each dataset.

### D.4    IMPLEMENTATION DETAILS OF SUBSECTION 4.3

**Applicability to large datasets.** The two datasets used in Table 2 are the *full* versions of the "Weather" and "Maps Routing" datasets from the TabReD benchmark Rubachev et al. (2024). Their smaller versions with subsampled training set were already included in Table 1 and were used when building Figure 3. The validation and test sets are the same for the small and large versions of these datasets, so the task metrics are comparable between the two versions. When running models on the large versions of the datasets, we reused the hyperparameters tuned for their small versions. Thus, this experiment can be seen as a quick assessment of the applicability of several tabular DL to large datasets without a strong focus on the task performance. All models, except for FT-Transformer, were evaluated under 3 random seeds. FT-Transformer was evaluated under 1 random seed.

### D.5    IMPLEMENTATION DETAILS OF SUBSECTION 5.1

**Experiment setup.** This paragraph complements the description of the experiment setup in subsection 5.1. Namely, in addition to what is mentioned in the main text:

- Dropout and weight decay are turned off.
- To get representative training profiles for all models, the learning rates are tuned separately for $\text{TabM}_{mini}^{k=1}$ and $\text{TabM}_{mini}^{k=32}$ on validation sets using the usual metrics (i.e. RMSE or accuracy) as the guidance. The grid for learning rate tuning was: `numpy.logspace(numpy.log10(1e-5), numpy.log10(5e-3), num=25)`.

### D.6    IMPLEMENTATION DETAILS OF SUBSECTION 5.2

**TabM[G].** Here, we clarify the implementation details for TabM[G] described in subsection 5.2. TabM[G] is obtained from a trained TabM by greedily selecting submodels from TabM starting from the best one and stopping when two conditions are simultaneously true for the first time: (1) adding

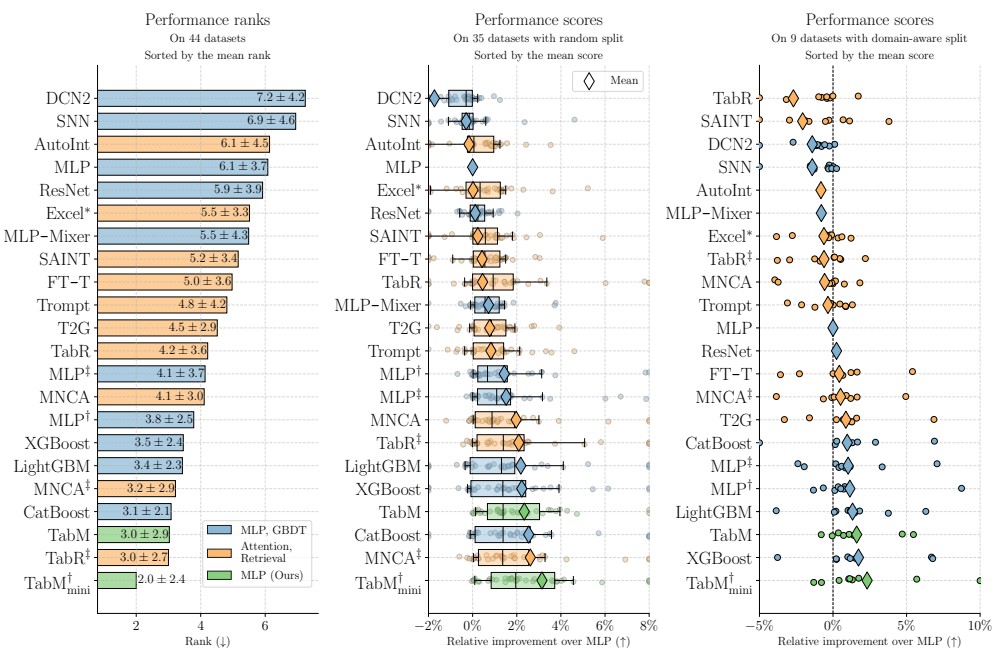

Figure 11: Same as Figure 3, but ROC-AUC is used as the metric for all classification datasets. The two multiclass datasets presented in our benchmark are not taken into account.

any new submodel does not improve the validation metric of the collective prediction; (2) the current validation metric is already better than that of the initial model with all $k$ submodels. To clarify, during the greedy selection, the $i$-th submodel is considered to be better than the $j$-th submodel if adding the $i$-th submodel to the aggregated prediction leads to better validation metrics (i.e. it is *not* the same as adding the submodel in the order of their individual validation metrics).

## D.7 IMPLEMENTATION DETAILS OF SUBSECTION 5.3

Figure 7 shows the mean percentage improvements (see subsection D.3) over MLP across 17 datasets: all datasets except for Covertype from Table 4, and all datasets from TabReD (Rubachev et al., 2024). We have used the dropout rate $0.1$ and tuned the learning rate separately for each value of $k$. The score on each dataset is averaged over 5 seeds.

## D.8 NON-LINEAR EMBEDDINGS FOR CONTINUOUS FEATURES

**Notation.** We use the notation based on † and ‡ only for brevity. Any other unambiguous notation can be used in future work.

**Updated piecewise-linear embeddings.** We use a slightly different implementation of the piecewise-linear embeddings compared to Gorishniy et al. (2022). Architecture-wise, our implementation corresponds to the "Q-L" and "T-L" variations from Table 2 in Gorishniy et al. (2022) (we use the quantile-based bins for simplicity). In practice, our implementation is significantly faster and uses a different parametrization and initialization. See the source code for details.

**Other models.** Since it is not feasible to test all combinations of backbones and embeddings, for baselines, we stick to the embeddings used in the original papers (applies to TabR (Gorishniy et al., 2024), ExcelFormer (Chen et al., 2023a) and ModernNCA (Ye et al., 2024)). For all models with feature embeddings (including TabM, MLP, TabR, ModernNCA, ExcelFormer), the embeddings-related details are commented in the corresponding sections below.

### D.9 TABM

**Feature embeddings.** $TabM_{mini}^\dagger$ and $TabM^\dagger$ are the versions of TabM with non-linear feature embeddings. $TabM_{mini}^\dagger$ and $TabM^\dagger$ use the updated piecewise-linear feature embeddings mentioned in subsection D.8.

Table 6 provides the hyperparameter tuning spaces for TabM and $TabM_{mini}$. Table 7 provides the hyperparameter tuning spaces for $TabM^\dagger$ and $TabM_{mini}^\dagger$.

Table 6: The hyperparameter tuning space for TabM and $TabM_{mini}$. Here, (B) = {Covertype, Microsoft, Table 5} and (A) contains all other datasets.

| Parameter | Distribution or Value |
|---|---|
| $k$ | 32 |
| # layers | $UniformInt[1, 5]$ |
| Width (hidden size) | $UniformInt[64, 1024]$ |
| Dropout rate | $\{0.0, Uniform[0.0, 0.5]\}$ |
| Learning rate | $LogUniform[1e\text{-}4, 5e\text{-}3]$ |
| Weight decay | $\{0, LogUniform[1e\text{-}4, 1e\text{-}1]\}$ |
| # Tuning iterations | (A) 100 (B) 50 |

Table 7: The hyperparameter tuning space for $TabM_{mini}^\dagger$ and $TabM^\dagger$. Here, (B) = {Covertype, Microsoft, Table 5} and (A) contains all other datasets.

| Parameter | Distribution or Value |
|---|---|
| $k$ | 32 |
| # layers | $UniformInt[1, 4]$ |
| Width (hidden size) | $UniformInt[64, 1024]$ |
| Dropout rate | $\{0.0, Uniform[0.0, 0.5]\}$ |
| # PLE bins | $UniformInt[8, 32]$ |
| Learning rate | $LogUniform[5e\text{-}5, 3e\text{-}3]$ |
| Weight decay | $\{0, LogUniform[1e\text{-}4, 1e\text{-}1]\}$ |
| # Tuning iterations | (A) 100 (B) 50 |

### D.10 MLP

**Feature embeddings.** $MLP^\dagger$ and $MLP^\ddagger$ are the versions of MLP with non-linear feature embeddings. $MLP^\dagger$ uses the updated piecewise-linear embeddings mentioned in subsection D.8. $MLP^\ddagger$ (also known as MLP-PLR) uses the periodic embeddings (Gorishniy et al., 2022). Technically, it is the `PeriodicEmbeddings` class from the `rtdl_num_embeddings` Python package. We tested two variations: with `lite=False` and `lite=True`. In the paper, only the former one is reported, but in the source code, the results for both are available.

Table 8, Table 9, Table 10 provide the hyperparameter tuning spaces for MLP, $MLP^\dagger$ and $MLP^\ddagger$, respectively.

### D.11 TABR

**Feature embeddings.** $TabR^\ddagger$ is the version of TabR with non-linear feature embeddings. $TabR^\ddagger$ uses the periodic embeddings (Gorishniy et al., 2022), specifically, `PeriodicEmbeddings(lite=True)` from the `rtdl_num_embeddings` Python package on most datasets. On the datasets from Table 5, $TabR^\ddagger$ uses the

Table 8: The hyperparameter tuning space for MLP.

| Parameter | Distribution |
|---|---|
| # layers | $\mathrm{UniformInt}[1, 6]$ |
| Width (hidden size) | $\mathrm{UniformInt}[64, 1024]$ |
| Dropout rate | $\{0.0, \mathrm{Uniform}[0.0, 0.5]\}$ |
| Learning rate | $\mathrm{LogUniform}[3e\text{-}5, 1e\text{-}3]$ |
| Weight decay | $\{0, \mathrm{LogUniform}[1e\text{-}4, 1e\text{-}1]\}$ |
| # Tuning iterations | 100 |

Table 9: The hyperparameter tuning space for MLP[†].

| Parameter | Distribution |
|---|---|
| # layers | $\mathrm{UniformInt}[1, 5]$ |
| Width (hidden size) | $\mathrm{UniformInt}[64, 1024]$ |
| Dropout rate | $\{0.0, \mathrm{Uniform}[0.0, 0.5]\}$ |
| Learning rate | $\mathrm{LogUniform}[3e\text{-}5, 1e\text{-}3]$ |
| Weight decay | $\{0, \mathrm{LogUniform}[1e\text{-}4, 1e\text{-}1]\}$ |
| d_embedding | $\mathrm{UniformInt}[8, 32]$ |
| n_bins | $\mathrm{UniformInt}[2, 128]$ |
| # Tuning iterations | 100 |

Table 10: The hyperparameter tuning space for MLP[‡].

| Parameter | Distribution |
|---|---|
| # layers | $\mathrm{UniformInt}[1, 5]$ |
| Width (hidden size) | $\mathrm{UniformInt}[64, 1024]$ |
| Dropout rate | $\{0.0, \mathrm{Uniform}[0.0, 0.5]\}$ |
| Learning rate | $\mathrm{LogUniform}[3e\text{-}5, 1e\text{-}3]$ |
| Weight decay | $\{0, \mathrm{LogUniform}[1e\text{-}4, 1e\text{-}1]\}$ |
| n_frequencies | $\mathrm{UniformInt}[16, 96]$ |
| d_embedding | $\mathrm{UniformInt}[16, 32]$ |
| frequency_init_scale | $\mathrm{LogUniform}[1e\text{-}2, 1e1]$ |
| # Tuning iterations | 100 |

`PeriodicEmbeddings(lite=True)` embeddings on the Sberbank Housing and Ecom Offers datasets, and `LinearReLUEmbeddings` on the rest (to fit the computations into the GPU memory, following the original TabR paper).

Since we follow the training and evaluation protocols from Gorishniy et al. (2024), and TabR was proposed in Gorishniy et al. (2024), we simply reuse the results for TabR. More details can be found in Appendix.D from Gorishniy et al. (2024). When tuning TabR[‡] on the datasets from Table 5, we have used 25 tuning iterations and the same tuning space as for TabR from Rubachev et al. (2024).

## D.12    FT-Transformer

We used the implementation from the "`rtdl_revisiting_models`" Python package. The results on datasets from Table 5 were copied from Rubachev et al. (2024), because the experiment setups are compatible.

Table 11: The hyperparameter tuning space for FT-Transformer Gorishniy et al. (2021). Here, (B) = {Covertype, Microsoft} and (A) contains all other datasets (except Table 5).

| Parameter | Distribution or Value |
|---|---|
| # blocks | $\text{UniformInt}[1, 4]$ |
| $d_{token}$ | $\text{UniformInt}[16, 384]$ |
| Attention dropout rate | $\text{Uniform}[0.0, 0.5]$ |
| FFN hidden dimension expansion rate | $\text{Uniform}[2/3, 8/3]$ |
| FFN dropout rate | $\text{Uniform}[0.0, 0.5]$ |
| Residual dropout rate | $\{0.0, \text{Uniform}[0.0, 0.2]\}$ |
| Learning rate | $\text{LogUniform}[3e\text{-}5, 1e\text{-}3]$ |
| Weight decay | $\{0, \text{LogUniform}[1e\text{-}4, 1e\text{-}1]\}$ |
| # Tuning iterations | (A) 100 (B) 50 |

## D.13    ModernNCA

**Feature embeddings.** We adapted the official implementation of Ye et al. (2024). We used periodic embeddings Gorishniy et al. (2022) (specifically, `PeriodicEmbeddings(lite=True)` from the `rtdl_num_embeddings` Python package) for ModernNCA[‡] and no embeddings for Modern-NCA. Table 12 and Table 13 provides hyperparameter tuning spaces for each ModernNCA and ModernNCA[‡].

Table 12: The hyperparameter tuning space for ModernNCA. Here, (C) = {Table 5}, (B) = {Covertype, Microsoft} and (A) contains all other datasets.

| Parameter | Distribution |
|---|---|
| # blocks | $\text{UniformInt}[0, 2]$ |
| $d_{block}$ | $\text{UniformInt}[64, 1024]$ |
| dim | $\text{UniformInt}[64, 1024]$ |
| Dropout rate | $\text{Uniform}[0.0, 0.5]$ |
| Sample rate | $\text{Uniform}[0.05, 0.6]$ |
| Learning rate | $\text{LogUniform}[1e\text{-}5, 1e\text{-}1]$ |
| Weight decay | $\{0, \text{LogUniform}[1e\text{-}6, 1e\text{-}3]\}$ |
| # Tuning iterations | (A) 100 (B, C) 50 |

## D.14    T2G-Former

We adapted the implementation and hyperparameters of Yan et al. (2023) from the official repository[1]. Table 14 provides hyperparameter tuning space.

## D.15    SAINT

We completely adapted hyperparameters and protocol from Gorishniy et al. (2024) to evaluate SAINT on Grinsztajn et al. (2022) benchmark. Results on datasets from Table 4 were directly taken from

---

[1]https://github.com/jyansir/t2g-former

Table 13: The hyperparameter tuning space for ModernNCA[‡]. Here, (C) = {Table 5}, (B) = {Covertype, Microsoft} and (A) contains all other datasets.

| Parameter | Distribution |
|---|---|
| # blocks | UniformInt$[0, 2]$ |
| $d_{block}$ | UniformInt$[64, 1024]$ |
| dim | UniformInt$[64, 1024]$ |
| Dropout rate | Uniform$[0.0, 0.5]$ |
| Sample rate | Uniform$[0.05, 0.6]$ |
| Learning rate | LogUniform$[1e\text{-}5, 1e\text{-}1]$ |
| Weight decay | $\{0, \text{LogUniform}[1e\text{-}6, 1e\text{-}3]\}$ |
| n_frequencies | UniformInt$[16, 96]$ |
| d_embedding | UniformInt$[16, 32]$ |
| frequency_init_scale | LogUniform$[0.01, 10]$ |
| # Tuning iterations | (A) 100 (B, C) 50 |

Table 14: The hyperparameter tuning space for T2G-Former Yan et al. (2023). Here, (C) = {Table 5}, (B) = {Covertype, Microsoft} and (A) contains all other datasets. Also, we used $50$ tuning iterations on some datasets from Grinsztajn et al. (2022).

| Parameter | Distribution or Value |
|---|---|
| # blocks | (A) UniformInt$[3, 4]$ (B, C) UniformInt$[1, 3]$ |
| $d_{token}$ | UniformInt$[64, 512]$ |
| Attention dropout rate | Uniform$[0.0, 0.5]$ |
| FFN hidden dimension expansion rate | (A, B) Uniform$[2/3, 8/3]$ (C) $4/3$ |
| FFN dropout rate | Uniform$[0.0, 0.5]$ |
| Residual dropout rate | $\{0.0, \text{Uniform}[0.0, 0.2]\}$ |
| Learning rate | LogUniform$[3e\text{-}5, 1e\text{-}3]$ |
| Col. Learning rate | LogUniform$[5e\text{-}3, 5e\text{-}2]$ |
| Weight decay | $\{0, \text{LogUniform}[1e\text{-}6, 1e\text{-}1]\}$ |
| # Tuning iterations | (A) 100 (B) 50 (C) 25 |

Gorishniy et al. (2024). Additional details can be found in Appendix.D from Gorishniy et al. (2024). We have used a default configuration on big datasets due to the very high cost of tuning (see Table 15).

### D.16 EXCELFORMER

**Feature embeddings.** ExcelFormer (Chen et al., 2023a) uses custom non-linear feature embeddings based on a GLU-style activation, see the original paper for details.

We adapted the implementation and hyperparameters of Chen et al. (2023a) from the official repository[2]. For a fair comparison with other models, we did not use the augmentation techniques from the paper in our experiments. See Table 16.

### D.17 CATBOOST, XGBOOST AND LIGHTGBM

Since our setup is directly taken from Gorishniy et al. (2024), we simply reused their results for GBDTs from the official repository[3]. Importantly, in a series of preliminary experiments, we

---

[2]https://github.com/WhatAShot/ExcelFormer
[3]https://github.com/yandex-research/tabular-dl-tabr

Table 15: The default hyperparameters for SAINT (Somepalli et al., 2021) on datasets from Rubachev et al. (2024).

| Parameter | Value |
|---|---|
| depth | 2 |
| $d_{token}$ | 32 |
| $n_{heads}$ | 4 |
| $d_{head}$ | 8 |
| Attention dropout rate | 0.1 |
| FFN hidden dimension expansion rate | 1 |
| FFN dropout rate | 0.8 |
| Learning rate | 1e-4 |
| Weight decay | 1e-2 |

Table 16: The hyperparameter tuning space for Excelformer Chen et al. (2023a). Here, (D) = {Homecredit, Maps Routing}, (C) = {Table 5 w/o (D)}, (B) = {Covertype, Microsoft} and (A) contains all other datasets.

| Parameter | Distribution or Value |
|---|---|
| # blocks | (A, B) UniformInt$[2, 5]$ (C) UniformInt$[2, 4]$ (D) UniformInt$[1, 3]$ |
| $d_{token}$ | (A, B) $\{32, 64, 128, 256\}$ (C) $\{16, 32, 64\}$ (D) $\{4, 8, 16, 32\}$ |
| $n_{heads}$ | (A,B) $\{4, 8, 16, 32\}$ (C) $\{4, 8, 16\}$ (D) 4 |
| Attention dropout rate | 0.3 |
| FFN dropout rate | 0.0 |
| Residual dropout rate | Uniform$[0.0, 0.5]$ |
| Learning rate | LogUniform$[3e\text{-}5, 1e\text{-}3]$ |
| Weight decay | $\{0, \text{LogUniform}[1e\text{-}4, 1e\text{-}1]\}$ |
| # Tuning iterations | (A) 100 (B) 50 (C, D) 25 |

confirmed that those results are reproducible in our instance of their setup. The details can be found in Appendix.D from Gorishniy et al. (2024). Results on datasets from Table 5 were copied from the paper (Rubachev et al., 2024).

### D.18  AUTOINT

We used an implementation from Gorishniy et al. (2021) which is an adapted official implementation[4].

### D.18.1  TABPFN

Since TabPFN accepts only less than 10K training samples we use different subsamples of size 10K for different random seeds. Also, TabPFN is not applicable to regressions and datasets with more than 100 features.

---

[4]https://github.com/shichence/AutoInt

Table 17: The hyperparameter tuning space for AutoInt (Song et al., 2019). Here, (B) = {Covertype, Microsoft} and (A) contains all other datasets.

| Parameter | Distribution |
|---|---|
| # blocks | $\text{UniformInt}[1, 6]$ |
| $d_{token}$ | $\text{UniformInt}[8, 64]$ |
| $n_{heads}$ | 2 |
| Attention dropout rate | $\{0, \text{Uniform}[0.0, 0.5]\}$ |
| Embedding dropout rate | $\{0, \text{Uniform}[0.0, 0.5]\}$ |
| Learning rate | $\text{LogUniform}[3e\text{-}5, 1e\text{-}3]$ |
| Weight decay | $\{0, \text{LogUniform}[1e\text{-}4, 1e\text{-}1]\}$ |
| # Tuning iterations | (A) 100 (B) 50 |

# E    PER-DATASET RESULTS WITH STANDARD DEVIATIONS

Table 18: Extended results for the main benchmark. Results are grouped by datasets. One ensemble consists of five models trained independently under different random seeds.

| | churn ↑ | | | california ↓ | |
|---|---|---|---|---|---|
| Method | Single model | Ensemble | Method | Single model | Ensemble |
| MLP | $0.8553 \pm 0.0029$ | $0.8582 \pm 0.0008$ | MLP | $0.4948 \pm 0.0058$ | $0.4880 \pm 0.0022$ |
| TabPFN | – | $0.8624 \pm 0.0008$ | TabPFN | – | – |
| ResNet | $0.8545 \pm 0.0044$ | $0.8565 \pm 0.0035$ | ResNet | $0.4915 \pm 0.0031$ | $0.4862 \pm 0.0017$ |
| DCN2 | $0.8567 \pm 0.0020$ | $0.8570 \pm 0.0017$ | DCN2 | $0.4971 \pm 0.0122$ | $0.4779 \pm 0.0022$ |
| SNN | $0.8506 \pm 0.0051$ | $0.8533 \pm 0.0033$ | SNN | $0.5033 \pm 0.0075$ | $0.4933 \pm 0.0035$ |
| Trompt | $0.8600 \pm nan$ | – | Trompt | $0.4579 \pm nan$ | – |
| AutoInt | $0.8607 \pm 0.0047$ | $0.8622 \pm 0.0003$ | AutoInt | $0.4682 \pm 0.0063$ | $0.4490 \pm 0.0028$ |
| MLP−Mixer | $0.8592 \pm 0.0036$ | $0.8630 \pm 0.0005$ | MLP−Mixer | $0.4746 \pm 0.0056$ | $0.4509 \pm 0.0029$ |
| Excel* | $0.8618 \pm 0.0023$ | $0.8625 \pm nan$ | Excel* | $0.4544 \pm 0.0048$ | $0.4350 \pm nan$ |
| SAINT | $0.8603 \pm 0.0029$ | – | SAINT | $0.4680 \pm 0.0048$ | – |
| FT−T | $0.8593 \pm 0.0028$ | $0.8598 \pm 0.0025$ | FT−T | $0.4635 \pm 0.0048$ | $0.4515 \pm 0.0016$ |
| T2G | $0.8613 \pm 0.0015$ | – | T2G | $0.4640 \pm 0.0100$ | $0.4462 \pm nan$ |
| MLP$^{\ddagger-\text{lite}}$ | $0.8624 \pm 0.0010$ | $0.8638 \pm 0.0012$ | MLP$^{\ddagger-\text{lite}}$ | $0.4652 \pm 0.0045$ | $0.4549 \pm 0.0006$ |
| MLP$^{\ddagger}$ | $0.8624 \pm 0.0026$ | $0.8640 \pm 0.0010$ | MLP$^{\ddagger}$ | $0.4597 \pm 0.0058$ | $0.4482 \pm 0.0026$ |
| MLP$^{\dagger}$ | $0.8580 \pm 0.0028$ | $0.8605 \pm 0.0018$ | MLP$^{\dagger}$ | $0.4530 \pm 0.0029$ | $0.4491 \pm 0.0010$ |
| XGBoost | $0.8605 \pm 0.0022$ | $0.8608 \pm 0.0013$ | XGBoost | $0.4327 \pm 0.0016$ | $0.4316 \pm 0.0007$ |
| LightGBM | $0.8600 \pm 0.0008$ | $0.8600 \pm 0.0000$ | LightGBM | $0.4352 \pm 0.0019$ | $0.4339 \pm 0.0008$ |
| CatBoost | $0.8582 \pm 0.0017$ | $0.8588 \pm 0.0008$ | CatBoost | $0.4294 \pm 0.0012$ | $0.4265 \pm 0.0003$ |
| TabR | $0.8599 \pm 0.0025$ | $0.8620 \pm 0.0023$ | TabR | $0.4030 \pm 0.0023$ | $0.3964 \pm 0.0013$ |
| TabR$^{\ddagger}$ | $0.8625 \pm 0.0021$ | – | TabR$^{\ddagger}$ | $0.3998 \pm 0.0033$ | – |
| MNCA | $0.8595 \pm 0.0028$ | $0.8615 \pm 0.0013$ | MNCA | $0.4239 \pm 0.0012$ | $0.4231 \pm 0.0005$ |
| MNCA$^{\ddagger}$ | $0.8606 \pm 0.0032$ | $0.8607 \pm 0.0008$ | MNCA$^{\ddagger}$ | $0.4142 \pm 0.0031$ | $0.4071 \pm 0.0029$ |
| TabM$^{\spadesuit}$ | $0.8613 \pm 0.0025$ | $0.8615 \pm 0.0005$ | TabM$^{\spadesuit}$ | $0.4509 \pm 0.0032$ | $0.4490 \pm 0.0018$ |
| TabM | $0.8605 \pm 0.0016$ | $0.8612 \pm 0.0008$ | TabM | $0.4414 \pm 0.0012$ | $0.4402 \pm 0.0001$ |
| TabM[G] | $0.8609 \pm 0.0024$ | – | TabM[G] | $0.4413 \pm 0.0020$ | – |
| TabM$_{\text{mini}}$ | $0.8633 \pm 0.0018$ | $0.8638 \pm 0.0012$ | TabM$_{\text{mini}}$ | $0.4479 \pm 0.0022$ | $0.4461 \pm 0.0011$ |
| TabM$^{\dagger}_{\text{mini}}$ | $0.8606 \pm 0.0023$ | $0.8630 \pm 0.0030$ | TabM$^{\dagger}_{\text{mini}}$ | $0.4275 \pm 0.0024$ | $0.4244 \pm 0.0006$ |

| house ↓ | | |
| --- | --- | --- |
| Method | Single model | Ensemble |
| MLP | $3.1117 \pm 0.0294$ | $3.0706 \pm 0.0140$ |
| TabPFN | – | – |
| ResNet | $3.1143 \pm 0.0258$ | $3.0706 \pm 0.0098$ |
| DCN2 | $3.3327 \pm 0.0878$ | $3.1303 \pm 0.0410$ |
| SNN | $3.2176 \pm 0.0376$ | $3.1320 \pm 0.0155$ |
| Trompt | $3.0638 \pm nan$ | – |
| AutoInt | $3.2157 \pm 0.0436$ | $3.1261 \pm 0.0095$ |
| MLP−Mixer | $3.1871 \pm 0.0519$ | $3.0184 \pm 0.0086$ |
| Excel* | $3.2460 \pm 0.0685$ | $3.1097 \pm nan$ |
| SAINT | $3.2424 \pm 0.0595$ | – |
| FT−T | $3.1823 \pm 0.0460$ | $3.0974 \pm 0.0334$ |
| T2G | $3.1613 \pm 0.0320$ | $3.0982 \pm nan$ |
| MLP$^{\ddagger-\text{lite}}$ | $3.0633 \pm 0.0248$ | $3.0170 \pm 0.0070$ |
| MLP$^{\ddagger}$ | $3.0775 \pm 0.0336$ | $3.0268 \pm 0.0170$ |
| MLP$^{\dagger}$ | $3.0999 \pm 0.0351$ | $3.0401 \pm 0.0071$ |
| XGBoost | $3.1773 \pm 0.0102$ | $3.1644 \pm 0.0068$ |
| LightGBM | $3.1774 \pm 0.0087$ | $3.1672 \pm 0.0050$ |
| CatBoost | $3.1172 \pm 0.0125$ | $3.1058 \pm 0.0022$ |
| TabR | $3.0667 \pm 0.0403$ | $2.9958 \pm 0.0270$ |
| TabR$^{\ddagger}$ | $3.1048 \pm 0.0410$ | – |
| MNCA | $3.0884 \pm 0.0286$ | $3.0538 \pm 0.0072$ |
| MNCA$^{\ddagger}$ | $3.0704 \pm 0.0388$ | $3.0149 \pm 0.0308$ |
| TabM$^{\spadesuit}$ | $3.0002 \pm 0.0182$ | $2.9796 \pm 0.0024$ |
| TabM | $3.0038 \pm 0.0097$ | $2.9906 \pm 0.0026$ |
| TabM[G] | $3.0082 \pm 0.0184$ | – |
| TabM$_{\text{mini}}$ | $3.0394 \pm 0.0139$ | $3.0206 \pm 0.0128$ |
| TabM$^{\dagger}_{\text{mini}}$ | $2.9976 \pm 0.0196$ | $2.9854 \pm 0.0076$ |

| adult ↑ | | |
| --- | --- | --- |
| Method | Single model | Ensemble |
| MLP | $0.8540 \pm 0.0018$ | $0.8559 \pm 0.0011$ |
| TabPFN | – | – |
| ResNet | $0.8554 \pm 0.0011$ | $0.8562 \pm 0.0006$ |
| DCN2 | $0.8582 \pm 0.0011$ | $0.8593 \pm 0.0002$ |
| SNN | $0.8582 \pm 0.0009$ | $0.8603 \pm 0.0012$ |
| Trompt | $0.8590 \pm nan$ | – |
| AutoInt | $0.8592 \pm 0.0016$ | $0.8612 \pm 0.0004$ |
| MLP−Mixer | $0.8598 \pm 0.0013$ | $0.8617 \pm 0.0002$ |
| Excel* | $0.8613 \pm 0.0024$ | $0.8641 \pm nan$ |
| SAINT | $0.8601 \pm 0.0019$ | – |
| FT−T | $0.8588 \pm 0.0015$ | $0.8608 \pm 0.0011$ |
| T2G | $0.8601 \pm 0.0011$ | $0.8622 \pm nan$ |
| MLP$^{\ddagger-\text{lite}}$ | $0.8693 \pm 0.0007$ | $0.8702 \pm 0.0006$ |
| MLP$^{\ddagger}$ | $0.8694 \pm 0.0011$ | $0.8704 \pm 0.0008$ |
| MLP$^{\dagger}$ | $0.8603 \pm 0.0009$ | $0.8616 \pm 0.0006$ |
| XGBoost | $0.8720 \pm 0.0006$ | $0.8723 \pm 0.0002$ |
| LightGBM | $0.8713 \pm 0.0007$ | $0.8721 \pm 0.0004$ |
| CatBoost | $0.8714 \pm 0.0012$ | $0.8723 \pm 0.0007$ |
| TabR | $0.8646 \pm 0.0022$ | $0.8680 \pm 0.0019$ |
| TabR$^{\ddagger}$ | $0.8699 \pm 0.0011$ | – |
| MNCA | $0.8677 \pm 0.0018$ | $0.8696 \pm 0.0003$ |
| MNCA$^{\ddagger}$ | $0.8717 \pm 0.0008$ | $0.8742 \pm 0.0006$ |
| TabM$^{\spadesuit}$ | $0.8582 \pm 0.0011$ | $0.8588 \pm 0.0003$ |
| TabM | $0.8575 \pm 0.0008$ | $0.8583 \pm 0.0004$ |
| TabM[G] | $0.8572 \pm 0.0010$ | – |
| TabM$_{\text{mini}}$ | $0.8598 \pm 0.0011$ | $0.8604 \pm 0.0000$ |
| TabM$^{\dagger}_{\text{mini}}$ | $0.8700 \pm 0.0007$ | $0.8701 \pm 0.0003$ |

| diamond ↓ | | |
| --- | --- | --- |
| Method | Single model | Ensemble |
| MLP | $0.1404 \pm 0.0012$ | $0.1362 \pm 0.0003$ |
| TabPFN | – | – |
| ResNet | $0.1396 \pm 0.0029$ | $0.1361 \pm 0.0011$ |
| DCN2 | $0.1420 \pm 0.0032$ | $0.1374 \pm 0.0020$ |
| SNN | $0.1473 \pm 0.0057$ | $0.1424 \pm 0.0008$ |
| Trompt | $0.1391 \pm nan$ | – |
| AutoInt | $0.1392 \pm 0.0014$ | $0.1361 \pm 0.0004$ |
| MLP−Mixer | $0.1400 \pm 0.0025$ | $0.1378 \pm 0.0008$ |
| Excel* | $0.1766 \pm 0.0023$ | $0.1712 \pm nan$ |
| SAINT | $0.1369 \pm 0.0019$ | – |
| FT−T | $0.1376 \pm 0.0013$ | $0.1360 \pm 0.0002$ |
| T2G | $0.1372 \pm 0.0011$ | $0.1346 \pm nan$ |
| MLP$^{\ddagger-\text{lite}}$ | $0.1342 \pm 0.0008$ | $0.1325 \pm 0.0004$ |
| MLP$^{\ddagger}$ | $0.1337 \pm 0.0010$ | $0.1317 \pm 0.0003$ |
| MLP$^{\dagger}$ | $0.1323 \pm 0.0010$ | $0.1301 \pm 0.0005$ |
| XGBoost | $0.1368 \pm 0.0004$ | $0.1363 \pm 0.0001$ |
| LightGBM | $0.1359 \pm 0.0002$ | $0.1358 \pm 0.0001$ |
| CatBoost | $0.1335 \pm 0.0006$ | $0.1327 \pm 0.0004$ |
| TabR | $0.1327 \pm 0.0010$ | $0.1311 \pm 0.0005$ |
| TabR$^{\ddagger}$ | $0.1333 \pm 0.0013$ | – |
| MNCA | $0.1370 \pm 0.0018$ | $0.1348 \pm 0.0005$ |
| MNCA$^{\ddagger}$ | $0.1327 \pm 0.0012$ | $0.1315 \pm 0.0006$ |
| TabM$^{\spadesuit}$ | $0.1342 \pm 0.0017$ | $0.1327 \pm 0.0004$ |
| TabM | $0.1310 \pm 0.0007$ | $0.1307 \pm 0.0002$ |
| TabM[G] | $0.1309 \pm 0.0008$ | – |
| TabM$_{\text{mini}}$ | $0.1323 \pm 0.0007$ | $0.1317 \pm 0.0002$ |
| TabM$^{\dagger}_{\text{mini}}$ | $0.1315 \pm 0.0006$ | $0.1312 \pm 0.0001$ |

| otto ↑ | | |
| --- | --- | --- |
| Method | Single model | Ensemble |
| MLP | $0.8175 \pm 0.0022$ | $0.8222 \pm 0.0007$ |
| TabPFN | – | $0.7408 \pm 0.0028$ |
| ResNet | $0.8174 \pm 0.0021$ | $0.8198 \pm 0.0006$ |
| DCN2 | $0.8064 \pm 0.0021$ | $0.8208 \pm 0.0023$ |
| SNN | $0.8087 \pm 0.0020$ | $0.8156 \pm 0.0013$ |
| Trompt | $0.8093 \pm nan$ | – |
| AutoInt | $0.8050 \pm 0.0034$ | $0.8111 \pm 0.0020$ |
| MLP−Mixer | $0.8092 \pm 0.0040$ | $0.8136 \pm 0.0010$ |
| Excel* | $0.8102 \pm 0.0022$ | $0.8220 \pm nan$ |
| SAINT | $0.8119 \pm 0.0018$ | – |
| FT−T | $0.8133 \pm 0.0033$ | $0.8221 \pm 0.0013$ |
| T2G | $0.8161 \pm 0.0019$ | $0.8272 \pm nan$ |
| MLP$^{\ddagger-\text{lite}}$ | $0.8190 \pm 0.0021$ | $0.8271 \pm 0.0015$ |
| MLP$^{\ddagger}$ | $0.8189 \pm 0.0015$ | $0.8253 \pm 0.0000$ |
| MLP$^{\dagger}$ | $0.8205 \pm 0.0021$ | $0.8290 \pm 0.0006$ |
| XGBoost | $0.8297 \pm 0.0011$ | $0.8316 \pm 0.0008$ |
| LightGBM | $0.8302 \pm 0.0009$ | $0.8316 \pm 0.0013$ |
| CatBoost | $0.8250 \pm 0.0013$ | $0.8268 \pm 0.0002$ |
| TabR | $0.8179 \pm 0.0022$ | $0.8236 \pm 0.0009$ |
| TabR$^{\ddagger}$ | $0.8246 \pm 0.0018$ | – |
| MNCA | $0.8275 \pm 0.0012$ | $0.8313 \pm 0.0006$ |
| MNCA$^{\ddagger}$ | $0.8265 \pm 0.0015$ | $0.8304 \pm 0.0006$ |
| TabM$^{\spadesuit}$ | $0.8268 \pm 0.0014$ | $0.8300 \pm 0.0007$ |
| TabM | $0.8275 \pm 0.0014$ | $0.8284 \pm 0.0005$ |
| TabM[G] | $0.8254 \pm 0.0022$ | – |
| TabM$_{\text{mini}}$ | $0.8282 \pm 0.0014$ | $0.8299 \pm 0.0005$ |
| TabM$^{\dagger}_{\text{mini}}$ | $0.8342 \pm 0.0012$ | $0.8356 \pm 0.0004$ |

| higgs-small ↑ | | |
|---|---|---|
| Method | Single model | Ensemble |
| MLP | $0.7180 \pm 0.0027$ | $0.7192 \pm 0.0005$ |
| TabPFN | – | $0.6727 \pm 0.0034$ |
| ResNet | $0.7256 \pm 0.0020$ | $0.7307 \pm 0.0001$ |
| DCN2 | $0.7164 \pm 0.0030$ | $0.7237 \pm 0.0011$ |
| SNN | $0.7142 \pm 0.0024$ | $0.7171 \pm 0.0020$ |
| Trompt | $0.7262 \pm nan$ | – |
| AutoInt | $0.7240 \pm 0.0028$ | $0.7287 \pm 0.0008$ |
| MLP−Mixer | $0.7248 \pm 0.0023$ | $0.7334 \pm 0.0007$ |
| Excel* | $0.7262 \pm 0.0017$ | $0.7329 \pm nan$ |
| SAINT | $0.7236 \pm 0.0019$ | – |
| FT−T | $0.7281 \pm 0.0016$ | $0.7334 \pm 0.0013$ |
| T2G | $0.7352 \pm 0.0037$ | $0.7400 \pm nan$ |
| $MLP^{\ddagger-lite}$ | $0.7260 \pm 0.0017$ | $0.7304 \pm 0.0008$ |
| $MLP^{\ddagger}$ | $0.7261 \pm 0.0010$ | $0.7270 \pm 0.0003$ |
| $MLP^{\dagger}$ | $0.7210 \pm 0.0016$ | $0.7252 \pm 0.0005$ |
| XGBoost | $0.7246 \pm 0.0015$ | $0.7264 \pm 0.0013$ |
| LightGBM | $0.7256 \pm 0.0009$ | $0.7263 \pm 0.0007$ |
| CatBoost | $0.7260 \pm 0.0011$ | $0.7273 \pm 0.0010$ |
| TabR | $0.7223 \pm 0.0010$ | $0.7257 \pm 0.0008$ |
| $TabR^{\ddagger}$ | $0.7294 \pm 0.0014$ | – |
| MNCA | $0.7263 \pm 0.0023$ | $0.7292 \pm 0.0006$ |
| $MNCA^{\ddagger}$ | $0.7300 \pm 0.0020$ | $0.7348 \pm 0.0008$ |
| TabM♠ | $0.7383 \pm 0.0028$ | $0.7409 \pm 0.0010$ |
| TabM | $0.7394 \pm 0.0018$ | $0.7409 \pm 0.0008$ |
| TabM[G] | $0.7392 \pm 0.0016$ | – |
| $TabM_{mini}$ | $0.7338 \pm 0.0011$ | $0.7345 \pm 0.0008$ |
| $TabM^{\dagger}_{mini}$ | $0.7361 \pm 0.0011$ | $0.7383 \pm 0.0008$ |

| black-friday ↓ | | |
|---|---|---|
| Method | Single model | Ensemble |
| MLP | $0.6955 \pm 0.0004$ | $0.6942 \pm 0.0002$ |
| TabPFN | – | – |
| ResNet | $0.6929 \pm 0.0008$ | $0.6907 \pm 0.0002$ |
| DCN2 | $0.6968 \pm 0.0013$ | $0.6936 \pm 0.0007$ |
| SNN | $0.6996 \pm 0.0013$ | $0.6978 \pm 0.0004$ |
| Trompt | $0.6983 \pm nan$ | – |
| AutoInt | $0.6994 \pm 0.0082$ | $0.6927 \pm 0.0021$ |
| MLP−Mixer | $0.6905 \pm 0.0021$ | $0.6851 \pm 0.0011$ |
| Excel* | $0.6947 \pm 0.0016$ | $0.6908 \pm nan$ |
| SAINT | $0.6934 \pm 0.0009$ | – |
| FT−T | $0.6987 \pm 0.0192$ | $0.6879 \pm 0.0023$ |
| T2G | $0.6887 \pm 0.0046$ | $0.6832 \pm nan$ |
| $MLP^{\ddagger-lite}$ | $0.6849 \pm 0.0006$ | $0.6824 \pm 0.0002$ |
| $MLP^{\ddagger}$ | $0.6857 \pm 0.0004$ | $0.6838 \pm 0.0002$ |
| $MLP^{\dagger}$ | $0.6836 \pm 0.0006$ | $0.6812 \pm 0.0002$ |
| XGBoost | $0.6806 \pm 0.0001$ | $0.6805 \pm 0.0000$ |
| LightGBM | $0.6799 \pm 0.0003$ | $0.6795 \pm 0.0001$ |
| CatBoost | $0.6822 \pm 0.0003$ | $0.6813 \pm 0.0002$ |
| TabR | $0.6899 \pm 0.0004$ | $0.6883 \pm 0.0002$ |
| $TabR^{\ddagger}$ | $0.6761 \pm 0.0009$ | – |
| MNCA | $0.6893 \pm 0.0004$ | $0.6883 \pm 0.0000$ |
| $MNCA^{\ddagger}$ | $0.6885 \pm 0.0007$ | $0.6863 \pm 0.0003$ |
| TabM♠ | $0.6875 \pm 0.0015$ | $0.6866 \pm 0.0003$ |
| TabM | $0.6869 \pm 0.0004$ | $0.6865 \pm 0.0001$ |
| TabM[G] | $0.6865 \pm 0.0005$ | – |
| $TabM_{mini}$ | $0.6863 \pm 0.0006$ | $0.6856 \pm 0.0003$ |
| $TabM^{\dagger}_{mini}$ | $0.6781 \pm 0.0004$ | $0.6773 \pm 0.0001$ |

| covtype2 ↑ | | |
|---|---|---|
| Method | Single model | Ensemble |
| MLP | $0.9630 \pm 0.0012$ | $0.9664 \pm 0.0004$ |
| TabPFN | – | $0.7606 \pm 0.0022$ |
| ResNet | $0.9638 \pm 0.0005$ | $0.9685 \pm 0.0003$ |
| DCN2 | $0.9622 \pm 0.0019$ | $0.9673 \pm 0.0011$ |
| SNN | $0.9636 \pm 0.0010$ | $0.9677 \pm 0.0002$ |
| Trompt | $0.9286 \pm nan$ | – |
| AutoInt | $0.9614 \pm 0.0016$ | $0.9696 \pm 0.0005$ |
| MLP−Mixer | $0.9663 \pm 0.0019$ | $0.9699 \pm 0.0014$ |
| Excel* | $0.9606 \pm 0.0018$ | $0.9670 \pm nan$ |
| SAINT | $0.9669 \pm 0.0010$ | – |
| FT−T | $0.9698 \pm 0.0008$ | $0.9731 \pm 0.0006$ |
| T2G | $0.9668 \pm 0.0008$ | $0.9708 \pm nan$ |
| $MLP^{\ddagger-lite}$ | $0.9690 \pm 0.0008$ | $0.9721 \pm 0.0006$ |
| $MLP^{\ddagger}$ | $0.9713 \pm 0.0006$ | $0.9758 \pm 0.0000$ |
| $MLP^{\dagger}$ | $0.9697 \pm 0.0008$ | $0.9721 \pm 0.0005$ |
| XGBoost | $0.9710 \pm 0.0002$ | $0.9713 \pm 0.0000$ |
| LightGBM | $0.9709 \pm 0.0003$ | – |
| CatBoost | $0.9670 \pm 0.0003$ | $0.9680 \pm 0.0002$ |
| TabR | $0.9737 \pm 0.0005$ | $0.9745 \pm 0.0006$ |
| $TabR^{\ddagger}$ | $0.9752 \pm 0.0003$ | – |
| MNCA | $0.9724 \pm 0.0003$ | $0.9729 \pm 0.0001$ |
| $MNCA^{\ddagger}$ | $0.9747 \pm 0.0002$ | $0.9747 \pm 0.0002$ |
| TabM♠ | $0.9712 \pm 0.0008$ | $0.9729 \pm 0.0003$ |
| TabM | $0.9735 \pm 0.0004$ | $0.9743 \pm 0.0001$ |
| TabM[G] | $0.9730 \pm 0.0005$ | – |
| $TabM_{mini}$ | $0.9710 \pm 0.0007$ | $0.9727 \pm 0.0002$ |
| $TabM^{\dagger}_{mini}$ | $0.9755 \pm 0.0003$ | $0.9762 \pm 0.0001$ |

| microsoft ↓ | | |
|---|---|---|
| Method | Single model | Ensemble |
| MLP | $0.7475 \pm 0.0003$ | $0.7460 \pm 0.0003$ |
| TabPFN | – | – |
| ResNet | $0.7472 \pm 0.0004$ | $0.7452 \pm 0.0004$ |
| DCN2 | $0.7499 \pm 0.0003$ | $0.7477 \pm 0.0001$ |
| SNN | $0.7488 \pm 0.0004$ | $0.7470 \pm 0.0001$ |
| Trompt | $0.7476 \pm nan$ | – |
| AutoInt | $0.7482 \pm 0.0005$ | $0.7455 \pm 0.0002$ |
| MLP−Mixer | $0.7482 \pm 0.0008$ | $0.7436 \pm 0.0001$ |
| Excel* | $0.7479 \pm 0.0007$ | $0.7442 \pm nan$ |
| SAINT | $0.7625 \pm 0.0066$ | – |
| FT−T | $0.7460 \pm 0.0007$ | $0.7422 \pm 0.0004$ |
| T2G | $0.7460 \pm 0.0006$ | $0.7427 \pm nan$ |
| $MLP^{\ddagger-lite}$ | $0.7446 \pm 0.0002$ | $0.7434 \pm 0.0002$ |
| $MLP^{\ddagger}$ | $0.7444 \pm 0.0003$ | $0.7429 \pm 0.0001$ |
| $MLP^{\dagger}$ | $0.7465 \pm 0.0005$ | $0.7448 \pm 0.0001$ |
| XGBoost | $0.7413 \pm 0.0001$ | $0.7410 \pm 0.0000$ |
| LightGBM | $0.7417 \pm 0.0001$ | $0.7413 \pm 0.0000$ |
| CatBoost | $0.7412 \pm 0.0001$ | $0.7406 \pm 0.0000$ |
| TabR | $0.7503 \pm 0.0006$ | $0.7485 \pm 0.0002$ |
| $TabR^{\ddagger}$ | $0.7501 \pm 0.0005$ | – |
| MNCA | $0.7458 \pm 0.0003$ | $0.7448 \pm 0.0002$ |
| $MNCA^{\ddagger}$ | $0.7460 \pm 0.0008$ | $0.7435 \pm 0.0004$ |
| TabM♠ | $0.7434 \pm 0.0003$ | $0.7424 \pm 0.0001$ |
| TabM | $0.7432 \pm 0.0004$ | $0.7426 \pm 0.0001$ |
| TabM[G] | $0.7432 \pm 0.0004$ | – |
| $TabM_{mini}$ | $0.7436 \pm 0.0002$ | $0.7430 \pm 0.0002$ |
| $TabM^{\dagger}_{mini}$ | $0.7423 \pm 0.0002$ | $0.7416 \pm 0.0001$ |

Table 19: Extended results for Grinsztajn et al. (2022) benchmark. Results are grouped by datasets. One ensemble consists of five models trained independently with different random seeds.

| wine ↑ | | |
|---|---|---|
| Method | Single model | Ensemble |
| MLP | $0.7778 \pm 0.0153$ | $0.7907 \pm 0.0117$ |
| TabPFN | – | $0.7908 \pm 0.0063$ |
| ResNet | $0.7710 \pm 0.0137$ | $0.7839 \pm 0.0083$ |
| DCN2 | $0.7492 \pm 0.0147$ | $0.7764 \pm 0.0095$ |
| SNN | $0.7818 \pm 0.0143$ | $0.7994 \pm 0.0097$ |
| Trompt | $0.7818 \pm 0.0081$ | – |
| AutoInt | $0.7745 \pm 0.0144$ | $0.7909 \pm 0.0160$ |
| MLP−Mixer | $0.7769 \pm 0.0149$ | $0.7950 \pm 0.0087$ |
| Excel* | $0.7631 \pm 0.0171$ | $0.7765 \pm 0.0121$ |
| SAINT | $0.7684 \pm 0.0144$ | – |
| FT−T | $0.7755 \pm 0.0133$ | $0.7894 \pm 0.0083$ |
| T2G | $0.7733 \pm 0.0118$ | $0.7933 \pm 0.0137$ |
| MLP$^{\ddagger-\text{lite}}$ | $0.7803 \pm 0.0157$ | $0.7964 \pm 0.0146$ |
| MLP$^{\ddagger}$ | $0.7733 \pm 0.0185$ | $0.7856 \pm 0.0160$ |
| MLP$^{\dagger}$ | $0.7814 \pm 0.0132$ | $0.7919 \pm 0.0098$ |
| XGBoost | $0.7949 \pm 0.0178$ | $0.8010 \pm 0.0186$ |
| LightGBM | $0.7890 \pm 0.0160$ | $0.7929 \pm 0.0106$ |
| CatBoost | $0.7994 \pm 0.0131$ | $0.8057 \pm 0.0098$ |
| TabR | $0.7936 \pm 0.0114$ | $0.8055 \pm 0.0057$ |
| TabR$^{\ddagger}$ | $0.7804 \pm 0.0148$ | – |
| MNCA | $0.7911 \pm 0.0135$ | $0.8005 \pm 0.0121$ |
| MNCA$^{\ddagger}$ | $0.7867 \pm 0.0113$ | $0.7953 \pm 0.0114$ |
| TabM$^{\spadesuit}$ | $0.7961 \pm 0.0136$ | $0.8011 \pm 0.0084$ |
| TabM | $0.7943 \pm 0.0124$ | $0.7985 \pm 0.0139$ |
| TabM[G] | $0.7879 \pm 0.0161$ | – |
| TabM$_{\text{mini}}$ | $0.7890 \pm 0.0130$ | $0.7937 \pm 0.0103$ |
| TabM$^{\dagger}_{\text{mini}}$ | $0.7839 \pm 0.0169$ | $0.7917 \pm 0.0143$ |

| phoneme ↑ | | |
|---|---|---|
| Method | Single model | Ensemble |
| MLP | $0.8525 \pm 0.0126$ | $0.8635 \pm 0.0099$ |
| TabPFN | – | $0.8684 \pm 0.0050$ |
| ResNet | $0.8456 \pm 0.0121$ | $0.8504 \pm 0.0066$ |
| DCN2 | $0.8342 \pm 0.0151$ | $0.8543 \pm 0.0118$ |
| SNN | $0.8596 \pm 0.0124$ | $0.8687 \pm 0.0080$ |
| Trompt | $0.8465 \pm 0.0205$ | – |
| AutoInt | $0.8623 \pm 0.0138$ | $0.8754 \pm 0.0095$ |
| MLP−Mixer | $0.8629 \pm 0.0123$ | $0.8757 \pm 0.0095$ |
| Excel* | $0.8551 \pm 0.0092$ | $0.8711 \pm 0.0081$ |
| SAINT | $0.8657 \pm 0.0130$ | – |
| FT−T | $0.8667 \pm 0.0127$ | $0.8795 \pm 0.0093$ |
| T2G | $0.8672 \pm 0.0166$ | $0.8765 \pm 0.0141$ |
| MLP$^{\ddagger-\text{lite}}$ | $0.8742 \pm 0.0120$ | $0.8861 \pm 0.0071$ |
| MLP$^{\ddagger}$ | $0.8757 \pm 0.0118$ | $0.8856 \pm 0.0065$ |
| MLP$^{\dagger}$ | $0.8647 \pm 0.0098$ | $0.8761 \pm 0.0076$ |
| XGBoost | $0.8682 \pm 0.0174$ | $0.8771 \pm 0.0156$ |
| LightGBM | $0.8702 \pm 0.0129$ | $0.8733 \pm 0.0126$ |
| CatBoost | $0.8827 \pm 0.0117$ | $0.8897 \pm 0.0055$ |
| TabR | $0.8781 \pm 0.0096$ | $0.8840 \pm 0.0054$ |
| TabR$^{\ddagger}$ | $0.8772 \pm 0.0087$ | – |
| MNCA | $0.8835 \pm 0.0079$ | $0.8861 \pm 0.0057$ |
| MNCA$^{\ddagger}$ | $0.8828 \pm 0.0082$ | $0.8925 \pm 0.0056$ |
| TabM$^{\spadesuit}$ | $0.8701 \pm 0.0167$ | $0.8766 \pm 0.0128$ |
| TabM | $0.8831 \pm 0.0121$ | $0.8880 \pm 0.0108$ |
| TabM[G] | $0.8762 \pm 0.0144$ | – |
| TabM$_{\text{mini}}$ | $0.8803 \pm 0.0098$ | $0.8842 \pm 0.0067$ |
| TabM$^{\dagger}_{\text{mini}}$ | $0.8780 \pm 0.0119$ | $0.8817 \pm 0.0101$ |

| analcatdata_supreme ↓ | | |
|---|---|---|
| Method | Single model | Ensemble |
| MLP | $0.0782 \pm 0.0081$ | $0.0766 \pm 0.0090$ |
| TabPFN | – | – |
| ResNet | $0.0852 \pm 0.0076$ | $0.0823 \pm 0.0078$ |
| DCN2 | $0.0811 \pm 0.0137$ | $0.0759 \pm 0.0086$ |
| SNN | $0.0826 \pm 0.0096$ | $0.0779 \pm 0.0098$ |
| Trompt | $0.0782 \pm 0.0095$ | – |
| AutoInt | $0.0783 \pm 0.0078$ | $0.0768 \pm 0.0083$ |
| MLP−Mixer | $0.0770 \pm 0.0082$ | $0.0759 \pm 0.0081$ |
| Excel* | $0.0796 \pm 0.0101$ | $0.0776 \pm 0.0101$ |
| SAINT | $0.0773 \pm 0.0078$ | – |
| FT−T | $0.0787 \pm 0.0086$ | $0.0775 \pm 0.0091$ |
| T2G | $0.0775 \pm 0.0081$ | $0.0763 \pm 0.0084$ |
| MLP$^{\ddagger-\text{lite}}$ | $0.0798 \pm 0.0088$ | $0.0769 \pm 0.0092$ |
| MLP$^{\ddagger}$ | $0.0786 \pm 0.0073$ | $0.0720 \pm 0.0053$ |
| MLP$^{\dagger}$ | $0.0774 \pm 0.0064$ | $0.0759 \pm 0.0063$ |
| XGBoost | $0.0801 \pm 0.0126$ | $0.0774 \pm 0.0107$ |
| LightGBM | $0.0778 \pm 0.0115$ | $0.0767 \pm 0.0110$ |
| CatBoost | $0.0780 \pm 0.0067$ | $0.0734 \pm 0.0022$ |
| TabR | $0.0803 \pm 0.0066$ | $0.0759 \pm 0.0046$ |
| TabR$^{\ddagger}$ | $0.0807 \pm 0.0088$ | – |
| MNCA | $0.0809 \pm 0.0072$ | $0.0784 \pm 0.0062$ |
| MNCA$^{\ddagger}$ | $0.0825 \pm 0.0090$ | $0.0793 \pm 0.0072$ |
| TabM$^{\spadesuit}$ | $0.0777 \pm 0.0099$ | $0.0769 \pm 0.0105$ |
| TabM | $0.0786 \pm 0.0055$ | $0.0781 \pm 0.0054$ |
| TabM[G] | $0.0808 \pm 0.0063$ | – |
| TabM$_{\text{mini}}$ | $0.0773 \pm 0.0077$ | $0.0763 \pm 0.0077$ |
| TabM$^{\dagger}_{\text{mini}}$ | $0.0764 \pm 0.0071$ | $0.0749 \pm 0.0076$ |

| Mercedes_Benz_Greener_Manufacturing ↓ | | |
|---|---|---|
| Method | Single model | Ensemble |
| MLP | $8.3045 \pm 0.8708$ | $8.2682 \pm 0.8992$ |
| TabPFN | – | – |
| ResNet | $8.4434 \pm 0.7982$ | $8.3178 \pm 0.8482$ |
| DCN2 | $8.3540 \pm 0.8314$ | $8.3021 \pm 0.8579$ |
| SNN | $8.2718 \pm 0.8152$ | $8.2236 \pm 0.8479$ |
| Trompt | $8.3409 \pm 0.9840$ | – |
| AutoInt | $8.4001 \pm 0.9256$ | $8.3237 \pm 0.9658$ |
| MLP−Mixer | $8.2860 \pm 0.8656$ | $8.2398 \pm 0.9023$ |
| Excel* | $8.2244 \pm 0.8514$ | $8.1918 \pm 0.9387$ |
| SAINT | $8.3556 \pm 0.9566$ | – |
| FT−T | $8.2252 \pm 0.8617$ | $8.1616 \pm 0.8834$ |
| T2G | $8.2120 \pm 0.8485$ | $8.1654 \pm 0.9339$ |
| MLP$^{\ddagger-\text{lite}}$ | $8.3045 \pm 0.8708$ | $8.2682 \pm 0.8992$ |
| MLP$^{\ddagger}$ | $8.3045 \pm 0.8708$ | $8.2682 \pm 0.8992$ |
| MLP$^{\dagger}$ | $8.3045 \pm 0.8708$ | $8.2682 \pm 0.8992$ |
| XGBoost | $8.2177 \pm 0.8175$ | $8.2092 \pm 0.8458$ |
| LightGBM | $8.2078 \pm 0.8231$ | $8.1618 \pm 0.8566$ |
| CatBoost | $8.1629 \pm 0.8193$ | $8.1554 \pm 0.8439$ |
| TabR | $8.3506 \pm 0.8149$ | $8.2694 \pm 0.8399$ |
| TabR$^{\ddagger}$ | $8.3187 \pm 0.8186$ | – |
| MNCA | $8.2557 \pm 0.8602$ | $8.1771 \pm 0.8710$ |
| MNCA$^{\ddagger}$ | $8.2557 \pm 0.8602$ | $8.1771 \pm 0.8710$ |
| TabM$^{\spadesuit}$ | $8.2215 \pm 0.8940$ | $8.1995 \pm 0.9130$ |
| TabM | $8.2052 \pm 0.9043$ | $8.1965 \pm 0.9306$ |
| TabM[G] | $8.2235 \pm 0.8867$ | – |
| TabM$_{\text{mini}}$ | $8.2075 \pm 0.9185$ | $8.1986 \pm 0.9442$ |
| TabM$^{\dagger}_{\text{mini}}$ | $8.2075 \pm 0.9185$ | $8.1986 \pm 0.9442$ |

KDDCup09_upselling ↑

| Method | Single model | Ensemble |
|---|---|---|
| MLP | $0.7759 \pm 0.0137$ | $0.7806 \pm 0.0125$ |
| TabPFN | – | – |
| ResNet | $0.7811 \pm 0.0124$ | $0.7861 \pm 0.0109$ |
| DCN2 | $0.7850 \pm 0.0161$ | $0.7884 \pm 0.0135$ |
| SNN | $0.7884 \pm 0.0122$ | $0.7940 \pm 0.0116$ |
| Trompt | $0.7994 \pm 0.0055$ | – |
| AutoInt | $0.8004 \pm 0.0075$ | $0.8037 \pm 0.0063$ |
| MLP−Mixer | $0.7979 \pm 0.0105$ | $0.8010 \pm 0.0094$ |
| Excel* | $0.7903 \pm 0.0074$ | $0.7939 \pm 0.0099$ |
| SAINT | $0.7942 \pm 0.0112$ | – |
| FT−T | $0.7957 \pm 0.0127$ | $0.7960 \pm 0.0139$ |
| T2G | $0.8037 \pm 0.0100$ | $0.7988 \pm 0.0084$ |
| MLP$^{\ddagger-\text{lite}}$ | $0.7962 \pm 0.0093$ | $0.7995 \pm 0.0105$ |
| MLP$^{\ddagger}$ | $0.8005 \pm 0.0097$ | $0.8032 \pm 0.0117$ |
| MLP$^{\dagger}$ | $0.7925 \pm 0.0123$ | $0.7963 \pm 0.0089$ |
| XGBoost | $0.7930 \pm 0.0108$ | $0.7950 \pm 0.0102$ |
| LightGBM | $0.7932 \pm 0.0119$ | $0.7969 \pm 0.0115$ |
| CatBoost | $0.7992 \pm 0.0117$ | $0.8010 \pm 0.0121$ |
| TabR | $0.7838 \pm 0.0136$ | $0.7859 \pm 0.0167$ |
| TabR$^{\ddagger}$ | $0.7908 \pm 0.0123$ | – |
| MNCA | $0.7939 \pm 0.0097$ | $0.7989 \pm 0.0115$ |
| MNCA$^{\ddagger}$ | $0.7960 \pm 0.0131$ | $0.8008 \pm 0.0110$ |
| TabM$^{\spadesuit}$ | $0.8002 \pm 0.0103$ | $0.8021 \pm 0.0074$ |
| TabM | $0.8024 \pm 0.0111$ | $0.8054 \pm 0.0123$ |
| TabM[G] | $0.7988 \pm 0.0118$ | – |
| TabM$_{\text{mini}}$ | $0.7971 \pm 0.0117$ | $0.7982 \pm 0.0107$ |
| TabM$_{\text{mini}}^{\dagger}$ | $0.8024 \pm 0.0075$ | $0.8035 \pm 0.0088$ |

kdd_ipums_la_97-small ↑

| Method | Single model | Ensemble |
|---|---|---|
| MLP | $0.8828 \pm 0.0061$ | $0.8845 \pm 0.0055$ |
| TabPFN | – | $0.8578 \pm 0.0046$ |
| ResNet | $0.8823 \pm 0.0070$ | $0.8824 \pm 0.0060$ |
| DCN2 | $0.8770 \pm 0.0072$ | $0.8824 \pm 0.0068$ |
| SNN | $0.8722 \pm 0.0093$ | $0.8733 \pm 0.0083$ |
| Trompt | $0.8847 \pm 0.0070$ | – |
| AutoInt | $0.8808 \pm 0.0083$ | $0.8830 \pm 0.0081$ |
| MLP−Mixer | $0.8762 \pm 0.0100$ | $0.8770 \pm 0.0088$ |
| Excel* | $0.8803 \pm 0.0054$ | $0.8823 \pm 0.0071$ |
| SAINT | $0.8837 \pm 0.0055$ | – |
| FT−T | $0.8795 \pm 0.0077$ | $0.8792 \pm 0.0062$ |
| T2G | $0.8833 \pm 0.0054$ | $0.8841 \pm 0.0062$ |
| MLP$^{\ddagger-\text{lite}}$ | $0.8765 \pm 0.0108$ | $0.8765 \pm 0.0108$ |
| MLP$^{\ddagger}$ | $0.8816 \pm 0.0057$ | $0.8818 \pm 0.0048$ |
| MLP$^{\dagger}$ | $0.8757 \pm 0.0101$ | $0.8756 \pm 0.0104$ |
| XGBoost | $0.8825 \pm 0.0089$ | $0.8835 \pm 0.0085$ |
| LightGBM | $0.8792 \pm 0.0075$ | $0.8802 \pm 0.0067$ |
| CatBoost | $0.8793 \pm 0.0088$ | $0.8803 \pm 0.0100$ |
| TabR | $0.8798 \pm 0.0081$ | $0.8819 \pm 0.0078$ |
| TabR$^{\ddagger}$ | $0.8831 \pm 0.0050$ | – |
| MNCA | $0.8819 \pm 0.0054$ | $0.8832 \pm 0.0048$ |
| MNCA$^{\ddagger}$ | $0.8837 \pm 0.0062$ | $0.8860 \pm 0.0059$ |
| TabM$^{\spadesuit}$ | $0.8845 \pm 0.0063$ | $0.8848 \pm 0.0070$ |
| TabM | $0.8823 \pm 0.0079$ | $0.8825 \pm 0.0071$ |
| TabM[G] | $0.8818 \pm 0.0082$ | – |
| TabM$_{\text{mini}}$ | $0.8784 \pm 0.0123$ | $0.8786 \pm 0.0133$ |
| TabM$_{\text{mini}}^{\dagger}$ | $0.8779 \pm 0.0094$ | $0.8784 \pm 0.0108$ |

wine_quality ↓

| Method | Single model | Ensemble |
|---|---|---|
| MLP | $0.6707 \pm 0.0178$ | $0.6530 \pm 0.0152$ |
| TabPFN | – | – |
| ResNet | $0.6687 \pm 0.0166$ | $0.6543 \pm 0.0170$ |
| DCN2 | $0.7010 \pm 0.0171$ | $0.6699 \pm 0.0139$ |
| SNN | $0.6604 \pm 0.0174$ | $0.6245 \pm 0.0140$ |
| Trompt | $0.6605 \pm 0.0153$ | – |
| AutoInt | $0.6840 \pm 0.0126$ | $0.6478 \pm 0.0146$ |
| MLP−Mixer | $0.6672 \pm 0.0263$ | $0.6294 \pm 0.0200$ |
| Excel* | $0.6881 \pm 0.0182$ | $0.6664 \pm 0.0179$ |
| SAINT | $0.6797 \pm 0.0161$ | – |
| FT−T | $0.6787 \pm 0.0149$ | $0.6564 \pm 0.0250$ |
| T2G | $0.6783 \pm 0.0170$ | $0.6570 \pm 0.0273$ |
| MLP$^{\ddagger-\text{lite}}$ | $0.6569 \pm 0.0167$ | $0.6328 \pm 0.0155$ |
| MLP$^{\ddagger}$ | $0.6532 \pm 0.0133$ | $0.6336 \pm 0.0140$ |
| MLP$^{\dagger}$ | $0.6721 \pm 0.0180$ | $0.6463 \pm 0.0262$ |
| XGBoost | $0.6039 \pm 0.0134$ | $0.6025 \pm 0.0139$ |
| LightGBM | $0.6135 \pm 0.0138$ | $0.6122 \pm 0.0144$ |
| CatBoost | $0.6088 \pm 0.0132$ | $0.6060 \pm 0.0137$ |
| TabR | $0.6315 \pm 0.0097$ | $0.6197 \pm 0.0096$ |
| TabR$^{\ddagger}$ | $0.6412 \pm 0.0105$ | – |
| MNCA | $0.6154 \pm 0.0083$ | $0.6058 \pm 0.0149$ |
| MNCA$^{\ddagger}$ | $0.6099 \pm 0.0144$ | $0.6028 \pm 0.0157$ |
| TabM$^{\spadesuit}$ | $0.6169 \pm 0.0123$ | $0.6131 \pm 0.0126$ |
| TabM | $0.6328 \pm 0.0172$ | $0.6297 \pm 0.0180$ |
| TabM[G] | $0.6369 \pm 0.0179$ | – |
| TabM$_{\text{mini}}$ | $0.6314 \pm 0.0142$ | $0.6272 \pm 0.0146$ |
| TabM$_{\text{mini}}^{\dagger}$ | $0.6294 \pm 0.0120$ | $0.6241 \pm 0.0118$ |

isolet ↓

| Method | Single model | Ensemble |
|---|---|---|
| MLP | $2.2744 \pm 0.2203$ | $2.0018 \pm 0.1111$ |
| TabPFN | – | – |
| ResNet | $2.2077 \pm 0.2248$ | $1.9206 \pm 0.1478$ |
| DCN2 | $2.2449 \pm 0.1579$ | $2.0176 \pm 0.0770$ |
| SNN | $2.4269 \pm 0.2382$ | $2.1142 \pm 0.1262$ |
| Trompt | $2.6219 \pm 0.0315$ | – |
| AutoInt | $2.6130 \pm 0.1658$ | $2.3308 \pm 0.1088$ |
| MLP−Mixer | $2.3344 \pm 0.2073$ | $2.0915 \pm 0.1159$ |
| Excel* | $2.8691 \pm 0.0882$ | $2.5989 \pm 0.0664$ |
| SAINT | $2.7696 \pm 0.0200$ | – |
| FT−T | $2.4879 \pm 0.2524$ | $2.1501 \pm 0.1506$ |
| T2G | $2.2867 \pm 0.2489$ | $1.9179 \pm 0.1530$ |
| MLP$^{\ddagger-\text{lite}}$ | $2.2719 \pm 0.1006$ | $2.1026 \pm 0.1088$ |
| MLP$^{\ddagger}$ | $2.1832 \pm 0.1124$ | $2.0775 \pm 0.0805$ |
| MLP$^{\dagger}$ | $2.0979 \pm 0.1779$ | $1.9283 \pm 0.1334$ |
| XGBoost | $2.7567 \pm 0.0470$ | $2.7294 \pm 0.0366$ |
| LightGBM | $2.7005 \pm 0.0296$ | $2.6903 \pm 0.0290$ |
| CatBoost | $2.8847 \pm 0.0227$ | $2.8574 \pm 0.0148$ |
| TabR | $1.9760 \pm 0.1738$ | $1.7627 \pm 0.1520$ |
| TabR$^{\ddagger}$ | $1.9919 \pm 0.1813$ | – |
| MNCA | $1.7905 \pm 0.1594$ | $1.6205 \pm 0.1676$ |
| MNCA$^{\ddagger}$ | $1.8912 \pm 0.1851$ | $1.7147 \pm 0.1348$ |
| TabM$^{\spadesuit}$ | $1.8831 \pm 0.1194$ | $1.8578 \pm 0.1088$ |
| TabM | $1.8433 \pm 0.1196$ | $1.8230 \pm 0.1197$ |
| TabM[G] | $1.9091 \pm 0.1345$ | – |
| TabM$_{\text{mini}}$ | $1.9421 \pm 0.0971$ | $1.9013 \pm 0.0813$ |
| TabM$_{\text{mini}}^{\dagger}$ | $1.7799 \pm 0.0859$ | $1.7560 \pm 0.0795$ |

| cpu_act ↓ | | |
|---|---|---|
| Method | Single model | Ensemble |
| MLP | $2.6814 \pm 0.2291$ | $2.4953 \pm 0.1150$ |
| TabPFN | – | – |
| ResNet | $2.3933 \pm 0.0641$ | $2.3005 \pm 0.0397$ |
| DCN2 | $2.7868 \pm 0.1999$ | $2.4884 \pm 0.0327$ |
| SNN | $2.5811 \pm 0.1480$ | $2.3863 \pm 0.0324$ |
| Trompt | $2.2133 \pm 0.0221$ | – |
| AutoInt | $2.2537 \pm 0.0536$ | $2.1708 \pm 0.0349$ |
| MLP−Mixer | $2.3079 \pm 0.0829$ | $2.1831 \pm 0.0470$ |
| Excel* | $2.3094 \pm 0.2401$ | $2.1411 \pm 0.0767$ |
| SAINT | $2.2781 \pm 0.0630$ | – |
| FT−T | $2.2394 \pm 0.0508$ | $2.1494 \pm 0.0268$ |
| T2G | $2.2111 \pm 0.0413$ | $2.1330 \pm 0.0316$ |
| MLP$^{\ddagger-\text{lite}}$ | $2.2730 \pm 0.0457$ | $2.1899 \pm 0.0419$ |
| MLP$^{\ddagger}$ | $2.2671 \pm 0.0383$ | $2.1940 \pm 0.0433$ |
| MLP$^{\dagger}$ | $2.3309 \pm 0.0719$ | $2.2516 \pm 0.0574$ |
| XGBoost | $2.5237 \pm 0.3530$ | $2.4723 \pm 0.3789$ |
| LightGBM | $2.2223 \pm 0.0894$ | $2.2067 \pm 0.0916$ |
| CatBoost | $2.1239 \pm 0.0489$ | $2.1092 \pm 0.0499$ |
| TabR | $2.2980 \pm 0.0529$ | $2.2228 \pm 0.0501$ |
| TabR$^{\ddagger}$ | $2.1278 \pm 0.0783$ | – |
| MNCA | $2.2603 \pm 0.0479$ | $2.2339 \pm 0.0508$ |
| MNCA$^{\ddagger}$ | $2.2105 \pm 0.0483$ | $2.1396 \pm 0.0474$ |
| TabM$^{\spadesuit}$ | $2.1940 \pm 0.0523$ | $2.1677 \pm 0.0487$ |
| TabM | $2.1402 \pm 0.0588$ | $2.1265 \pm 0.0580$ |
| TabM[G] | $2.1549 \pm 0.0626$ | – |
| TabM$_{\text{mini}}$ | $2.1638 \pm 0.0420$ | $2.1508 \pm 0.0416$ |
| TabM$^{\dagger}_{\text{mini}}$ | $2.1391 \pm 0.0542$ | $2.1221 \pm 0.0570$ |

| bank-marketing ↑ | | |
|---|---|---|
| Method | Single model | Ensemble |
| MLP | $0.7860 \pm 0.0057$ | $0.7887 \pm 0.0052$ |
| TabPFN | – | $0.7894 \pm 0.0091$ |
| ResNet | $0.7921 \pm 0.0076$ | $0.7932 \pm 0.0066$ |
| DCN2 | $0.7859 \pm 0.0068$ | $0.7917 \pm 0.0078$ |
| SNN | $0.7836 \pm 0.0074$ | $0.7882 \pm 0.0054$ |
| Trompt | $0.7975 \pm 0.0080$ | – |
| AutoInt | $0.7917 \pm 0.0071$ | $0.7956 \pm 0.0058$ |
| MLP−Mixer | $0.7954 \pm 0.0059$ | $0.8001 \pm 0.0048$ |
| Excel* | $0.7957 \pm 0.0090$ | $0.7985 \pm 0.0106$ |
| SAINT | $0.7953 \pm 0.0058$ | – |
| FT−T | $0.7918 \pm 0.0076$ | $0.7951 \pm 0.0071$ |
| T2G | $0.7918 \pm 0.0058$ | $0.7955 \pm 0.0047$ |
| MLP$^{\ddagger-\text{lite}}$ | $0.7947 \pm 0.0101$ | $0.7977 \pm 0.0117$ |
| MLP$^{\ddagger}$ | $0.7988 \pm 0.0092$ | $0.8024 \pm 0.0093$ |
| MLP$^{\dagger}$ | $0.7981 \pm 0.0065$ | $0.8008 \pm 0.0057$ |
| XGBoost | $0.8013 \pm 0.0081$ | $0.8030 \pm 0.0076$ |
| LightGBM | $0.8006 \pm 0.0078$ | $0.8013 \pm 0.0072$ |
| CatBoost | $0.8026 \pm 0.0068$ | $0.8056 \pm 0.0082$ |
| TabR | $0.7995 \pm 0.0054$ | $0.8015 \pm 0.0037$ |
| TabR$^{\ddagger}$ | $0.8023 \pm 0.0088$ | – |
| MNCA | $0.7961 \pm 0.0065$ | $0.8003 \pm 0.0077$ |
| MNCA$^{\ddagger}$ | $0.7977 \pm 0.0081$ | $0.8010 \pm 0.0084$ |
| TabM$^{\spadesuit}$ | $0.7908 \pm 0.0068$ | $0.7915 \pm 0.0068$ |
| TabM | $0.7944 \pm 0.0060$ | $0.7944 \pm 0.0052$ |
| TabM[G] | $0.7935 \pm 0.0064$ | – |
| TabM$_{\text{mini}}$ | $0.7941 \pm 0.0055$ | $0.7943 \pm 0.0045$ |
| TabM$^{\dagger}_{\text{mini}}$ | $0.7989 \pm 0.0086$ | $0.8002 \pm 0.0074$ |

| Brazilian_houses ↓ | | |
|---|---|---|
| Method | Single model | Ensemble |
| MLP | $0.0473 \pm 0.0179$ | $0.0440 \pm 0.0207$ |
| TabPFN | – | – |
| ResNet | $0.0505 \pm 0.0181$ | $0.0458 \pm 0.0207$ |
| DCN2 | $0.0477 \pm 0.0172$ | $0.0427 \pm 0.0207$ |
| SNN | $0.0630 \pm 0.0162$ | $0.0556 \pm 0.0175$ |
| Trompt | $0.0404 \pm 0.0266$ | – |
| AutoInt | $0.0470 \pm 0.0192$ | $0.0437 \pm 0.0217$ |
| MLP−Mixer | $0.0513 \pm 0.0234$ | $0.0484 \pm 0.0262$ |
| Excel* | $0.0450 \pm 0.0156$ | $0.0418 \pm 0.0190$ |
| SAINT | $0.0479 \pm 0.0205$ | – |
| FT−T | $0.0438 \pm 0.0181$ | $0.0412 \pm 0.0204$ |
| T2G | $0.0468 \pm 0.0165$ | $0.0436 \pm 0.0211$ |
| MLP$^{\ddagger-\text{lite}}$ | $0.0426 \pm 0.0180$ | $0.0397 \pm 0.0206$ |
| MLP$^{\ddagger}$ | $0.0437 \pm 0.0203$ | $0.0407 \pm 0.0230$ |
| MLP$^{\dagger}$ | $0.0421 \pm 0.0209$ | $0.0409 \pm 0.0226$ |
| XGBoost | $0.0541 \pm 0.0270$ | $0.0535 \pm 0.0287$ |
| LightGBM | $0.0603 \pm 0.0249$ | $0.0589 \pm 0.0271$ |
| CatBoost | $0.0468 \pm 0.0312$ | $0.0456 \pm 0.0332$ |
| TabR | $0.0490 \pm 0.0152$ | $0.0454 \pm 0.0170$ |
| TabR$^{\ddagger}$ | $0.0451 \pm 0.0163$ | – |
| MNCA | $0.0527 \pm 0.0157$ | $0.0509 \pm 0.0180$ |
| MNCA$^{\ddagger}$ | $0.0553 \pm 0.0192$ | $0.0511 \pm 0.0191$ |
| TabM$^{\spadesuit}$ | $0.0443 \pm 0.0213$ | $0.0431 \pm 0.0233$ |
| TabM | $0.0417 \pm 0.0208$ | $0.0413 \pm 0.0222$ |
| TabM[G] | $0.0424 \pm 0.0201$ | – |
| TabM$_{\text{mini}}$ | $0.0433 \pm 0.0232$ | $0.0428 \pm 0.0247$ |
| TabM$^{\dagger}_{\text{mini}}$ | $0.0416 \pm 0.0215$ | $0.0406 \pm 0.0230$ |

| MagicTelescope ↑ | | |
|---|---|---|
| Method | Single model | Ensemble |
| MLP | $0.8539 \pm 0.0060$ | $0.8566 \pm 0.0061$ |
| TabPFN | – | $0.8579 \pm 0.0064$ |
| ResNet | $0.8589 \pm 0.0068$ | $0.8651 \pm 0.0049$ |
| DCN2 | $0.8432 \pm 0.0074$ | $0.8490 \pm 0.0046$ |
| SNN | $0.8536 \pm 0.0052$ | $0.8567 \pm 0.0047$ |
| Trompt | $0.8605 \pm 0.0102$ | – |
| AutoInt | $0.8522 \pm 0.0056$ | $0.8560 \pm 0.0034$ |
| MLP−Mixer | $0.8571 \pm 0.0080$ | $0.8624 \pm 0.0044$ |
| Excel* | $0.8480 \pm 0.0090$ | $0.8543 \pm 0.0075$ |
| SAINT | $0.8595 \pm 0.0060$ | – |
| FT−T | $0.8588 \pm 0.0046$ | $0.8643 \pm 0.0037$ |
| T2G | $0.8553 \pm 0.0055$ | $0.8595 \pm 0.0051$ |
| MLP$^{\ddagger-\text{lite}}$ | $0.8591 \pm 0.0061$ | $0.8626 \pm 0.0044$ |
| MLP$^{\ddagger}$ | $0.8575 \pm 0.0056$ | $0.8605 \pm 0.0051$ |
| MLP$^{\dagger}$ | $0.8593 \pm 0.0054$ | $0.8621 \pm 0.0037$ |
| XGBoost | $0.8550 \pm 0.0094$ | $0.8589 \pm 0.0110$ |
| LightGBM | $0.8547 \pm 0.0085$ | $0.8556 \pm 0.0086$ |
| CatBoost | $0.8586 \pm 0.0070$ | $0.8588 \pm 0.0077$ |
| TabR | $0.8682 \pm 0.0058$ | $0.8729 \pm 0.0038$ |
| TabR$^{\ddagger}$ | $0.8641 \pm 0.0052$ | – |
| MNCA | $0.8602 \pm 0.0061$ | $0.8628 \pm 0.0041$ |
| MNCA$^{\ddagger}$ | $0.8622 \pm 0.0085$ | $0.8681 \pm 0.0064$ |
| TabM$^{\spadesuit}$ | $0.8607 \pm 0.0058$ | $0.8622 \pm 0.0050$ |
| TabM | $0.8622 \pm 0.0049$ | $0.8631 \pm 0.0046$ |
| TabM[G] | $0.8600 \pm 0.0055$ | – |
| TabM$_{\text{mini}}$ | $0.8606 \pm 0.0055$ | $0.8618 \pm 0.0049$ |
| TabM$^{\dagger}_{\text{mini}}$ | $0.8644 \pm 0.0088$ | $0.8673 \pm 0.0075$ |

| Ailerons ↓ | | |
|---|---|---|
| Method | Single model | Ensemble |
| MLP | $0.0002 \pm 0.0000$ | $0.0002 \pm 0.0000$ |
| TabPFN | – | – |
| ResNet | $0.0002 \pm 0.0000$ | $0.0002 \pm 0.0000$ |
| DCN2 | $0.0002 \pm 0.0000$ | $0.0002 \pm 0.0000$ |
| SNN | $0.0002 \pm 0.0000$ | $0.0002 \pm 0.0000$ |
| Trompt | $0.0002 \pm 0.0000$ | – |
| AutoInt | $0.0002 \pm 0.0000$ | $0.0002 \pm 0.0000$ |
| MLP−Mixer | $0.0002 \pm 0.0000$ | $0.0002 \pm 0.0000$ |
| Excel* | $0.0002 \pm 0.0000$ | $0.0002 \pm 0.0000$ |
| SAINT | $0.0002 \pm 0.0000$ | – |
| FT−T | $0.0002 \pm 0.0000$ | $0.0002 \pm 0.0000$ |
| T2G | $0.0002 \pm 0.0000$ | $0.0002 \pm 0.0000$ |
| MLP$^{\ddagger-\text{lite}}$ | $0.0002 \pm 0.0000$ | $0.0002 \pm 0.0000$ |
| MLP$^{\ddagger}$ | $0.0002 \pm 0.0000$ | $0.0002 \pm 0.0000$ |
| MLP$^{\dagger}$ | $0.0002 \pm 0.0000$ | $0.0002 \pm 0.0000$ |
| XGBoost | $0.0002 \pm 0.0000$ | $0.0002 \pm 0.0000$ |
| LightGBM | $0.0002 \pm 0.0000$ | $0.0002 \pm 0.0000$ |
| CatBoost | $0.0002 \pm 0.0000$ | $0.0002 \pm 0.0000$ |
| TabR | $0.0002 \pm 0.0000$ | $0.0002 \pm 0.0000$ |
| TabR$^{\ddagger}$ | $0.0002 \pm 0.0000$ | – |
| MNCA | $0.0002 \pm 0.0000$ | $0.0002 \pm 0.0000$ |
| MNCA$^{\ddagger}$ | $0.0002 \pm 0.0000$ | $0.0002 \pm 0.0000$ |
| TabM$^{\spadesuit}$ | $0.0002 \pm 0.0000$ | $0.0002 \pm 0.0000$ |
| TabM | $0.0002 \pm 0.0000$ | $0.0002 \pm 0.0000$ |
| TabM[G] | $0.0002 \pm 0.0000$ | – |
| TabM$_{\text{mini}}$ | $0.0002 \pm 0.0000$ | $0.0002 \pm 0.0000$ |
| TabM$^{\dagger}_{\text{mini}}$ | $0.0002 \pm 0.0000$ | $0.0002 \pm 0.0000$ |

| MiamiHousing2016 ↓ | | |
|---|---|---|
| Method | Single model | Ensemble |
| MLP | $0.1614 \pm 0.0033$ | $0.1574 \pm 0.0043$ |
| TabPFN | – | – |
| ResNet | $0.1548 \pm 0.0030$ | $0.1511 \pm 0.0027$ |
| DCN2 | $0.1683 \pm 0.0099$ | $0.1575 \pm 0.0047$ |
| SNN | $0.1618 \pm 0.0029$ | $0.1557 \pm 0.0021$ |
| Trompt | $0.1478 \pm 0.0028$ | – |
| AutoInt | $0.1537 \pm 0.0035$ | $0.1478 \pm 0.0027$ |
| MLP−Mixer | $0.1527 \pm 0.0037$ | $0.1479 \pm 0.0033$ |
| Excel* | $0.1519 \pm 0.0038$ | $0.1442 \pm 0.0022$ |
| SAINT | $0.1507 \pm 0.0022$ | – |
| FT−T | $0.1514 \pm 0.0029$ | $0.1462 \pm 0.0031$ |
| T2G | $0.1523 \pm 0.0023$ | $0.1478 \pm 0.0024$ |
| MLP$^{\ddagger-\text{lite}}$ | $0.1514 \pm 0.0025$ | $0.1479 \pm 0.0017$ |
| MLP$^{\ddagger}$ | $0.1512 \pm 0.0019$ | $0.1470 \pm 0.0024$ |
| MLP$^{\dagger}$ | $0.1461 \pm 0.0015$ | $0.1433 \pm 0.0022$ |
| XGBoost | $0.1440 \pm 0.0029$ | $0.1434 \pm 0.0029$ |
| LightGBM | $0.1461 \pm 0.0025$ | $0.1455 \pm 0.0030$ |
| CatBoost | $0.1417 \pm 0.0021$ | $0.1408 \pm 0.0026$ |
| TabR | $0.1417 \pm 0.0025$ | $0.1390 \pm 0.0020$ |
| TabR$^{\ddagger}$ | $0.1392 \pm 0.0023$ | – |
| MNCA | $0.1503 \pm 0.0040$ | $0.1477 \pm 0.0032$ |
| MNCA$^{\ddagger}$ | $0.1475 \pm 0.0031$ | $0.1438 \pm 0.0024$ |
| TabM$^{\spadesuit}$ | $0.1483 \pm 0.0030$ | $0.1465 \pm 0.0029$ |
| TabM | $0.1478 \pm 0.0012$ | $0.1471 \pm 0.0011$ |
| TabM[G] | $0.1482 \pm 0.0012$ | – |
| TabM$_{\text{mini}}$ | $0.1481 \pm 0.0021$ | $0.1471 \pm 0.0020$ |
| TabM$^{\dagger}_{\text{mini}}$ | $0.1408 \pm 0.0019$ | $0.1399 \pm 0.0018$ |

| OnlineNewsPopularity ↓ | | |
|---|---|---|
| Method | Single model | Ensemble |
| MLP | $0.8643 \pm 0.0007$ | $0.8632 \pm 0.0005$ |
| TabPFN | – | – |
| ResNet | $0.8665 \pm 0.0011$ | $0.8639 \pm 0.0000$ |
| DCN2 | $0.8714 \pm 0.0013$ | $0.8648 \pm 0.0004$ |
| SNN | $0.8692 \pm 0.0015$ | $0.8665 \pm 0.0005$ |
| Trompt | $0.8623 \pm nan$ | – |
| AutoInt | $0.8636 \pm 0.0022$ | $0.8596 \pm 0.0008$ |
| MLP−Mixer | $0.8615 \pm 0.0008$ | $0.8598 \pm 0.0004$ |
| Excel* | $0.8605 \pm 0.0024$ | $0.8556 \pm nan$ |
| SAINT | $0.8600 \pm 0.0007$ | – |
| FT−T | $0.8629 \pm 0.0019$ | $0.8603 \pm 0.0000$ |
| T2G | $0.8632 \pm 0.0009$ | $0.8572 \pm nan$ |
| MLP$^{\ddagger-\text{lite}}$ | $0.8604 \pm 0.0009$ | $0.8591 \pm 0.0004$ |
| MLP$^{\ddagger}$ | $0.8594 \pm 0.0004$ | $0.8585 \pm 0.0001$ |
| MLP$^{\dagger}$ | $0.8585 \pm 0.0003$ | $0.8581 \pm 0.0001$ |
| XGBoost | $0.8545 \pm 0.0002$ | $0.8543 \pm 0.0000$ |
| LightGBM | $0.8546 \pm 0.0002$ | $0.8544 \pm 0.0000$ |
| CatBoost | $0.8532 \pm 0.0003$ | $0.8527 \pm 0.0001$ |
| TabR | $0.8677 \pm 0.0013$ | $0.8633 \pm 0.0009$ |
| TabR$^{\ddagger}$ | $0.8624 \pm 0.0011$ | – |
| MNCA | $0.8651 \pm 0.0003$ | $0.8650 \pm 0.0002$ |
| MNCA$^{\ddagger}$ | $0.8647 \pm 0.0010$ | $0.8624 \pm 0.0006$ |
| TabM$^{\spadesuit}$ | $0.8584 \pm 0.0003$ | $0.8581 \pm 0.0001$ |
| TabM | $0.8579 \pm 0.0003$ | $0.8575 \pm 0.0001$ |
| TabM[G] | $0.8579 \pm 0.0004$ | – |
| TabM$_{\text{mini}}$ | $0.8588 \pm 0.0003$ | $0.8581 \pm 0.0003$ |
| TabM$^{\dagger}_{\text{mini}}$ | $0.8563 \pm 0.0004$ | $0.8558 \pm 0.0002$ |

| credit ↑ | | |
|---|---|---|
| Method | Single model | Ensemble |
| MLP | $0.7735 \pm 0.0042$ | $0.7729 \pm 0.0047$ |
| TabPFN | – | $0.7636 \pm 0.0045$ |
| ResNet | $0.7721 \pm 0.0033$ | $0.7738 \pm 0.0027$ |
| DCN2 | $0.7703 \pm 0.0034$ | $0.7746 \pm 0.0026$ |
| SNN | $0.7712 \pm 0.0045$ | $0.7716 \pm 0.0059$ |
| Trompt | $0.7740 \pm 0.0006$ | – |
| AutoInt | $0.7737 \pm 0.0050$ | $0.7765 \pm 0.0058$ |
| MLP−Mixer | $0.7748 \pm 0.0038$ | $0.7768 \pm 0.0059$ |
| Excel* | $0.7724 \pm 0.0038$ | $0.7740 \pm 0.0069$ |
| SAINT | $0.7739 \pm 0.0052$ | – |
| FT−T | $0.7745 \pm 0.0041$ | $0.7767 \pm 0.0040$ |
| T2G | $0.7744 \pm 0.0046$ | $0.7762 \pm 0.0057$ |
| MLP$^{\ddagger-\text{lite}}$ | $0.7749 \pm 0.0055$ | $0.7767 \pm 0.0075$ |
| MLP$^{\ddagger}$ | $0.7734 \pm 0.0034$ | $0.7747 \pm 0.0043$ |
| MLP$^{\dagger}$ | $0.7758 \pm 0.0040$ | $0.7772 \pm 0.0055$ |
| XGBoost | $0.7698 \pm 0.0027$ | $0.7706 \pm 0.0029$ |
| LightGBM | $0.7686 \pm 0.0028$ | $0.7726 \pm 0.0034$ |
| CatBoost | $0.7734 \pm 0.0035$ | $0.7752 \pm 0.0038$ |
| TabR | $0.7730 \pm 0.0043$ | $0.7740 \pm 0.0040$ |
| TabR$^{\ddagger}$ | $0.7723 \pm 0.0037$ | – |
| MNCA | $0.7739 \pm 0.0032$ | $0.7757 \pm 0.0026$ |
| MNCA$^{\ddagger}$ | $0.7734 \pm 0.0045$ | $0.7754 \pm 0.0040$ |
| TabM$^{\spadesuit}$ | $0.7751 \pm 0.0042$ | $0.7755 \pm 0.0049$ |
| TabM | $0.7760 \pm 0.0043$ | $0.7771 \pm 0.0044$ |
| TabM[G] | $0.7754 \pm 0.0045$ | – |
| TabM$_{\text{mini}}$ | $0.7752 \pm 0.0047$ | $0.7754 \pm 0.0048$ |
| TabM$^{\dagger}_{\text{mini}}$ | $0.7761 \pm 0.0033$ | $0.7760 \pm 0.0028$ |

| elevators ↓ | | |
|---|---|---|
| Method | Single model | Ensemble |
| MLP | $0.0020 \pm 0.0001$ | $0.0019 \pm 0.0000$ |
| TabPFN | − | − |
| ResNet | $0.0019 \pm 0.0000$ | $0.0019 \pm 0.0000$ |
| DCN2 | $0.0019 \pm 0.0000$ | $0.0019 \pm 0.0000$ |
| SNN | $0.0020 \pm 0.0001$ | $0.0019 \pm 0.0000$ |
| Trompt | $0.0018 \pm 0.0000$ | − |
| AutoInt | $0.0019 \pm 0.0000$ | $0.0018 \pm 0.0000$ |
| MLP−Mixer | $0.0019 \pm 0.0000$ | $0.0018 \pm 0.0000$ |
| Excel* | $0.0019 \pm 0.0000$ | $0.0018 \pm 0.0000$ |
| SAINT | $0.0018 \pm 0.0000$ | − |
| FT−T | $0.0019 \pm 0.0000$ | $0.0018 \pm 0.0000$ |
| T2G | $0.0019 \pm 0.0000$ | $0.0018 \pm 0.0000$ |
| $\text{MLP}^{\ddagger-\text{lite}}$ | $0.0019 \pm 0.0000$ | $0.0018 \pm 0.0000$ |
| $\text{MLP}^{\ddagger}$ | $0.0018 \pm 0.0000$ | $0.0018 \pm 0.0000$ |
| $\text{MLP}^{\dagger}$ | $0.0018 \pm 0.0000$ | $0.0018 \pm 0.0000$ |
| XGBoost | $0.0020 \pm 0.0000$ | $0.0020 \pm 0.0000$ |
| LightGBM | $0.0020 \pm 0.0000$ | $0.0020 \pm 0.0000$ |
| CatBoost | $0.0020 \pm 0.0000$ | $0.0019 \pm 0.0000$ |
| TabR | $0.0049 \pm 0.0000$ | $0.0049 \pm 0.0000$ |
| $\text{TabR}^{\ddagger}$ | $0.0019 \pm 0.0001$ | − |
| MNCA | $0.0019 \pm 0.0000$ | $0.0019 \pm 0.0000$ |
| $\text{MNCA}^{\ddagger}$ | $0.0018 \pm 0.0000$ | $0.0018 \pm 0.0000$ |
| $\text{TabM}^{\spadesuit}$ | $0.0019 \pm 0.0000$ | $0.0018 \pm 0.0000$ |
| TabM | $0.0018 \pm 0.0000$ | $0.0018 \pm 0.0000$ |
| TabM[G] | $0.0018 \pm 0.0000$ | − |
| $\text{TabM}_{\text{mini}}$ | $0.0018 \pm 0.0000$ | $0.0018 \pm 0.0000$ |
| $\text{TabM}_{\text{mini}}^{\dagger}$ | $0.0018 \pm 0.0000$ | $0.0018 \pm 0.0000$ |

| fifa ↓ | | |
|---|---|---|
| Method | Single model | Ensemble |
| MLP | $0.8038 \pm 0.0124$ | $0.8011 \pm 0.0143$ |
| TabPFN | − | − |
| ResNet | $0.8025 \pm 0.0140$ | $0.7985 \pm 0.0149$ |
| DCN2 | $0.8046 \pm 0.0135$ | $0.7993 \pm 0.0129$ |
| SNN | $0.8074 \pm 0.0140$ | $0.8031 \pm 0.0147$ |
| Trompt | $0.7880 \pm 0.0180$ | − |
| AutoInt | $0.7923 \pm 0.0128$ | $0.7886 \pm 0.0127$ |
| MLP−Mixer | $0.7936 \pm 0.0119$ | $0.7903 \pm 0.0133$ |
| Excel* | $0.7909 \pm 0.0111$ | $0.7862 \pm 0.0161$ |
| SAINT | $0.7901 \pm 0.0118$ | − |
| FT−T | $0.7928 \pm 0.0132$ | $0.7888 \pm 0.0130$ |
| T2G | $0.7928 \pm 0.0139$ | $0.7904 \pm 0.0183$ |
| $\text{MLP}^{\ddagger-\text{lite}}$ | $0.7940 \pm 0.0118$ | $0.7898 \pm 0.0141$ |
| $\text{MLP}^{\ddagger}$ | $0.7907 \pm 0.0092$ | $0.7870 \pm 0.0096$ |
| $\text{MLP}^{\dagger}$ | $0.7806 \pm 0.0104$ | $0.7800 \pm 0.0114$ |
| XGBoost | $0.7800 \pm 0.0108$ | $0.7795 \pm 0.0114$ |
| LightGBM | $0.7806 \pm 0.0120$ | $0.7787 \pm 0.0122$ |
| CatBoost | $0.7835 \pm 0.0116$ | $0.7817 \pm 0.0114$ |
| TabR | $0.7902 \pm 0.0119$ | $0.7863 \pm 0.0120$ |
| $\text{TabR}^{\ddagger}$ | $0.7914 \pm 0.0136$ | − |
| MNCA | $0.7967 \pm 0.0138$ | $0.7933 \pm 0.0145$ |
| $\text{MNCA}^{\ddagger}$ | $0.7909 \pm 0.0107$ | $0.7866 \pm 0.0106$ |
| $\text{TabM}^{\spadesuit}$ | $0.7974 \pm 0.0144$ | $0.7954 \pm 0.0160$ |
| TabM | $0.7953 \pm 0.0135$ | $0.7942 \pm 0.0148$ |
| TabM[G] | $0.7948 \pm 0.0135$ | − |
| $\text{TabM}_{\text{mini}}$ | $0.7938 \pm 0.0156$ | $0.7920 \pm 0.0176$ |
| $\text{TabM}_{\text{mini}}^{\dagger}$ | $0.7771 \pm 0.0107$ | $0.7761 \pm 0.0117$ |

| house_sales ↓ | | |
|---|---|---|
| Method | Single model | Ensemble |
| MLP | $0.1790 \pm 0.0009$ | $0.1763 \pm 0.0003$ |
| TabPFN | − | − |
| ResNet | $0.1755 \pm 0.0014$ | $0.1738 \pm 0.0006$ |
| DCN2 | $0.1862 \pm 0.0032$ | $0.1778 \pm 0.0015$ |
| SNN | $0.1800 \pm 0.0008$ | $0.1770 \pm 0.0004$ |
| Trompt | $0.1667 \pm nan$ | − |
| AutoInt | $0.1700 \pm 0.0014$ | $0.1670 \pm 0.0008$ |
| MLP−Mixer | $0.1704 \pm 0.0007$ | $0.1690 \pm 0.0005$ |
| Excel* | $0.1713 \pm 0.0010$ | $0.1668 \pm nan$ |
| SAINT | $0.1713 \pm 0.0015$ | − |
| FT−T | $0.1690 \pm 0.0010$ | $0.1659 \pm 0.0004$ |
| T2G | $0.1689 \pm 0.0010$ | $0.1664 \pm nan$ |
| $\text{MLP}^{\ddagger-\text{lite}}$ | $0.1699 \pm 0.0008$ | $0.1687 \pm 0.0007$ |
| $\text{MLP}^{\ddagger}$ | $0.1690 \pm 0.0005$ | $0.1676 \pm 0.0003$ |
| $\text{MLP}^{\dagger}$ | $0.1687 \pm 0.0004$ | $0.1681 \pm 0.0001$ |
| XGBoost | $0.1694 \pm 0.0003$ | $0.1689 \pm 0.0001$ |
| LightGBM | $0.1692 \pm 0.0004$ | $0.1686 \pm 0.0001$ |
| CatBoost | $0.1669 \pm 0.0001$ | $0.1667 \pm 0.0000$ |
| TabR | $0.1689 \pm 0.0009$ | $0.1657 \pm 0.0003$ |
| $\text{TabR}^{\ddagger}$ | $0.1636 \pm 0.0009$ | − |
| MNCA | $0.1737 \pm 0.0013$ | $0.1714 \pm 0.0005$ |
| $\text{MNCA}^{\ddagger}$ | $0.1694 \pm 0.0007$ | $0.1670 \pm 0.0003$ |
| $\text{TabM}^{\spadesuit}$ | $0.1692 \pm 0.0011$ | $0.1680 \pm 0.0005$ |
| TabM | $0.1666 \pm 0.0003$ | $0.1662 \pm 0.0002$ |
| TabM[G] | $0.1667 \pm 0.0003$ | − |
| $\text{TabM}_{\text{mini}}$ | $0.1673 \pm 0.0004$ | $0.1668 \pm 0.0001$ |
| $\text{TabM}_{\text{mini}}^{\dagger}$ | $0.1652 \pm 0.0003$ | $0.1644 \pm 0.0001$ |

| medical_charges ↓ | | |
|---|---|---|
| Method | Single model | Ensemble |
| MLP | $0.0816 \pm 0.0001$ | $0.0814 \pm 0.0000$ |
| TabPFN | − | − |
| ResNet | $0.0824 \pm 0.0003$ | $0.0817 \pm 0.0001$ |
| DCN2 | $0.0818 \pm 0.0003$ | $0.0815 \pm 0.0001$ |
| SNN | $0.0827 \pm 0.0006$ | $0.0817 \pm 0.0001$ |
| Trompt | $0.0812 \pm nan$ | − |
| AutoInt | $0.0822 \pm 0.0007$ | $0.0814 \pm 0.0001$ |
| MLP−Mixer | $0.0814 \pm 0.0002$ | $0.0811 \pm 0.0000$ |
| Excel* | $0.0817 \pm 0.0004$ | $0.0813 \pm nan$ |
| SAINT | $0.0814 \pm 0.0002$ | − |
| FT−T | $0.0814 \pm 0.0002$ | $0.0812 \pm 0.0000$ |
| T2G | $0.0813 \pm 0.0002$ | $0.0811 \pm nan$ |
| $\text{MLP}^{\ddagger-\text{lite}}$ | $0.0812 \pm 0.0002$ | $0.0810 \pm 0.0000$ |
| $\text{MLP}^{\ddagger}$ | $0.0812 \pm 0.0001$ | $0.0809 \pm 0.0001$ |
| $\text{MLP}^{\dagger}$ | $0.0812 \pm 0.0000$ | $0.0811 \pm 0.0000$ |
| XGBoost | $0.0825 \pm 0.0001$ | $0.0825 \pm 0.0000$ |
| LightGBM | $0.0820 \pm 0.0000$ | $0.0820 \pm 0.0000$ |
| CatBoost | $0.0816 \pm 0.0000$ | $0.0815 \pm 0.0000$ |
| TabR | $0.0815 \pm 0.0002$ | $0.0812 \pm 0.0000$ |
| $\text{TabR}^{\ddagger}$ | $0.0811 \pm 0.0001$ | − |
| MNCA | $0.0811 \pm 0.0001$ | $0.0810 \pm 0.0000$ |
| $\text{MNCA}^{\ddagger}$ | $0.0809 \pm 0.0000$ | $0.0808 \pm 0.0000$ |
| $\text{TabM}^{\spadesuit}$ | $0.0813 \pm 0.0001$ | $0.0812 \pm 0.0000$ |
| TabM | $0.0812 \pm 0.0000$ | $0.0812 \pm 0.0000$ |
| TabM[G] | $0.0812 \pm 0.0000$ | − |
| $\text{TabM}_{\text{mini}}$ | $0.0813 \pm 0.0000$ | $0.0813 \pm 0.0000$ |
| $\text{TabM}_{\text{mini}}^{\dagger}$ | $0.0811 \pm 0.0001$ | $0.0811 \pm 0.0000$ |

| pol ↓ | | |
|---|---|---|
| Method | Single model | Ensemble |
| MLP | $5.5244 \pm 0.5768$ | $4.9945 \pm 0.5923$ |
| TabPFN | – | – |
| ResNet | $6.3739 \pm 0.6286$ | $5.8181 \pm 0.6054$ |
| DCN2 | $6.5374 \pm 0.9479$ | $5.1814 \pm 0.7775$ |
| SNN | $6.1816 \pm 0.7366$ | $5.5959 \pm 0.8243$ |
| Trompt | $3.2337 \pm 0.0605$ | – |
| AutoInt | $3.3295 \pm 0.3379$ | $2.7999 \pm 0.1776$ |
| MLP−Mixer | $3.2011 \pm 0.2921$ | $2.8698 \pm 0.2577$ |
| Excel* | $3.0682 \pm 0.2389$ | $2.5816 \pm 0.0368$ |
| SAINT | $2.7203 \pm 0.1858$ | – |
| FT−T | $2.6974 \pm 0.1666$ | $2.3718 \pm 0.0724$ |
| T2G | $2.9539 \pm 0.1994$ | $2.6282 \pm 0.0730$ |
| $\text{MLP}^{\ddagger-\text{lite}}$ | $2.8239 \pm 0.2173$ | $2.5266 \pm 0.0605$ |
| $\text{MLP}^{\ddagger}$ | $2.5452 \pm 0.1221$ | $2.3700 \pm 0.0867$ |
| $\text{MLP}^{\dagger}$ | $2.4958 \pm 0.1292$ | $2.3651 \pm 0.1223$ |
| XGBoost | $4.2963 \pm 0.0644$ | $4.2548 \pm 0.0488$ |
| LightGBM | $4.2320 \pm 0.3369$ | $4.1880 \pm 0.3110$ |
| CatBoost | $3.6320 \pm 0.1006$ | $3.5505 \pm 0.0896$ |
| TabR | $6.0708 \pm 0.5368$ | $5.5578 \pm 0.4036$ |
| $\text{TabR}^{\ddagger}$ | $2.5770 \pm 0.1689$ | – |
| MNCA | $5.7878 \pm 0.4884$ | $5.3773 \pm 0.5463$ |
| $\text{MNCA}^{\ddagger}$ | $2.9083 \pm 0.1364$ | $2.6717 \pm 0.0530$ |
| $\text{TabM}^{\spadesuit}$ | $3.3595 \pm 0.4017$ | $3.2130 \pm 0.3979$ |
| TabM | $3.0198 \pm 0.2975$ | $2.9595 \pm 0.3107$ |
| TabM[G] | $3.0358 \pm 0.3077$ | – |
| $\text{TabM}_{\text{mini}}$ | $3.1351 \pm 0.1952$ | $3.0478 \pm 0.2061$ |
| $\text{TabM}_{\text{mini}}^{\dagger}$ | $2.2808 \pm 0.0343$ | $2.2383 \pm 0.0111$ |

| superconduct ↓ | | |
|---|---|---|
| Method | Single model | Ensemble |
| MLP | $10.8740 \pm 0.0868$ | $10.4118 \pm 0.0429$ |
| TabPFN | – | – |
| ResNet | $10.7711 \pm 0.1454$ | $10.3495 \pm 0.0168$ |
| DCN2 | $10.8108 \pm 0.0957$ | $10.4342 \pm 0.0179$ |
| SNN | $10.8562 \pm 0.1300$ | $10.3342 \pm 0.0509$ |
| Trompt | $10.4442 \pm nan$ | – |
| AutoInt | $11.0019 \pm 0.1391$ | $10.4469 \pm 0.0521$ |
| MLP−Mixer | $10.7502 \pm 0.0800$ | $10.3281 \pm 0.0450$ |
| Excel* | $11.0879 \pm 0.1571$ | $10.4094 \pm nan$ |
| SAINT | $10.7807 \pm 0.1074$ | – |
| FT−T | $10.8256 \pm 0.1692$ | $10.3391 \pm 0.0794$ |
| T2G | $10.8310 \pm 0.1406$ | $10.3017 \pm nan$ |
| $\text{MLP}^{\ddagger-\text{lite}}$ | $10.5058 \pm 0.0758$ | $10.2322 \pm 0.0463$ |
| $\text{MLP}^{\ddagger}$ | $10.5061 \pm 0.0330$ | $10.2440 \pm 0.0127$ |
| $\text{MLP}^{\dagger}$ | $10.7220 \pm 0.0757$ | $10.3758 \pm 0.0606$ |
| XGBoost | $10.1610 \pm 0.0201$ | $10.1413 \pm 0.0025$ |
| LightGBM | $10.1634 \pm 0.0118$ | $10.1552 \pm 0.0050$ |
| CatBoost | $10.2422 \pm 0.0222$ | $10.2116 \pm 0.0058$ |
| TabR | $10.8842 \pm 0.1073$ | $10.4800 \pm 0.0280$ |
| $\text{TabR}^{\ddagger}$ | $10.3835 \pm 0.0562$ | – |
| MNCA | $10.4419 \pm 0.0640$ | $10.2926 \pm 0.0261$ |
| $\text{MNCA}^{\ddagger}$ | $10.5651 \pm 0.0616$ | $10.3155 \pm 0.0253$ |
| $\text{TabM}^{\spadesuit}$ | $10.3379 \pm 0.0338$ | $10.1943 \pm 0.0291$ |
| TabM | $10.2628 \pm 0.0275$ | $10.2300 \pm 0.0108$ |
| TabM[G] | $10.2572 \pm 0.0463$ | – |
| $\text{TabM}_{\text{mini}}$ | $10.2472 \pm 0.0208$ | $10.2094 \pm 0.0057$ |
| $\text{TabM}_{\text{mini}}^{\dagger}$ | $10.1326 \pm 0.0186$ | $10.0866 \pm 0.0070$ |

| jannis ↑ | | |
|---|---|---|
| Method | Single model | Ensemble |
| MLP | $0.7840 \pm 0.0018$ | $0.7872 \pm 0.0007$ |
| TabPFN | – | $0.7419 \pm 0.0018$ |
| ResNet | $0.7923 \pm 0.0024$ | $0.7958 \pm 0.0010$ |
| DCN2 | $0.7712 \pm 0.0029$ | $0.7825 \pm 0.0009$ |
| SNN | $0.7818 \pm 0.0025$ | $0.7859 \pm 0.0011$ |
| Trompt | $0.8027 \pm nan$ | – |
| AutoInt | $0.7933 \pm 0.0018$ | $0.7983 \pm 0.0013$ |
| MLP−Mixer | $0.7927 \pm 0.0025$ | $0.8019 \pm 0.0012$ |
| Excel* | $0.7954 \pm 0.0015$ | $0.8021 \pm nan$ |
| SAINT | $0.7971 \pm 0.0028$ | – |
| FT−T | $0.7940 \pm 0.0028$ | $0.7998 \pm 0.0006$ |
| T2G | $0.7998 \pm 0.0024$ | $0.8052 \pm nan$ |
| $\text{MLP}^{\ddagger-\text{lite}}$ | $0.7923 \pm 0.0018$ | $0.7945 \pm 0.0010$ |
| $\text{MLP}^{\ddagger}$ | $0.7947 \pm 0.0017$ | $0.7967 \pm 0.0011$ |
| $\text{MLP}^{\dagger}$ | $0.7891 \pm 0.0013$ | $0.7900 \pm 0.0006$ |
| XGBoost | $0.7967 \pm 0.0019$ | $0.7998 \pm 0.0007$ |
| LightGBM | $0.7956 \pm 0.0017$ | $0.7968 \pm 0.0005$ |
| CatBoost | $0.7985 \pm 0.0018$ | $0.8009 \pm 0.0012$ |
| TabR | $0.7983 \pm 0.0022$ | $0.8023 \pm 0.0018$ |
| $\text{TabR}^{\ddagger}$ | $0.8051 \pm 0.0023$ | – |
| MNCA | $0.7993 \pm 0.0019$ | $0.8042 \pm 0.0013$ |
| $\text{MNCA}^{\ddagger}$ | $0.8068 \pm 0.0021$ | $0.8128 \pm 0.0007$ |
| $\text{TabM}^{\spadesuit}$ | $0.8066 \pm 0.0015$ | $0.8075 \pm 0.0004$ |
| TabM | $0.8080 \pm 0.0019$ | $0.8102 \pm 0.0017$ |
| TabM[G] | $0.8064 \pm 0.0018$ | – |
| $\text{TabM}_{\text{mini}}$ | $0.8053 \pm 0.0012$ | $0.8066 \pm 0.0001$ |
| $\text{TabM}_{\text{mini}}^{\dagger}$ | $0.8078 \pm 0.0008$ | $0.8086 \pm 0.0005$ |

| MiniBooNE ↑ | | |
|---|---|---|
| Method | Single model | Ensemble |
| MLP | $0.9480 \pm 0.0007$ | $0.9498 \pm 0.0001$ |
| TabPFN | – | $0.9266 \pm 0.0012$ |
| ResNet | $0.9488 \pm 0.0011$ | $0.9504 \pm 0.0005$ |
| DCN2 | $0.9433 \pm 0.0011$ | $0.9470 \pm 0.0010$ |
| SNN | $0.9476 \pm 0.0013$ | $0.9491 \pm 0.0010$ |
| Trompt | $0.9473 \pm nan$ | – |
| AutoInt | $0.9447 \pm 0.0014$ | $0.9473 \pm 0.0010$ |
| MLP−Mixer | $0.9446 \pm 0.0014$ | $0.9483 \pm 0.0002$ |
| Excel* | $0.9430 \pm 0.0015$ | $0.9451 \pm nan$ |
| SAINT | $0.9471 \pm 0.0009$ | – |
| FT−T | $0.9467 \pm 0.0014$ | $0.9486 \pm 0.0010$ |
| T2G | $0.9475 \pm 0.0014$ | $0.9508 \pm nan$ |
| $\text{MLP}^{\ddagger-\text{lite}}$ | $0.9466 \pm 0.0009$ | $0.9478 \pm 0.0004$ |
| $\text{MLP}^{\ddagger}$ | $0.9473 \pm 0.0010$ | $0.9493 \pm 0.0004$ |
| $\text{MLP}^{\dagger}$ | $0.9482 \pm 0.0008$ | $0.9492 \pm 0.0001$ |
| XGBoost | $0.9436 \pm 0.0006$ | $0.9452 \pm 0.0003$ |
| LightGBM | $0.9422 \pm 0.0009$ | $0.9427 \pm 0.0003$ |
| CatBoost | $0.9453 \pm 0.0008$ | $0.9459 \pm 0.0005$ |
| TabR | $0.9487 \pm 0.0008$ | $0.9500 \pm 0.0002$ |
| $\text{TabR}^{\ddagger}$ | $0.9475 \pm 0.0007$ | – |
| MNCA | $0.9488 \pm 0.0010$ | $0.9505 \pm 0.0001$ |
| $\text{MNCA}^{\ddagger}$ | $0.9493 \pm 0.0012$ | $0.9501 \pm 0.0008$ |
| $\text{TabM}^{\spadesuit}$ | $0.9500 \pm 0.0005$ | $0.9505 \pm 0.0002$ |
| TabM | $0.9503 \pm 0.0006$ | $0.9501 \pm 0.0002$ |
| TabM[G] | $0.9496 \pm 0.0010$ | – |
| $\text{TabM}_{\text{mini}}$ | $0.9495 \pm 0.0005$ | $0.9500 \pm 0.0002$ |
| $\text{TabM}_{\text{mini}}^{\dagger}$ | $0.9490 \pm 0.0004$ | $0.9492 \pm 0.0002$ |

| nyc-taxi-green-dec-2016 ↓ | | |
|---|---|---|
| Method | Single model | Ensemble |
| MLP | $0.3951 \pm 0.0009$ | $0.3921 \pm 0.0003$ |
| TabPFN | – | – |
| ResNet | $0.3899 \pm 0.0016$ | $0.3873 \pm 0.0009$ |
| DCN2 | $0.3919 \pm 0.0009$ | $0.3889 \pm 0.0003$ |
| SNN | $0.3933 \pm 0.0013$ | $0.3899 \pm 0.0004$ |
| Trompt | $0.3979 \pm nan$ | – |
| AutoInt | $0.4084 \pm 0.0256$ | $0.3967 \pm 0.0059$ |
| MLP−Mixer | $0.3914 \pm 0.0026$ | $0.3861 \pm 0.0013$ |
| Excel* | $0.3969 \pm 0.0036$ | $0.3897 \pm nan$ |
| SAINT | $0.3905 \pm 0.0013$ | – |
| FT−T | $0.3937 \pm 0.0064$ | $0.3889 \pm 0.0018$ |
| T2G | $0.3908 \pm 0.0045$ | $0.3858 \pm nan$ |
| $\text{MLP}^{\ddagger-\text{lite}}$ | $0.3812 \pm 0.0018$ | $0.3761 \pm 0.0016$ |
| $\text{MLP}^{\ddagger}$ | $0.3795 \pm 0.0016$ | $0.3733 \pm 0.0013$ |
| $\text{MLP}^{\dagger}$ | $0.3680 \pm 0.0006$ | $0.3653 \pm 0.0005$ |
| XGBoost | $0.3792 \pm 0.0002$ | $0.3787 \pm 0.0000$ |
| LightGBM | $0.3688 \pm 0.0002$ | $0.3684 \pm 0.0000$ |
| CatBoost | $0.3647 \pm 0.0005$ | $0.3632 \pm 0.0003$ |
| TabR | $0.3577 \pm 0.0222$ | $0.3380 \pm 0.0027$ |
| $\text{TabR}^{\ddagger}$ | $0.3725 \pm 0.0091$ | – |
| MNCA | $0.3728 \pm 0.0012$ | $0.3720 \pm 0.0010$ |
| $\text{MNCA}^{\ddagger}$ | $0.3536 \pm 0.0052$ | $0.3407 \pm 0.0009$ |
| $\text{TabM}^{\spadesuit}$ | $0.3866 \pm 0.0006$ | $0.3855 \pm 0.0003$ |
| TabM | $0.3849 \pm 0.0005$ | $0.3843 \pm 0.0002$ |
| TabM[G] | $0.3848 \pm 0.0005$ | – |
| $\text{TabM}_{\text{mini}}$ | $0.3853 \pm 0.0005$ | $0.3845 \pm 0.0003$ |
| $\text{TabM}^{\dagger}_{\text{mini}}$ | $0.3485 \pm 0.0038$ | $0.3448 \pm 0.0020$ |

| particulate-matter-ukair-2017 ↓ | | |
|---|---|---|
| Method | Single model | Ensemble |
| MLP | $0.3759 \pm 0.0004$ | $0.3729 \pm 0.0003$ |
| TabPFN | – | – |
| ResNet | $0.3743 \pm 0.0007$ | $0.3718 \pm 0.0005$ |
| DCN2 | $0.3759 \pm 0.0012$ | $0.3738 \pm 0.0004$ |
| SNN | $0.3790 \pm 0.0007$ | $0.3744 \pm 0.0002$ |
| Trompt | $0.3700 \pm nan$ | – |
| AutoInt | $0.3723 \pm 0.0011$ | $0.3692 \pm 0.0010$ |
| MLP−Mixer | $0.3741 \pm 0.0010$ | $0.3698 \pm 0.0004$ |
| Excel* | $0.3699 \pm 0.0014$ | $0.3652 \pm nan$ |
| SAINT | $0.3704 \pm 0.0014$ | – |
| FT−T | $0.3735 \pm 0.0012$ | $0.3686 \pm 0.0004$ |
| T2G | $0.3676 \pm 0.0024$ | $0.3631 \pm nan$ |
| $\text{MLP}^{\ddagger-\text{lite}}$ | $0.3665 \pm 0.0008$ | $0.3642 \pm 0.0003$ |
| $\text{MLP}^{\ddagger}$ | $0.3657 \pm 0.0007$ | $0.3629 \pm 0.0002$ |
| $\text{MLP}^{\dagger}$ | $0.3649 \pm 0.0011$ | $0.3637 \pm 0.0008$ |
| XGBoost | $0.3641 \pm 0.0001$ | $0.3640 \pm 0.0000$ |
| LightGBM | $0.3637 \pm 0.0001$ | $0.3635 \pm 0.0000$ |
| CatBoost | $0.3647 \pm 0.0004$ | $0.3637 \pm 0.0002$ |
| TabR | $0.3613 \pm 0.0005$ | $0.3590 \pm 0.0002$ |
| $\text{TabR}^{\ddagger}$ | $0.3596 \pm 0.0004$ | – |
| MNCA | $0.3670 \pm 0.0004$ | $0.3649 \pm 0.0002$ |
| $\text{MNCA}^{\ddagger}$ | $0.3646 \pm 0.0001$ | $0.3643 \pm 0.0000$ |
| $\text{TabM}^{\spadesuit}$ | $0.3686 \pm 0.0006$ | $0.3679 \pm 0.0003$ |
| TabM | $0.3671 \pm 0.0007$ | $0.3665 \pm 0.0002$ |
| TabM[G] | $0.3667 \pm 0.0009$ | – |
| $\text{TabM}_{\text{mini}}$ | $0.3664 \pm 0.0006$ | $0.3655 \pm 0.0002$ |
| $\text{TabM}^{\dagger}_{\text{mini}}$ | $0.3593 \pm 0.0004$ | $0.3589 \pm 0.0000$ |

| road-safety ↑ | | |
|---|---|---|
| Method | Single model | Ensemble |
| MLP | $0.7857 \pm 0.0019$ | $0.7873 \pm 0.0004$ |
| TabPFN | – | $0.7338 \pm 0.0032$ |
| ResNet | $0.7875 \pm 0.0007$ | $0.7898 \pm 0.0008$ |
| DCN2 | $0.7781 \pm 0.0014$ | $0.7823 \pm 0.0012$ |
| SNN | $0.7847 \pm 0.0010$ | $0.7865 \pm 0.0002$ |
| Trompt | $0.7804 \pm nan$ | – |
| AutoInt | $0.7826 \pm 0.0030$ | $0.7883 \pm 0.0013$ |
| MLP−Mixer | $0.7878 \pm 0.0032$ | $0.7919 \pm 0.0015$ |
| Excel* | $0.7864 \pm 0.0053$ | $0.7907 \pm nan$ |
| SAINT | $0.7584 \pm 0.0584$ | – |
| FT−T | $0.7907 \pm 0.0012$ | $0.7943 \pm 0.0007$ |
| T2G | $0.7912 \pm 0.0026$ | $0.7961 \pm nan$ |
| $\text{MLP}^{\ddagger-\text{lite}}$ | $0.7867 \pm 0.0018$ | $0.7903 \pm 0.0002$ |
| $\text{MLP}^{\ddagger}$ | $0.7853 \pm 0.0014$ | $0.7881 \pm 0.0007$ |
| $\text{MLP}^{\dagger}$ | $0.7899 \pm 0.0009$ | $0.7935 \pm 0.0003$ |
| XGBoost | $0.8101 \pm 0.0017$ | $0.8129 \pm 0.0004$ |
| LightGBM | $0.7982 \pm 0.0012$ | $0.7996 \pm 0.0005$ |
| CatBoost | $0.8012 \pm 0.0009$ | $0.8022 \pm 0.0002$ |
| TabR | $0.8403 \pm 0.0014$ | $0.8441 \pm 0.0005$ |
| $\text{TabR}^{\ddagger}$ | $0.8374 \pm 0.0013$ | – |
| MNCA | $0.8080 \pm 0.0013$ | $0.8121 \pm 0.0006$ |
| $\text{MNCA}^{\ddagger}$ | $0.8232 \pm 0.0017$ | $0.8287 \pm 0.0008$ |
| $\text{TabM}^{\spadesuit}$ | $0.7946 \pm 0.0013$ | $0.7961 \pm 0.0005$ |
| TabM | $0.7958 \pm 0.0011$ | $0.7968 \pm 0.0004$ |
| TabM[G] | $0.7954 \pm 0.0016$ | – |
| $\text{TabM}_{\text{mini}}$ | $0.7933 \pm 0.0030$ | $0.7970 \pm 0.0006$ |
| $\text{TabM}^{\dagger}_{\text{mini}}$ | $0.7999 \pm 0.0023$ | $0.8059 \pm 0.0012$ |

| year ↓ | | |
|---|---|---|
| Method | Single model | Ensemble |
| MLP | $8.9628 \pm 0.0232$ | $8.8931 \pm 0.0066$ |
| TabPFN | – | – |
| ResNet | $8.9658 \pm 0.0239$ | $8.8755 \pm 0.0066$ |
| DCN2 | $9.2761 \pm 0.0401$ | $9.0640 \pm 0.0156$ |
| SNN | $9.0054 \pm 0.0256$ | $8.9351 \pm 0.0073$ |
| Trompt | $8.9707 \pm nan$ | – |
| AutoInt | $9.0430 \pm 0.0280$ | $8.9619 \pm 0.0092$ |
| MLP−Mixer | $8.9589 \pm 0.0182$ | $8.9086 \pm 0.0177$ |
| Excel* | $9.0395 \pm 0.0266$ | $8.9551 \pm nan$ |
| SAINT | $9.0248 \pm 0.0225$ | – |
| FT−T | $9.0005 \pm 0.0215$ | $8.9360 \pm 0.0013$ |
| T2G | $8.9775 \pm 0.0138$ | $8.8979 \pm nan$ |
| $\text{MLP}^{\ddagger-\text{lite}}$ | $8.9355 \pm 0.0103$ | $8.9063 \pm 0.0030$ |
| $\text{MLP}^{\ddagger}$ | $8.9455 \pm 0.0173$ | $8.9083 \pm 0.0046$ |
| $\text{MLP}^{\dagger}$ | $8.9379 \pm 0.0206$ | $8.8753 \pm 0.0038$ |
| XGBoost | $9.0307 \pm 0.0028$ | $9.0245 \pm 0.0015$ |
| LightGBM | $9.0200 \pm 0.0025$ | $9.0128 \pm 0.0015$ |
| CatBoost | $9.0370 \pm 0.0073$ | $9.0054 \pm 0.0028$ |
| TabR | $9.0069 \pm 0.0152$ | $8.9132 \pm 0.0088$ |
| $\text{TabR}^{\ddagger}$ | $8.9721 \pm 0.0105$ | – |
| MNCA | $8.9476 \pm 0.0152$ | $8.8977 \pm 0.0037$ |
| $\text{MNCA}^{\ddagger}$ | $8.8973 \pm 0.0082$ | $8.8550 \pm 0.0031$ |
| $\text{TabM}^{\spadesuit}$ | $8.8701 \pm 0.0110$ | $8.8517 \pm 0.0022$ |
| TabM | $8.8705 \pm 0.0043$ | $8.8642 \pm 0.0028$ |
| TabM[G] | $8.8723 \pm 0.0080$ | – |
| $\text{TabM}_{\text{mini}}$ | $8.9164 \pm 0.0089$ | $8.9021 \pm 0.0036$ |
| $\text{TabM}^{\dagger}_{\text{mini}}$ | $8.8737 \pm 0.0119$ | $8.8564 \pm 0.0054$ |

Table 20: Extended results for TabReD Rubachev et al. (2024) benchmark. Results are grouped by datasets. One ensemble consists of five models trained independently under different random seeds.

| sberbank-housing ↓ | | |
| --- | --- | --- |
| Method | Single model | Ensemble |
| MLP | $0.2529 \pm 0.0078$ | $0.2474 \pm 0.0052$ |
| TabPFN | – | – |
| ResNet | – | – |
| DCN2 | $0.2616 \pm 0.0049$ | $0.2506 \pm 0.0015$ |
| SNN | $0.2671 \pm 0.0140$ | $0.2555 \pm 0.0033$ |
| Trompt | $0.2509 \pm nan$ | – |
| AutoInt | – | – |
| MLP−Mixer | – | – |
| Excel* | $0.2533 \pm 0.0046$ | $0.2485 \pm nan$ |
| SAINT | $0.2467 \pm 0.0019$ | – |
| FT−T | $0.2440 \pm 0.0038$ | $0.2367 \pm 0.0010$ |
| T2G | $0.2416 \pm 0.0025$ | $0.2343 \pm nan$ |
| MLP$^{\ddagger-\text{lite}}$ | $0.2528 \pm 0.0055$ | $0.2503 \pm 0.0029$ |
| MLP$^\ddagger$ | $0.2412 \pm 0.0031$ | $0.2355 \pm 0.0006$ |
| MLP$^\dagger$ | $0.2383 \pm 0.0032$ | $0.2327 \pm 0.0009$ |
| XGBoost | $0.2419 \pm 0.0012$ | $0.2416 \pm 0.0007$ |
| LightGBM | $0.2468 \pm 0.0009$ | $0.2467 \pm 0.0002$ |
| CatBoost | $0.2482 \pm 0.0034$ | $0.2473 \pm 0.0016$ |
| TabR | $0.2820 \pm 0.0323$ | $0.2603 \pm 0.0048$ |
| TabR$^\ddagger$ | $0.2542 \pm 0.0101$ | – |
| MNCA | $0.2593 \pm 0.0053$ | $0.2520 \pm 0.0032$ |
| MNCA$^\ddagger$ | $0.2448 \pm 0.0039$ | $0.2404 \pm 0.0025$ |
| TabM$^\spadesuit$ | $0.2469 \pm 0.0035$ | $0.2440 \pm 0.0026$ |
| TabM | $0.2439 \pm 0.0021$ | $0.2428 \pm 0.0006$ |
| TabM[G] | $0.2436 \pm 0.0027$ | – |
| TabM$_{\text{mini}}$ | $0.2433 \pm 0.0017$ | $0.2422 \pm 0.0004$ |
| TabM$^\dagger_{\text{mini}}$ | $0.2334 \pm 0.0018$ | $0.2324 \pm 0.0009$ |

| ecom-offers ↑ | | |
| --- | --- | --- |
| Method | Single model | Ensemble |
| MLP | $0.5989 \pm 0.0017$ | $0.5995 \pm 0.0011$ |
| TabPFN | – | – |
| ResNet | – | – |
| DCN2 | $0.5996 \pm 0.0043$ | $0.6039 \pm 0.0028$ |
| SNN | $0.5912 \pm 0.0056$ | $0.5961 \pm 0.0033$ |
| Trompt | $0.5803 \pm nan$ | – |
| AutoInt | – | – |
| MLP−Mixer | – | – |
| Excel* | $0.5759 \pm 0.0066$ | $0.5759 \pm nan$ |
| SAINT | $0.5812 \pm 0.0098$ | – |
| FT−T | $0.5775 \pm 0.0063$ | $0.5817 \pm 0.0021$ |
| T2G | $0.5791 \pm 0.0056$ | $0.5824 \pm nan$ |
| MLP$^{\ddagger-\text{lite}}$ | $0.5800 \pm 0.0029$ | $0.5819 \pm 0.0011$ |
| MLP$^\ddagger$ | $0.5846 \pm 0.0048$ | $0.5872 \pm 0.0018$ |
| MLP$^\dagger$ | $0.5949 \pm 0.0013$ | $0.5953 \pm 0.0006$ |
| XGBoost | $0.5763 \pm 0.0072$ | $0.5917 \pm 0.0035$ |
| LightGBM | $0.5758 \pm 0.0006$ | $0.5758 \pm 0.0003$ |
| CatBoost | $0.5596 \pm 0.0068$ | $0.5067 \pm 0.0011$ |
| TabR | $0.5943 \pm 0.0019$ | $0.5977 \pm 0.0009$ |
| TabR$^\ddagger$ | $0.5762 \pm 0.0052$ | – |
| MNCA | $0.5765 \pm 0.0087$ | $0.5820 \pm 0.0047$ |
| MNCA$^\ddagger$ | $0.5758 \pm 0.0050$ | $0.5796 \pm 0.0009$ |
| TabM$^\spadesuit$ | $0.5948 \pm 0.0006$ | $0.5952 \pm 0.0004$ |
| TabM | $0.5941 \pm 0.0003$ | $0.5941 \pm 0.0000$ |
| TabM[G] | $0.5970 \pm 0.0010$ | – |
| TabM$_{\text{mini}}$ | $0.5942 \pm 0.0003$ | $0.5943 \pm 0.0001$ |
| TabM$^\dagger_{\text{mini}}$ | $0.5910 \pm 0.0012$ | $0.5913 \pm 0.0002$ |

| maps-routing ↓ | | |
| --- | --- | --- |
| Method | Single model | Ensemble |
| MLP | $0.1625 \pm 0.0001$ | $0.1621 \pm 0.0000$ |
| TabPFN | – | – |
| ResNet | – | – |
| DCN2 | $0.1656 \pm 0.0004$ | $0.1636 \pm 0.0001$ |
| SNN | $0.1634 \pm 0.0002$ | $0.1625 \pm 0.0000$ |
| Trompt | $0.1624 \pm nan$ | – |
| AutoInt | – | – |
| MLP−Mixer | – | – |
| Excel* | $0.1628 \pm 0.0001$ | $0.1621 \pm nan$ |
| SAINT | $0.1634 \pm nan$ | – |
| FT−T | $0.1625 \pm 0.0003$ | $0.1619 \pm 0.0001$ |
| T2G | $0.1616 \pm 0.0001$ | $0.1608 \pm nan$ |
| MLP$^{\ddagger-\text{lite}}$ | $0.1618 \pm 0.0002$ | $0.1613 \pm 0.0000$ |
| MLP$^\ddagger$ | $0.1618 \pm 0.0002$ | $0.1613 \pm 0.0001$ |
| MLP$^\dagger$ | $0.1620 \pm 0.0002$ | $0.1614 \pm 0.0000$ |
| XGBoost | $0.1616 \pm 0.0001$ | $0.1614 \pm 0.0000$ |
| LightGBM | $0.1618 \pm 0.0000$ | $0.1616 \pm 0.0000$ |
| CatBoost | $0.1619 \pm 0.0001$ | $0.1615 \pm 0.0000$ |
| TabR | $0.1639 \pm 0.0003$ | $0.1622 \pm 0.0002$ |
| TabR$^\ddagger$ | $0.1622 \pm 0.0002$ | – |
| MNCA | $0.1625 \pm 0.0001$ | $0.1621 \pm 0.0001$ |
| MNCA$^\ddagger$ | $0.1627 \pm 0.0002$ | $0.1623 \pm 0.0001$ |
| TabM$^\spadesuit$ | $0.1612 \pm 0.0001$ | $0.1609 \pm 0.0000$ |
| TabM | $0.1612 \pm 0.0001$ | $0.1610 \pm 0.0001$ |
| TabM[G] | $0.1611 \pm 0.0001$ | – |
| TabM$_{\text{mini}}$ | $0.1612 \pm 0.0001$ | $0.1610 \pm 0.0000$ |
| TabM$^\dagger_{\text{mini}}$ | $0.1610 \pm 0.0001$ | $0.1609 \pm 0.0000$ |

| homesite-insurance ↑ | | |
| --- | --- | --- |
| Method | Single model | Ensemble |
| MLP | $0.9506 \pm 0.0005$ | $0.9514 \pm 0.0001$ |
| TabPFN | – | – |
| ResNet | – | – |
| DCN2 | $0.9398 \pm 0.0053$ | $0.9432 \pm 0.0018$ |
| SNN | $0.9473 \pm 0.0013$ | $0.9484 \pm 0.0007$ |
| Trompt | $0.9588 \pm nan$ | – |
| AutoInt | – | – |
| MLP−Mixer | – | – |
| Excel* | $0.9622 \pm 0.0004$ | $0.9635 \pm nan$ |
| SAINT | $0.9613 \pm nan$ | – |
| FT−T | $0.9622 \pm 0.0006$ | $0.9633 \pm 0.0001$ |
| T2G | $0.9624 \pm 0.0006$ | $0.9637 \pm nan$ |
| MLP$^{\ddagger-\text{lite}}$ | $0.9609 \pm 0.0009$ | $0.9626 \pm 0.0003$ |
| MLP$^\ddagger$ | $0.9617 \pm 0.0004$ | $0.9630 \pm 0.0002$ |
| MLP$^\dagger$ | $0.9582 \pm 0.0014$ | $0.9599 \pm 0.0002$ |
| XGBoost | $0.9601 \pm 0.0002$ | $0.9602 \pm 0.0000$ |
| LightGBM | $0.9603 \pm 0.0002$ | $0.9604 \pm 0.0001$ |
| CatBoost | $0.9606 \pm 0.0003$ | $0.9609 \pm 0.0001$ |
| TabR | $0.9487 \pm 0.0014$ | $0.9505 \pm 0.0001$ |
| TabR$^\ddagger$ | $0.9556 \pm 0.0021$ | – |
| MNCA | $0.9514 \pm 0.0038$ | $0.9522 \pm 0.0027$ |
| MNCA$^\ddagger$ | $0.9620 \pm 0.0006$ | $0.9635 \pm 0.0002$ |
| TabM$^\spadesuit$ | $0.9641 \pm 0.0004$ | $0.9644 \pm 0.0003$ |
| TabM | $0.9640 \pm 0.0002$ | $0.9642 \pm 0.0001$ |
| TabM[G] | $0.9641 \pm 0.0003$ | – |
| TabM$_{\text{mini}}$ | $0.9643 \pm 0.0003$ | $0.9645 \pm 0.0001$ |
| TabM$^\dagger_{\text{mini}}$ | $0.9631 \pm 0.0003$ | $0.9634 \pm 0.0001$ |

| cooking-time ↓ | | |
| --- | --- | --- |
| Method | Single model | Ensemble |
| MLP | $0.4828 \pm 0.0002$ | $0.4822 \pm 0.0000$ |
| TabPFN | – | – |
| ResNet | – | – |
| DCN2 | $0.4834 \pm 0.0003$ | $0.4822 \pm 0.0001$ |
| SNN | $0.4835 \pm 0.0006$ | $0.4818 \pm 0.0002$ |
| Trompt | $0.4809 \pm nan$ | – |
| AutoInt | – | – |
| MLP−Mixer | – | – |
| Excel$^*$ | $0.4821 \pm 0.0005$ | $0.4808 \pm nan$ |
| SAINT | $0.4840 \pm nan$ | – |
| FT−T | $0.4820 \pm 0.0008$ | $0.4813 \pm 0.0005$ |
| T2G | $0.4809 \pm 0.0008$ | $0.4797 \pm nan$ |
| MLP$^{\ddagger-\text{lite}}$ | $0.4811 \pm 0.0004$ | $0.4805 \pm 0.0001$ |
| MLP$^{\ddagger}$ | $0.4809 \pm 0.0006$ | $0.4804 \pm 0.0003$ |
| MLP$^{\dagger}$ | $0.4812 \pm 0.0004$ | $0.4807 \pm 0.0002$ |
| XGBoost | $0.4823 \pm 0.0001$ | $0.4821 \pm 0.0000$ |
| LightGBM | $0.4826 \pm 0.0001$ | $0.4825 \pm 0.0001$ |
| CatBoost | $0.4823 \pm 0.0001$ | $0.4820 \pm 0.0001$ |
| TabR | $0.4828 \pm 0.0008$ | $0.4814 \pm 0.0004$ |
| TabR$^{\ddagger}$ | $0.4818 \pm 0.0006$ | – |
| MNCA | $0.4825 \pm 0.0004$ | $0.4819 \pm 0.0003$ |
| MNCA$^{\ddagger}$ | $0.4818 \pm 0.0005$ | $0.4809 \pm 0.0003$ |
| TabM$^{\spadesuit}$ | $0.4803 \pm 0.0006$ | $0.4797 \pm 0.0003$ |
| TabM | $0.4804 \pm 0.0002$ | $0.4802 \pm 0.0000$ |
| TabM[G] | $0.4800 \pm 0.0002$ | – |
| TabM$_{\text{mini}}$ | $0.4803 \pm 0.0001$ | $0.4801 \pm 0.0001$ |
| TabM$^{\dagger}_{\text{mini}}$ | $0.4804 \pm 0.0001$ | $0.4803 \pm 0.0000$ |

| homecredit-default ↑ | | |
| --- | --- | --- |
| Method | Single model | Ensemble |
| MLP | $0.8538 \pm 0.0014$ | $0.8566 \pm 0.0005$ |
| TabPFN | – | – |
| ResNet | – | – |
| DCN2 | $0.8471 \pm 0.0019$ | $0.8549 \pm 0.0002$ |
| SNN | $0.8541 \pm 0.0016$ | $0.8569 \pm 0.0010$ |
| Trompt | $0.8355 \pm nan$ | – |
| AutoInt | – | – |
| MLP−Mixer | – | – |
| Excel$^*$ | $0.8513 \pm 0.0024$ | $0.8564 \pm nan$ |
| SAINT | $0.8377 \pm nan$ | – |
| FT−T | $0.8571 \pm 0.0023$ | $0.8611 \pm 0.0013$ |
| T2G | $0.8597 \pm 0.0007$ | $0.8629 \pm nan$ |
| MLP$^{\ddagger-\text{lite}}$ | $0.8598 \pm 0.0009$ | $0.8607 \pm 0.0003$ |
| MLP$^{\ddagger}$ | $0.8572 \pm 0.0011$ | $0.8590 \pm 0.0003$ |
| MLP$^{\dagger}$ | $0.8568 \pm 0.0039$ | $0.8614 \pm 0.0014$ |
| XGBoost | $0.8670 \pm 0.0005$ | $0.8674 \pm 0.0001$ |
| LightGBM | $0.8664 \pm 0.0004$ | $0.8667 \pm 0.0000$ |
| CatBoost | $0.8627 \pm nan$ | – |
| TabR | $0.8501 \pm 0.0027$ | $0.8548 \pm 0.0003$ |
| TabR$^{\ddagger}$ | $0.8547 \pm 0.0021$ | – |
| MNCA | $0.8531 \pm 0.0018$ | $0.8569 \pm 0.0004$ |
| MNCA$^{\ddagger}$ | $0.8544 \pm 0.0033$ | $0.8606 \pm 0.0024$ |
| TabM$^{\spadesuit}$ | $0.8583 \pm 0.0010$ | $0.8599 \pm 0.0006$ |
| TabM | $0.8599 \pm 0.0010$ | $0.8607 \pm 0.0002$ |
| TabM[G] | $0.8588 \pm 0.0013$ | – |
| TabM$_{\text{mini}}$ | $0.8605 \pm 0.0010$ | $0.8614 \pm 0.0007$ |
| TabM$^{\dagger}_{\text{mini}}$ | $0.8635 \pm 0.0008$ | $0.8646 \pm 0.0004$ |

| delivery-eta ↓ | | |
| --- | --- | --- |
| Method | Single model | Ensemble |
| MLP | $0.5493 \pm 0.0007$ | $0.5478 \pm 0.0006$ |
| TabPFN | – | – |
| ResNet | – | – |
| DCN2 | $0.5516 \pm 0.0014$ | $0.5495 \pm 0.0004$ |
| SNN | $0.5495 \pm 0.0008$ | $0.5479 \pm 0.0001$ |
| Trompt | $0.5519 \pm nan$ | – |
| AutoInt | – | – |
| MLP−Mixer | – | – |
| Excel$^*$ | $0.5552 \pm 0.0030$ | $0.5524 \pm nan$ |
| SAINT | $0.5528 \pm nan$ | – |
| FT−T | $0.5542 \pm 0.0026$ | $0.5523 \pm 0.0018$ |
| T2G | $0.5527 \pm 0.0016$ | $0.5512 \pm nan$ |
| MLP$^{\ddagger-\text{lite}}$ | $0.5521 \pm 0.0014$ | $0.5512 \pm 0.0005$ |
| MLP$^{\ddagger}$ | $0.5535 \pm 0.0019$ | $0.5526 \pm 0.0009$ |
| MLP$^{\dagger}$ | $0.5521 \pm 0.0019$ | $0.5511 \pm 0.0007$ |
| XGBoost | $0.5468 \pm 0.0002$ | $0.5463 \pm 0.0001$ |
| LightGBM | $0.5468 \pm 0.0001$ | $0.5465 \pm 0.0000$ |
| CatBoost | $0.5465 \pm 0.0001$ | $0.5461 \pm 0.0000$ |
| TabR | $0.5514 \pm 0.0024$ | $0.5480 \pm 0.0005$ |
| TabR$^{\ddagger}$ | $0.5520 \pm 0.0015$ | – |
| MNCA | $0.5498 \pm 0.0007$ | $0.5488 \pm 0.0002$ |
| MNCA$^{\ddagger}$ | $0.5507 \pm 0.0013$ | $0.5494 \pm 0.0006$ |
| TabM$^{\spadesuit}$ | $0.5510 \pm 0.0015$ | $0.5504 \pm 0.0004$ |
| TabM | $0.5494 \pm 0.0004$ | $0.5492 \pm 0.0001$ |
| TabM[G] | $0.5509 \pm 0.0003$ | – |
| TabM$_{\text{mini}}$ | $0.5497 \pm 0.0007$ | $0.5495 \pm 0.0003$ |
| TabM$^{\dagger}_{\text{mini}}$ | $0.5510 \pm 0.0019$ | $0.5502 \pm 0.0000$ |

| weather ↓ | | |
| --- | --- | --- |
| Method | Single model | Ensemble |
| MLP | $1.5378 \pm 0.0054$ | $1.5111 \pm 0.0029$ |
| TabPFN | – | – |
| ResNet | – | – |
| DCN2 | $1.5606 \pm 0.0057$ | $1.5292 \pm 0.0028$ |
| SNN | $1.5280 \pm 0.0085$ | $1.5013 \pm 0.0034$ |
| Trompt | $1.5187 \pm nan$ | – |
| AutoInt | – | – |
| MLP−Mixer | – | – |
| Excel$^*$ | $1.5131 \pm 0.0022$ | $1.4707 \pm nan$ |
| SAINT | $1.5097 \pm 0.0045$ | – |
| FT−T | $1.5104 \pm 0.0097$ | $1.4719 \pm 0.0040$ |
| T2G | $1.4849 \pm 0.0087$ | $1.4513 \pm nan$ |
| MLP$^{\ddagger-\text{lite}}$ | $1.5170 \pm 0.0040$ | $1.4953 \pm 0.0023$ |
| MLP$^{\ddagger}$ | $1.5139 \pm 0.0031$ | $1.4978 \pm 0.0020$ |
| MLP$^{\dagger}$ | $1.5162 \pm 0.0020$ | $1.5066 \pm 0.0008$ |
| XGBoost | $1.4671 \pm 0.0006$ | $1.4629 \pm 0.0002$ |
| LightGBM | $1.4625 \pm 0.0008$ | $1.4581 \pm 0.0003$ |
| CatBoost | $1.4688 \pm 0.0019$ | – |
| TabR | $1.4666 \pm 0.0039$ | $1.4547 \pm 0.0008$ |
| TabR$^{\ddagger}$ | $1.4458 \pm 0.0018$ | – |
| MNCA | $1.5062 \pm 0.0054$ | $1.4822 \pm 0.0013$ |
| MNCA$^{\ddagger}$ | $1.5008 \pm 0.0034$ | $1.4782 \pm 0.0011$ |
| TabM$^{\spadesuit}$ | $1.4786 \pm 0.0039$ | $1.4715 \pm 0.0020$ |
| TabM | $1.4722 \pm 0.0024$ | $1.4675 \pm 0.0009$ |
| TabM[G] | $1.4728 \pm 0.0022$ | – |
| TabM$_{\text{mini}}$ | $1.4716 \pm 0.0016$ | $1.4669 \pm 0.0010$ |
| TabM$^{\dagger}_{\text{mini}}$ | $1.4651 \pm 0.0020$ | $1.4581 \pm 0.0016$ |

