# OpenReview forum: "TabM: Advancing tabular deep learning with parameter-efficient ensembling"
_ICLR.cc/2025/Conference — ICLR 2025 Poster_

### Official Review · Reviewer_aGcY · 2024-11-01

**Soundness:** 3
**Presentation:** 3
**Contribution:** 3
**Rating:** 8
**Confidence:** 4

**Summary:**

The paper applies the BatchEnsemble technique to MLPs for tabular data, and investigates several modifications. The results show an improvement over MLPs and several deep baseline models and GBRT on a broach benchmark of 50 datasets from the literature. The experiments show that in the first adapter layer in particular is extremely critical, and results in the majority of gains.

**Strengths:**

The paper discusses the adoption of the BatchNorm architecture for the tabular setting, and describes several interesting ablations.
The empirical evaluation is broad and in-depth and well presented.
The empirical results are quite strong against a broad variety of baselines.
The paper is well written.

**Weaknesses:**

- The technical novelty of the paper is somewhat small; however, this is made up for by the in-depth analysis with somewhat surprising results and the good performance of the proposed model.
- The paper doesn't describe the relationship to dropout, even though dropout was original describes as an efficient ensemble approach. Given the simple nature of TabM_mini, there seem to be some obvious parallels between drop-out and TabM_mini that I think are worth discussing. In particular, TabM_mini actually contains a dropout layer, and it would be interesting to evaluate how the two forms of ensembling complement each other.
- The comparison doesn't include TabPFN, an extremely strong baseline. McElfresh showed that even subsampling data to at most 3000 datapoints, TabPFN outperforms most other models.
- Given the focus on improving MLP performance, a comparison with regularization cocktails and "Better by Default: Strong Pre-Tuned MLPs and Boosted Trees on Tabular Data" might be relevant but not essential.
- Given the somewhat surprising result about the importance of the initial adapter, it would be great to have more ablations on the different components. In particular, it's unclear in how far the initial adapter and last layer are coupled. How does the performance change if the best performing last layer is used for all the initial adapters? I.e. are the initial layer and last layer co-adapted or do they work independently?

## Minor comments
Figure 2 cuts off points at 8%. It seems there's a lot of points on exactly that line, which seems a bit suspicious. Are these all the outliers that are clipped to the limits of the figure? It would be great to indicate outliers in the plot.

Table 2 is far below the mention and maybe should be moved up.

Table 2, MLP on the Maps dataset is missing units in duration.

The title of section 5.3 should read "How does the performance of TabM depend on k?"

Line 257: "requires a special care" should be "requires special care"

Line 160: multiplication with 100% is not mathematically meaningful.

**Questions:**

- Are number of layers for MLP and TabM tuned as hyper-parameters? What are typical values for each?

- Why is k not included in the hyper-parameter search for TabM?

- How do you ensure diversity for the MLPs in the ensemble in Figure 6?

- How do you initialize the last layers in TabM to ensure diversity?

- Is it possible that TabM_mini outperforms the real ensemble because of more diversity in the initialization?

- Is it correct that when doing the full ensemble, adding an adapter into each model would lead to an equivalent ensemble? I was wondering why this ablation was not performed, but I assume random initialization of weights for each member of the ensemble makes the adapters irrelevant.

---

> ### Author Response · Authors · 2024-11-21
> **Reply (Part 1/2)**
>
> We thank the reviewer for the extensive feedback.
>
> *Two quick comments before we start*:
>
> - There are many questions in the review, so our reply takes multiple posts.
> - (minor) It seems that some of the weaknesses are closer to questions in spirit.
>
> > The technical novelty of the paper is somewhat small
>
> Please, see the "Big picture" part of the global post. While our method is easy to implement (which we see as its strength), it has been missing for many years, and our main focus in this work is to bring the significant missing value to the field of tabular DL. And this still required a non-trivial exploration of technical ideas (see the details below the "Big picture" part of the global post).
>
> > The paper doesn't describe the relationship to dropout, even though dropout was original describes as an efficient ensemble approach. Given the simple nature of TabM_mini, there seem to be some obvious parallels between drop-out and TabM_mini that I think are worth discussing.
>
> We should admit that we do not fully understand the question. A network trained with dropout can indeed be viewed as an efficient ensemble, known as "dropout ensemble", and for example is already mentioned in Section A2. However, there are many other ensembling methods, and it is unclear, why dropout should be highlighted separately, how this is connected to the simplicity of TabM-mini, and what is implied by the "obvious parallels". We are open for discussion here, but currently, we are uncertain how to interpret the request.
>
> > In particular, TabM_mini actually contains a dropout layer, and it would be interesting to evaluate how the two forms of ensembling complement each other.
>
> To answer the question, we evaluate MLP and TabM-mini of the same depth 3 and width 512 with and without dropout. Plus, we include $\mathrm{MLP}^{\times 32}$ -- a traditional deep ensemble of 32 MLPs without dropout as an additional baseline. The table below shows the ranks and relative improvements over MLP with dropout on 18 datasets.
>
> | Name                                   | Rank ↓       | Relative improvement over MLP w/o dropout ↑ |
> | -------------------------------------- | ------------ | ------------------------------------------- |
> | $\mathrm{MLP}$ w/o dropout             | $3.89 \pm 0.9$  | $0.0$% $\pm$ $0.0$%                                  |
> | $\mathrm{MLP}$                         | $3.11 \pm 0.9$  | $0.3$% $\pm$ $1.52$%                                 |
> | $\mathrm{MLP}^{\times 32}$ w/o dropout | $2.17 \pm 0.79$ | $1.53$% $\pm$ $1.55$%                                |
> | $\mathrm{TabM_{mini}}$ w/o dropout     | $1.83 \pm 0.86$ | $1.69$% $\pm$ $1.80$%                               |
> | $\mathrm{TabM_{mini}}$                 | $1.67 \pm 1.08$ | $1.75$% $\pm$ $1.97$%                                |
>
> Conclusion: the deep ensembling and "mini" ensembling are significantly more impactful than dropout.
>
> > The comparison doesn't include TabPFN, an extremely strong baseline. McElfresh showed that even subsampling data to at most 3000 datapoints, TabPFN outperforms most other models.
>
> The original submission *already includes* TabPFN, and with even larger number of subsampled objects (10K) than suggested (3K). However, it is included only in Appendix F, because:
> - TabPFN is not competitive on our benchmark, which is expected, because we experiment on datasets *orders of magnitude larger* (up to millions of objects) than the target scope of TabPFN.
> - TabPFN has too many limitations for our benchmark, which prevents a proper evaluation (not applicable to regression tasks, not applicable to more than 100 features, not applicable to large datasets).
>
> > Given the focus on improving MLP performance, a comparison with regularization cocktails and "Better by Default: Strong Pre-Tuned MLPs and Boosted Trees on Tabular Data" might be relevant but not essential.
>
> Both papers are already cited in the original submission, because indeed, they are tangentially relevant and can serve as an indirect inspiration, showing that improving MLPs is generally possible. However, our scope is specifically *architectures* (for example, this is communicated at the beginning of the abstract). So we do not compare with non-architectural techniques that are actively used in the suggested papers.

---

> ### Author Response · Authors · 2024-11-21
> **Reply (Part 2/2)**
>
> > In particular, it's unclear in how far the initial adapter and last layer are coupled. How does the performance change if the best performing last layer is used for all the initial adapters? I.e. are the initial layer and last layer co-adapted or do they work independently?
>
> Perhaps, the best *visual* answer is the new illustration of the model in Figure 9.
>
> First, we quickly remind that TabM is just an efficient parametrization of $k$ MLPs, so it can be explicitly materialized as $k$ separate MLPs. However, due to the almost complete weight sharing, it is possible to pack all $k$ MLPs in one model (which gives TabM). At the same time, mathematically, the forward passes of the $k$ implicit MLPs are still independent (while in practice, they are efficiently performed in parallel).
>
> Now, regarding the question. The $i$-th adapter is strictly coupled with the $i$-th prediction head, because both of them belong only to the $i$-th ensemble member, and not to any other member. So randomly combining $i$-th adapter with $j$-th prediction head ($i \ne j$) is the same as swapping layers of different models, i.e. this is not expected to work "correctly" without any additional assumptions.
>
> > Are these all the outliers that are clipped to the limits of the figure?
>
> Yes, this is correct. We clarified this in the caption in the new revision.
>
> > Table 2, MLP on the Maps dataset is missing units in duration.
>
> This is now fixed, thanks! It is 10 minutes.
>
> > The title of section 5.3 should read "How does the performance of TabM depend on k?"
> > ...
> > Line 257: "requires a special care" should be "requires special care"
>
> This is now also fixed, thanks for pointing to this!
>
> > Are number of layers for MLP and TabM tuned as hyper-parameters? What are typical values for each?
>
> Yes, they are tuned, as specified in "Implementation details" in Appendix. The allowed values are from 1 to 6 for MLP and from 1 to 5 for TabM.
>
> > Why is k not included in the hyper-parameter search for TabM?
>
> $k$ implicitly affects the effective batch size by cloning the input $k$ times. Tuning batch size and learning rate simultaneously is usually suboptimal, because the optimal values of these hyperparameters are coupled in a non-trivial way, which makes the job harder for TPE-like hyperparameter tuning algorithms. So we fix both $k$ and batch size, and tune only the learning rate.
>
> > How do you ensure diversity for the MLPs in the ensemble in Figure 6?
>
> By using different random seeds, which affects the random initialization, training batch sequences, dropout activations, etc. In general, for deep ensembles, it is known that different initializations are already enough to obtain meaningfully diverse models.
>
> > How do you initialize the last layers in TabM to ensure diversity?
>
> For TabM, the diversity comes only from the random initialization of the non-shared weights, including the prediction heads. There are no additional tricks aimed at increasing diversity. To clarify, our approach is in line prior work on parameter-efficient ensembles.
>
> > Is it possible that TabM_mini outperforms the real ensemble because of more diversity in the initialization?
>
> In TabM-mini, the $k$ implicit ensemble members share almost all of their weights, so by construction, the initialization of the $k$ models in TabM is *less* diverse than the initialization of $k$ fully independent models.
>
> > Is it correct that when doing the full ensemble, adding an adapter into each model would lead to an equivalent ensemble?
>
> Yes, this is correct.
>
> > I assume random initialization of weights for each member of the ensemble makes the adapters irrelevant.
>
> Yes, our reasoning was that the adapters in the beginning randomly initialized with $\pm 1$ do not make a difference for plain MLPs.

---

> ### Author Response · Authors · 2024-11-25
> **Invitation to discussion**
>
> Dear reviewer, we will be glad to hear if our rebuttal addresses the concerns. We performed the dropout-related experiments and answered all other questions.

---

> > ### Comment · Reviewer_aGcY · 2024-11-26
> >
> > Thank you for your response. I appreciate you running additional ablations regarding drop-out, I think these are very interesting and show the value of your work.
> >
> > To elaborate on the reason why I thought it was particularly relevant, in your summary you say:
> > > Parameter-efficient ensembles have been known for many years, yet there has been no successful adaptation of this paradigm for tabular DL. Our study identifies and closes this gap.
> >
> > However, drop-out is a parameter efficient ensemble that is widely used in tabular data, and that you in fact also use. Given that it is the only parameter efficient ensemble method that is widely used, it seemed to me to deserve a call-out. I think your additional results alleviate any hesitation about the relative contributions.
> >
> > I also appreciate you pointing out the results with TabPFN that are presented in the appendix. The results make sense in light of the dataset sizes you study.
> >
> > Regarding co-adaptation of adapter an classification head:
> >
> > > So randomly combining i-th adapter with j-th prediction head is the same as swapping layers of different models, i.e. this is not expected to work "correctly" without any additional assumptions.
> >
> > I understand the architecture; my question was whether this expectation actually holds true empirically or not, given the strong weight sharing. If you think of the models as completely distinct, certainly it should not work, but they are mostly not distinct.
> > I was hoping this quick experiment would add to my (and maybe other readers') understanding of the method, but I would not insist on it on the very short rebuttal timeline.
> >
> > After reading the remaining responses and the additional results provided by the authors, I will update my score to "accept".

---

### Official Review · Reviewer_For5 · 2024-11-02

**Soundness:** 2
**Presentation:** 3
**Contribution:** 2
**Rating:** 6
**Confidence:** 3

**Summary:**

This paper proposes TabM, a deep learning model for tabular data based on MLP enhanced with BatchEnsemble, along with custom modifications designed to improve the model's efficiency and performance. The authors position TabM as an efficient alternative to more complex architectures like Transformer-based models, claiming it achieves superior performance without their added computational cost. The paper also provides a detailed analysis, suggesting that parameter-efficient ensembling can effectively address overfitting and improve optimization dynamics in tabular MLPs.

**Strengths:**

1. The paper is well-structured and clearly written, making the methods, results, and analysis easy to understand.
2. TabM demonstrates competitive performance, positioning it as an efficient alternative for tabular data modeling.
3. By introducing BatchEnsemble techniques into tabular data modeling, the authors have adapted and enhanced a previously underutilized approach in this domain.
4. The paper provides a thorough analysis of each component's impact on model performance, contributing valuable insights.

**Weaknesses:**

1. Although the authors mention similar methods, a more detailed comparison with other related approaches for tabular data, such as Trompt, would be helpful. Despite structural differences, both methods share commonalities in prediction, such as averaging multiple head (cycle) outputs and summing losses across heads (cycles).
2. Comparisons with standard ensemble methods are insufficient. Incorporating results from methods like model soup could help position TabM's performance.
3. Dataset scope is somewhat limited, particularly for classification tasks. For a more comprehensive evaluation, the authors might consider incorporating the datasets from Tabzilla, which could provide richer classification benchmarks and enable a more detailed analysis, such as comparing performance across binary, multiclass, and regression tasks or evaluating model performance on datasets of varying sizes.

[1] Kuan-Yu Chen, Ping-Han Chiang, Hsin-Rung Chou, Ting-Wei Chen, Tien-Hao Chang: Trompt: Towards a Better Deep Neural Network for Tabular Data. In ICML

[2] Duncan C. McElfresh, Sujay Khandagale, Jonathan Valverde, Vishak Prasad C., Ganesh Ramakrishnan, Micah Goldblum, Colin White: When Do Neural Nets Outperform Boosted Trees on Tabular Data? In NeurIPS

**Questions:**

1. In Table 2 (p.10), the last column of the MLP row lists "10." Could the authors confirm if it is measured in minutes, as there’s no unit mentioned?
2. In line 1027, the authors state, *For numerical features, by default, we used a slightly modified version of the quantile normalization from the Scikit-learn package, with rare exceptions when it turned out to be detrimental.* Could the authors clarify how they define “detrimental” in this context and provide further explanation?
3. Could the authors provide more detailed diversity metrics for the predictions of the 32 base models?
4. Line 431 notes, *Interestingly, the best prediction head of TabM performs no better than the plain MLP.* Would an ensemble of 32 independently and undertrained MLPs yield similarly strong performance?
5. The ranking method used is unconventional. Is this ranking method novel to this paper? Could the authors also provide results for a more traditional average rank metric to facilitate comparison, as it would be interesting to see how these two measures differ?
6. One advantage of using deep learning for tabular data is the ability to generate embeddings that are useful for downstream tasks and multimodal learning. How does TabM determine the representation of each sample? Is it based on concatenation, averaging across base models, or some other method? Additionally, could the authors provide a comparison of TabM’s embeddings with those from other deep learning approaches on specific tasks to showcase their effectiveness?

---
Lastly, if the authors can address the concerns noted in the weaknesses and questions, I will consider raising my score accordingly.

---

> ### Author Response · Authors · 2024-11-21
> **Reply (Part 1/3)**
>
> We thank the reviewer for the detailed feedback. The strengths mentioned in the review resonate a lot with what we try to achieve with our project. So we did our best to address the remaining concerns below.
>
> > more detailed comparison with other related approaches for tabular data, such as Trompt
>
> Trompt is already cited (Section 2) and evaluated in our submission in Appendix F. However, it has multiple problems, so it is presented only in Appendix F and is not included in the main text:
> - There is no official implementation of Trompt.
> - The performance is not competitive, especially given the following point.
> - Trompt is *extremely* slow to train. Hyperparameter tuning is impossible beyond small datasets.
>
> > Comparisons with standard ensemble methods are insufficient. Incorporating results from methods like model soup could help position TabM's performance.
>
> - The standard ensemble is already evaluated as $\text{MLP}^{\times 32}$ in Section 5.
> - Section A2 already discusses why we start exactly from BatchEnsemble and not from any other ensembling method. In short, BatchEnsemble is the most architecture-agnostic, practical, and easy-to-use option among all available methods, so it is a reasonable choice for our project. Plus, full-fledged comparisons of ensemble methods across many domains are already performed in dedicated studies.
> - The suggested "model soup" is not an ensemble method, it is a weight averaging method: "Model soups: **averaging weights** of multiple fine-tuned models improves accuracy without increasing inference time". It means that it produces *one* model making *one* prediction without additional inference costs. By contrast, TabM imitates an ensemble of models producing *multiple* predictions, and it has additional costs compared to plain single models.
>
> > Dataset scope is somewhat limited
>
> We share the reviewer's enthusiasm for better benchmarks. However, it seems that our benchmark takes a step forward compared to prior studies. Our perspective is as follows:
>
> - For example, let's consider large datasets. Our study already covers more datasets with >1M objects than **15 out of the 19** papers on GBDT and tabular DL cited in the "Related work" section. The paper on LightGBM is *the only* paper that uses more (5) than our paper (3). The papers on XGBoost, AutoInt, and NODE use 3 large datasets, and all other papers use less, with many papers using **zero** large datasets.
> - Let's consider datasets with a large number of features. Our study already covers more datasets with >100 features than **19 out of 19** papers on GBDT and tabular DL cited in the "Related work" section.
> - Furthermore, we are one of the first studies integrating the modern challenging TabReD benchmark `[1]`, with more objects and features than in many prior benchmarks, and with real-world time-based splits. Please, see the details in the "Datasets" paragraph of Section 3.1, and the corresponding results on the right side of Figure 4.
> - Limitations: our study does not cover the niches of small datasets (intuitively, <2K objects) and datasets with a high `n_features / n_objects` ratio. These niches can be different in terms of typical baselines, hyperparameter grids, and model ranking, and can require a different experiment setup (e.g. our train-val-test split and our hyperparameter tuning protocol may need changes for small datasets).
>
> > particularly for classification tasks
>
> With the benchmarks from prior work that we used, regression tasks indeed happened to be more presented. However, given the below evaluation on TabZilla, it seems that our results generalize well on unseen classification tasks.
>
> (TabZilla is covered in the next post)

---

> ### Author Response · Authors · 2024-11-21
> **Reply (Part 2/3)**
>
> > consider incorporating the datasets from Tabzilla
>
> *(TL;DR)* The results on 15 datasets from TabZilla are in line with our paper:
>
> | Model                          | Mean Rank ($\downarrow$) | Relative improvement over MLP ($\uparrow$) |
> | ------------------------------ | ------------------------ | -------------------------------------- |
> | $\mathrm{FT\text{-}T}$         | $3.07 \pm 1.27$          | $0.07$% $\pm$ $2.08$%                  |
> | $\mathrm{MLP}$                 | $3.00 \pm 1.73$          | $0.00$% $\pm$ $0.00$%                    |
> | $\mathrm{MNCA}$                | $2.07 \pm0.70$           | $1.86$% $\pm$ $3.81$%                    |
> | $\mathrm{TabM}$                | $1.87 \pm 1.30$          | $2.14$% $\pm$ $2.55$%                    |
> | $\mathrm{XGBoost}$             | $1.80 \pm 1.21$          | $2.45$% $\pm$ $3.39$%                    |
> | $\mathrm{TabM}^\dagger_{mini}$ | $1.47 \pm 0.52$          | $2.68$% $\pm$ $3.36$%                    |
>
> *(Details)* After removing small datasets (<2K objects, as mentioned earlier), the datasets already used in our paper, duplicated datasets, and non-tabular datasets (e.g. board games), we are left with 15 out of 36 datasets. The above table is obtained using these 15 datasets.
>
> > In Table 2 (p.10), the last column of the MLP row lists "10." Could the authors confirm if it is measured in minutes, as there’s no unit mentioned?
>
> Yes, this is 10 minutes, and is now fixed in the new revision, thanks!
>
> > Could the authors clarify how they define “detrimental” in this context and provide further explanation?
>
> This part is fully inherited from "TabR: Tabular Deep Learning Meets Nearest Neighbors in 2023", ICLR 2024. But in fact, it applies only to *one* dataset. As a quick test, we evaluated MLP on this one dataset with the normalization turned on, and indeed observed performance degradation from 0.819 to 0.806 (accuracy), which is significant. Again, this part is inherited, but from our quick test, it seems to be reasonable.
>
> > Could the authors provide more detailed diversity metrics for the predictions of the 32 base models?
>
> *TL;DR.* On average over 18 datasets, the diversity of $k = 32$ predictions for $\mathrm{TabM_{mini}}$ is $\times 1.88$ *higher* than for the standard ensemble of $k$ MLPs.
>
> *Technical details.* We define the diversity of $k$ predictions as the mean deviation of the individual predictions from their collective mean prediction (i.e. prediction of an ensemble).
> For regressions, it is simply the standard deviation of predictions. For classification, it is the mean KL divergence between individual predictions and the collective mean prediction:
>
> * $y^k_i$ is the prediction of the $k$-th head/model for the $i$-th sample
> * $\bar{y_i}$ is the prediction of ensemble (deep or TabM)
> * Regressions: $\frac{1}{KN}\sum_{i}^N \sum_{k}^K (y^k_i - \bar{y_i})^2$
> * Classification: $\frac{1}{KN}\sum_{i}^N \sum_{k}^K KL(y^k_i || \bar{y_i})$
>
> > Would an ensemble of 32 independently and undertrained MLPs yield similarly strong performance?
>
> By definition, TabM is an efficient parametrization of 32 MLPs, so in theory, it should be possible to train them separately in a way that makes them no worse than TabM. But how to do that may be a non-trivial question. Below, we evaluate a naive way to "undertrain" MLPs on 18 datasets from our benchmark. To obtain an ensemble of 32 undertrained MLPs, we train each MLP only for a fraction of the optimal number of epochs (defined by the full run with early stopping). The results are provided below and show that the naive undertraining leads to worse performance compared to the standard ensemble of fully trained MLPs. Meanwhile, $\mathrm{TabM_{mini}}$ remains superior. Notation: $\mathrm{MLP[0.25]}$ denotes that MLP is trained for a quarter of the optimal number of epochs.
>
>
> | Name                             | Rank          | Relative improvement to MLP |
> | :------------------------------- | ------------- | --------------------------: |
> | $\mathrm{MLP[0.25]}^{\times 32}$ | $4.94\pm1.26$ |       $-1.12$% $\pm$  $2.13$% |
> | $\mathrm{MLP}$                   | $4.5\pm0.86$  |       $0.0$%  $\pm$    $0.0$% |
> | $\mathrm{MLP[0.5]}^{\times 32}$  | $3.39\pm1.04$ |         $0.46$% $\pm$ $0.9$% |
> | $\mathrm{MLP[0.75]}^{\times 32}$ | $2.5\pm0.86$  |        $0.94$%  $\pm$ $1.08$% |
> | $\mathrm{MLP}^{\times32}$        | $1.94\pm0.64$ |        $1.36$%  $\pm$ $1.46$\% |
> | $\mathrm{TabM_{mini}}$           | $1.56\pm1.2$  |        $2.06$%  $\pm$ $2.16$% |

---

> > ### Author Response · Authors · 2024-11-21
> > **Reply (Part 3/3)**
> >
> > > The ranking method used is unconventional. Is this ranking method novel to this paper?
> >
> > The comment seems to imply that there is *one* ranking method in our paper, but our paper uses *three* ranking methods:
> >
> > 1. The traditional average ranks across all datasets (the left part of Figure 4). They are defined in Section E3 (the text there is improved the new revision). In short, when computing model ranks on individual datasets, the standard deviations of metrics across random seeds are taken into account. The approach *without* taking the standard deviations into account is covered by the last ranking method in this list.
> > 2. The ranking by the mean relative improvements over MLP across all datasets (the middle and right part of Figure 4). The corresponding metric is defined in Section 3.1. For example, Figure 4 allows seeing the *weakest* part of the performance profiles, including failure cases on individual datasets. It shows how reliably models perform on datasets that are "inconvenient" for them. This can be seen from lower quantiles or even from individual data points.
> > 3. The critical difference diagram in Figure 12 -- a ranking method often used in the literature. Contrary to the first type of ranks, the standard deviations within tasks are *not* taken into account.
> >
> > *(Note that, according to the above, the order of models is different across all parts of Figure 4)*
> >
> > > Could the authors also provide results for a more traditional average rank metric to facilitate comparison
> >
> > As mentioned above, this ranking method is already provided.
> >
> > > how these two measures differ?
> >
> > The motivation for having multiple ranking methods is exactly to give multiple perspectives on the field. TabM, simple MLPs, and GBDTs turn out to be the most robust w.r.t. to the ranking method.
> >
> > > One advantage of using deep learning for tabular data is the ability to generate embeddings that are useful for downstream tasks and multimodal learning. How does TabM determine the representation of each sample?
> >
> > This is already discussed in Section A7. In short, this is a natural limitation of TabM inherited from ensembles in general, so indeed, it does *not* implement the usual notion of "object embedding", but instead produces $k$ embeddings. This is unlikely that we can propose a novel workaround here. Among simple strategies, we highlight (1) naive concatenation (perhaps, followed by normalization and/or some sort of dimensionality reduction); (2) distilling TabM into a traditional single model.

---

> > > ### Author Response · Authors · 2024-11-25
> > > **Invitation to discussion**
> > >
> > > Dear reviewer, we did our best to address the concerns expressed in the review. In particular, we:
> > >
> > > - Performed evaluation on the Tabzilla benchmark.
> > > - Computed the diversity of predictions of TabM.
> > > - Conducted the experiment with undertraining.
> > > - Answered other questions.
> > >
> > > We will be glad to hear your feedback and continue the discussion.

---

> > > > ### Comment · Reviewer_For5 · 2024-11-25
> > > >
> > > > After reviewing the authors' rebuttal, I am satisfied with their detailed responses, which addressed my main concerns effectively. I have updated my score to 6.

---

### Official Review · Reviewer_4UNA · 2024-11-04

**Soundness:** 3
**Presentation:** 3
**Contribution:** 2
**Rating:** 6
**Confidence:** 3

**Summary:**

This paper come up with an efficient ensembling method for tabular deep learning — which integrate BatchEnsemble with multilayer perceptron. This method exhibits great improvements in their benchmark.

**Strengths:**

- [Major] This method is simple yet effective.
- [Major] The paper is easy to read.

**Weaknesses:**

- [Medium] The proposed method is not much novel. Note that this concern is not the main reason that I chose to reject this paper. If other concerns listed can be addressed (e.g., more technical contributions, comprehensive empirical study for depper understandings), I think this paper is still very useful.
- [Major] More experiments are needed to fully understand the behavior of TabM. In the abstract, the authors mentioned “we show that parameter-efficient ensembling is not an arbitrary trick, but rather a highly effective way to reduce overfitting and improve optimization dynamics of tabular MLPs.” However, in the paper, there is no discussion why TabM helps with reducing overfitting and optimization dynamics. Of course the experiments proves this, but it seems like a general benefits of ensemble models. At least, some explanation supported by experiments is needed to fully understand this method — especially this method is already very simple and not much novel, understanding this better is important to have sufficient technical contribution.
- [Major] Baselines missing. This method is an ensemble method. At least we could compare it to other naive ensemble method. For instance, how much speed up it provides compared to normal ensemble; how robust it compared to normal ensemble; etc.
- [Major] Datasets missing. Yes this paper test on many datasets — but the more important thing is the diversity. For instance, how about high-dimensional datasets and large datasets (I know it is on Table 2, but only evaluated on two datasets and three baselines, moreover, only TabMmini has been evaluated). I suggest the authors to use TabZilla hard benchmark, and then evaluate the method’s capability under different circumstances. These kind of experiments can make the findings more interesting.

**Questions:**

See weakness.

**Details Of Ethics Concerns:**

No.

---

> ### Author Response · Authors · 2024-11-21
> **Reply (Part 1/2)**
>
> We thank the reviewer for the comments, and we are looking forward to the discussion.
>
> > Contribution: 2 ... The proposed method is not much novel ... this method is already very simple and not much novel ...
>
> Please see the “Big picture” part of our global post. While our method is easy to implement (which we see as its strength), it has been missing for many years, and our main focus in this work is to bring the significant missing value to the field of tabular DL. And this still required a non-trivial exploration of technical ideas (see the details below the "Big picture" part of the global post).
>
> > there is no discussion why TabM helps with reducing overfitting ... it seems like a general benefits of ensemble models.
>
> Yes, in Section 5, we show that the usual mechanics of ensembles work as intended and are highly effective for TabM. We adjusted the wording in the abstract to better align readers' expectations with the content.
>
> > Baselines missing ... compare it to other naive ensemble method
>
> This comparison is already presented in the paper. Namely, the figure in Section 5.1 contains $MLP^{\times 32}$ -- the naive ensemble method.
>
> > For instance, how much speed up it provides compared to normal ensemble
>
> If we compare the training time between TabM with $k = 32$ and single MLP multiplied by $32$, then we get $\times 13.7$ speed-up on average across datasets. As for the inference throughput on CPU, the speed-up is $\times 14$. This is a naive approach, since, in theory, the training of MLPs can be performed in parallel. However, this approach gives the right intuition that TabM is more efficient than naive ensembles, and the overhead is less than $\times k$ compared to single MLP.
>
> > in Table 2 ... only three baselines, moreover, only TabMmini has been evaluated
>
> As explained in Section 4.4, the goal of the experiments reported in Table 2 is to test the ability of various model families to scale to large datasets, which does not require evaluating all models.
>
> > Datasets ... how about high-dimensional datasets and large datasets ...
>
> We share the reviewer's enthusiasm for better benchmarks. However, it seems that our benchmark takes a step forward compared to prior studies. Our perspective is as follows:
>
> - Regarding large datasets, our study already covers more datasets with >1M objects than **15 out of the 19** papers on GBDT and tabular DL cited in the "Related work" section. The paper on LightGBM is the only paper that uses more (five) than our paper (three). The papers on XGBoost, AutoInt, and NODE use three large datasets, and all other papers use less, with many papers using **zero** large datasets.
> - Regarding high-dimensional datasets, our study already covers more datasets with >100 features than **19 out of the 19** papers on GBDT and tabular DL cited in the "Related work" section.
> - Furthermore, we are one of the first studies integrating the modern challenging TabReD benchmark `[1]`, with more objects and features than in many prior benchmarks, and with real-world time-based splits. Please, see the details in the "Datasets" paragraph of Section 3.1, and the corresponding results on the right side of Figure 4.
> - Limitations: our study does not cover the niches of small datasets (intuitively, <2K objects) and datasets with a high `n_features / n_objects` ratio. These niches can be different in terms of typical baselines, hyperparameter grids, and model ranking, and can require a different experiment setup (e.g. our train-val-test split and our hyperparameter tuning protocol may need changes for small datasets).
>
> (TabZilla is discussed in the next post)

---

> ### Author Response · Authors · 2024-11-21
> **Reply (Part 2/2)**
>
> > I suggest the authors to use TabZilla hard benchmark
>
> *(TL;DR)* The results on 15 datasets from TabZilla are in line with our paper:
>
> | Model                          | Mean Rank ($\downarrow$) | Relative improvement over MLP ($\uparrow$) |
> | ------------------------------ | ------------------------ | -------------------------------------- |
> | $\mathrm{FT\text{-}T}$         | $3.07 \pm 1.27$          | $0.07$% $\pm$ $2.08$%                  |
> | $\mathrm{MLP}$                 | $3.00 \pm 1.73$          | $0.00$% $\pm$ $0.00$%                    |
> | $\mathrm{MNCA}$                | $2.07 \pm0.70$           | $1.86$% $\pm$ $3.81$%                    |
> | $\mathrm{TabM}$                | $1.87 \pm 1.30$          | $2.14$% $\pm$ $2.55$%                    |
> | $\mathrm{XGBoost}$             | $1.80 \pm 1.21$          | $2.45$% $\pm$ $3.39$%                    |
> | $\mathrm{TabM}^\dagger_{mini}$ | $1.47 \pm 0.52$          | $2.68$% $\pm$ $3.36$%                    |
>
> *(Details)* After removing small datasets (<2K objects, as mentioned earlier), the datasets already used in our paper, duplicated datasets, and non-tabular datasets (e.g. board games), we are left with 15 out of 36 datasets. The above table is obtained using these 15 datasets.

---

> ### Author Response · Authors · 2024-11-25
> **Invitation to discussion**
>
> Dear reviewer, we are looking forward to discussion. In the rebuttal, we:
> - Performed the evaluation on the Tabzilla benchmark, as requested.
> - Commented on all other concerns, and hopefully addressed them.

---

> ### Comment · Reviewer_4UNA · 2024-11-26
>
> Thank the authors for the thoughtful response. I am willing to increase my score to 6.

---

> > ### Author Response · Authors · 2024-11-28
> > **New revision**
> >
> > We thank the reviewer `4UNA` for taking the time to read our rebuttal.
> >
> > In the original review, the reviewer recommended to extend our analysis on TabM. **The new revision provides the extended analysis**, namely:
> >
> > - We added Section A.2 with *analysis on the sources of TabM's power.*
> > - We improved the text of the *analysis of the gradient diversity* in Section A.3 (previously, this section was located in a different place in Appendix).

---

### Official Review · Reviewer_CrVx · 2024-11-08

**Soundness:** 3
**Presentation:** 3
**Contribution:** 3
**Rating:** 6
**Confidence:** 5

**Summary:**

The authors propose TabM, a variation of an MLP with $k$ heads that uses a shared backbone, and only certain aspects of the architecture in the beginning  and in the end are specialized. The heads are trained independently. The work is compared with prior DL methods (plain and attention architectures), retrieval-based methods, and tree-based methods on a diverse collection of classification and regression tasks. Based on the results, the proposed method manages to outperform all baselines with tuned hyperparameters.

**Strengths:**

- The proposed method achieves better performance compared to the other methods with tuned hyperparameters.
- The set of considered baselines is extensive.
- Extensive results on classification/regression datasets. Additionally, results are provided on well-known benchmarks in the domain.
- The authors provide extensive ablations of the different components of the method. The authors additionally provide a time comparison of the different methods.

**Weaknesses:**

- The work is difficult to read, the captions of Figure 2 and Figure 3 are long and blend with the core manuscript text, making it hard to keep track of the manuscript flow.

**Questions:**

- Table 2, for the MLP (Maps Routing dataset) the time unit is missing. I am guessing it is 10 minutes.

---

> ### Author Response · Authors · 2024-11-21
> **Reply**
>
> We thank the reviewer for the positive feedback.
>
> > the captions of Figure 2 and Figure 3 are long and blend with the core manuscript text
>
> We note two things:
> - Perhaps, the reviewer will be interested in the new illustration of the models in Figure 9. We are thinking of using Figure 9 instead of both Figure 1 and Figure 3. We are open to further feedback.
> - Our approach is to make figures somewhat self-sufficient, without constantly referencing the main text, which requires putting a bit of extra information to captions.
>
> > Table 2, for the MLP (Maps Routing dataset) the time unit is missing. I am guessing it is 10 minutes.
>
> This is fixed in the new revision, thanks! Yes, it is 10 minutes.

---

> ### Author Response · Authors · 2024-11-25
> **Invitation to discussion**
>
> Dear reviewer, is there anything preventing the verdict from being more positive? We will be glad to receive further feedback to improve the paper.

---

> > ### Comment · Reviewer_CrVx · 2024-11-27
> > **Rebuttal Response**
> >
> > I would like to thank the authors for the reply. I have read the individual reviews and the responses to the individual reviews from the authors.
> >
> > Regarding Figure 9, I am not sure, I guess I would need to see the final version of the manuscript. I mentioned Figure 1 and Figure 3, as it disrupted the flow of my reading. As the manuscript was not revised to adjust my concern, I leave it in good faith to the authors to improve upon it. This is also a matter of personal preference, in case I am the only one that observed an issue with it.
> >
> > Moreover, the motivation part of batch ensemble while interesting, was slightly confusing in my perspective. As in the name, batch ensemble features a batch, while the proposed method makes use of different views of the same training instance. Unless my understanding is not correct.
> >
> > Based on the above, I will retain my original score and recommend for acceptance.

---

### Author Response · Authors · 2024-11-21
**Global reply**

We thank the reviewers for their feedback and suggestions. In this post, we summarize the global context for further discussion:

1. **A summary of the positive comments** from the reviews.
2. **The big picture** as an easy reference for the discussion.

The PDF received a minor update to address small issues mentioned in the reviews. Plus, we added Figure 9 -- an additional illustration of the model.

## Positive things

- **Performance/effectiveness of the proposed method**
    - `CrVx`: *"The proposed method achieves better performance compared to the other methods ..."*
    - `4UNA`: *"The method is simple yet effective"*
    - `For5`: *"TabM demonstrates competitive performance ... efficient alternative for tabular data modeling"*
    - `aGcY`: *"The empirical results are quite strong"*
- **Model design**
    - `CrVx`: *"extensive ablations ... time comparison ..."*
    - `For5`: *"thorough analysis of each component's impact on model performance, contributing valuable insights."*
    - `aGcY`: *"interesting ablations"*
- **Empirical evaluation**
    - `CrVx`: *"The set of considered baselines is extensive ... extensive results on classification/regression datasets ... Additionally, results are provided on well-known benchmarks in the domain."*
    - `aGcY`: *"The empirical evaluation is broad and in-depth ... broad variety of baselines"*
- **Presentation**
    - `4UNA`: *"The paper is easy to read"*
    - `For5`: *"The paper is well-structured and clearly written, making the methods, results, and analysis easy to understand."*
    - `aGcY`: *"The empirical evaluation is ... well presented"*

## Big picture

It seems that the reviews do not fully agree on things like technical novelty and contribution scores. To find the common ground, we briefly summarize the big picture:

- `(A)` Our study identifies an overlooked opportunity to make a significant positive impact in the field of tabular DL architectures.
- `(B)` The technical context allows us to make this impact a reality without making a conceptual breakthrough, but we still *do* need to explore and identify the essential technical elements and ideas (see the details below).
- `(C)` The significant change in the performance-efficiency-simplicity coordinates, caused by TabM, *along with the issues described in the second paragraph of Introduction*, motivates us to perform a detailed reevaluation of tabular DL architectures. This gives a fresh and informative perspective on the modern state of the field and can have its own methodological impact on future studies (see details below).
- `(D)` We mainly focus on the above points. While we provide additional analysis, that we subjectively find interesting and important, we do not aim to explore all properties related to the ensemble-like nature of TabM in this one paper.

---

Additional details on the above points:

`(A)`
- The balance of task performance, simplicity, efficiency, and scalability, provided by TabM, is _not_ available in tabular DL architectures without TabM.
- Parameter-efficient ensembles have been known for many years, yet there has been no successful adaptation of this paradigm for tabular DL. Our study identifies and closes this gap.

`(B)` Examples of technical elements:

- Efficient ensembling
    - Seeing BatchEnsemble as a set of adapters that can be turned on/off.
    - The idea of "minimal" ensembling with only _one_ adapter per model compared to O(N) adapters for methods like BatchEnsemble and FiLM-Ensemble (N is the number of modified layers).
    - The idea of "safe" initialization of BatchEnsemble, directly inspired by the previous point.
- Tabular DL
    - The finding that the very first adapter is important for tabular models.

`(C)` In the story, this point is reflected in the abstract (*"renders the landscape of tabular DL in a new light"*) and contributions (*"fresh perspective on tabular DL"*). To give an in-depth perspective on results, we experiment with what to measure and how to present it for a given set of models and datasets. Examples:

- The fact that a sophisticated model performs no better or worse than plain MLP on 25% of tasks can significantly affect its positioning. In that regard, the middle and right parts of Figure 4 allow reasoning about the *weakest* parts of performance profiles. Namely, using the lower quantiles and individual data points, it is possible to see how *reliable* a model is on "inconvenient" datasets, how frequent and significant are its *failure modes*, etc.
- Another big factor is the efficiency-related properties of models. For example, when the task performance metrics from Figure 4 are combined with efficiency metrics from Figure 5, attention-based and retrieval-based models start looking less practical.

---

### Author Response · Authors · 2024-11-28
**To all reviewers (new revision)**

We thank the reviewers for taking part in the discussion!

**We have uploaded a new revision of the paper:**

- For the reviewer `CrVx`: we replaced the *two* model-related figures with *one* new figure, and placed it *at the top of the page*, which hopefully improves the presentation.
- For the reviewer `4UNA`, we extended the *analysis*:
    - We added Section A.2 with *analysis on the sources of TabM's power.*
    - We improved the text of the *analysis of the gradient diversity* in Section A.3 (previously, this section was located in a different place in Appendix).

P.S. The above changes could affect the figure and section enumerations used previously in the discussion.

---

### Meta-Review · Area_Chair_jFXc · 2024-12-19

**Metareview:**

The authors propose TabM an efficient ensembling architecture that uses MLPs as its backbone, where the main weight matrix of a linear layer is shared between the ensemble members, and only member-specific adapters are learned. The proposed method outperforms traditional ensembles of neural networks and it offers a faster runtime compared to the plain ensemble counterpart.

The experimental protocol features a comparison with an extensive number of baselines/datasets where the proposed method achieves the best overall performance. However, in terms of novelty, the proposed approach is somehow limited.

Initially, the reviewers had concerns regarding the efficiency of the proposed method and the complexity of the datasets included in the experimental protocol. However, the authors provided a thorough rebuttal. After the rebuttal the reviewers agree that the proposed method has a strong performance and it includes an extensive experimental protocol, yielding valuable results that would be helpful to the community.

I agree with the reviewers that the novelty of the proposed method is limited, however, the proposed work provides valuable insights and a strong baseline in the realm of tabular data. Based on the previous points, I recommend acceptance.

**Additional Comments On Reviewer Discussion:**

The authors engaged with the reviewers during the rebuttal. As a result of the thorough rebuttal responses by the authors, the reviewers reached a consensus in favor of accepting.

---

### Decision · Program_Chairs · 2025-01-22

Accept (Poster)